# CoInD: Enabling Logical Compositions in Diffusion Models

**Sachit Gaudi**   **Gautam Sreekumar**   **Vishnu Naresh Boddeti**
Michigan State University
{gaudisac,sreekum1,vishnu}@msu.edu

## Abstract

How can we learn generative models to sample data with arbitrary logical compositions of statistically independent attributes? The prevailing solution is to sample from distributions expressed as a composition of attributes' *conditional marginal distributions* under the assumption that they are statistically independent. This paper shows that standard conditional diffusion models violate this assumption, even when all attribute compositions are observed during training. And, this violation is significantly more severe when only a subset of the compositions is observed. We propose CoInD to address this problem. It explicitly enforces statistical independence between the *conditional marginal distributions* by minimizing Fisher's divergence between the *joint* and *marginal* distributions. The theoretical advantages of CoInD are reflected in both qualitative and quantitative experiments, demonstrating a significantly more faithful and controlled generation of samples for arbitrary logical compositions of attributes. The benefit is more pronounced for scenarios that current solutions relying on the assumption of *conditionally independent marginals* struggle with, namely, logical compositions involving the NOT operation and when only a subset of compositions are observed during training. Our code is available at https://github.com/sachit3022/compositional-generation/

## 1 Introduction

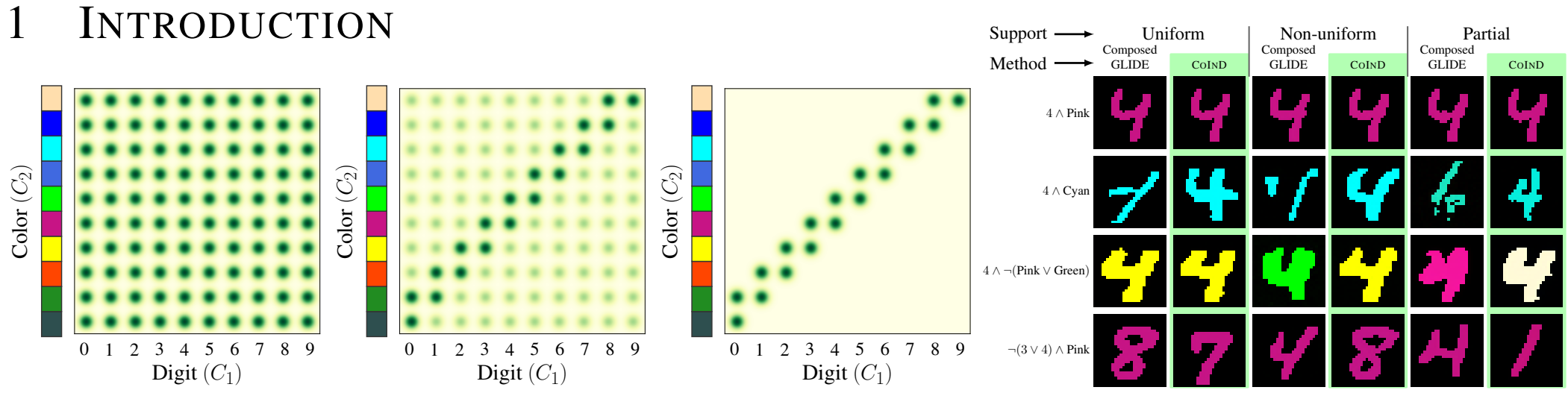

(a) Uniform (b) Non-uniform (c) Partial support (d) Generated samples

Figure 1: **Generative Modeling of Logical Compositions.** (a-c) Consider the task of generating MNIST samples for any logical composition of digits and colors by learning on observational data of different supports. (d) Standard diffusion models fail to generate data with arbitrary logical compositions of attributes. We generate data from simple unseen compositions (row 2), and more complex logical compositions (rows 3,4) through CoInD, even under non-uniform and partial support.

Many applications of generative models, including image editing (Kim et al., 2022; Brooks et al., 2023), desire explicit and independent control over statistically independent attributes. For example, in face generation, one might want to control the amount of hair, smile, etc., independently. This challenge relates to the broader task of logical compositionality in generative models, where the goal is to combine attributes according to logical relations. Consider the illustrative task in Fig. 1 of generating realistic samples of colored handwritten digits with explicit and independent control over the composition of color and digit. For example, "generate an image of digit 4 while excluding the colors green and pink". This composition can be logically expressed as "$4 \wedge \neg$[Green $\vee$ Pink]", where $\wedge$, $\vee$, and $\neg$ represent the three primitive logical operators AND, OR, and NOT, respectively.

Existing solutions (Liu et al., 2022; Du et al., 2020; Nie et al., 2021) realize this goal by mapping the logical expressions into a probability distribution involving the *conditional marginal distributions*

$p(\text{image} \mid \text{digit} = 4)$, $p(\text{image} \mid \text{color} \neq \text{Green})$, and $p(\text{image} \mid \text{color} \neq \text{Pink})$, and sampling from it. These *marginal* distributions are obtained either by learning separate energy-based models for each compositional attribute (Du et al., 2020; Nie et al., 2021) or by factorizing the attributes' learned *joint* distribution Liu et al. (2022). Both approaches, however, are predicated on the critical assumption that the conditional *marginal* distributions are statistically independent of each other.

Employing the approaches mentioned above, for instance Liu et al. (2022), to our illustrative example, we observe that when the conditional diffusion model is learned on data with non-uniform (Fig. 1b) or partial (Fig. 1c) support of the compositional attributes, the models fail to generate realistic samples (columns 3 and 5 of row 2 in Fig. 1d) or generate realistic samples with logically inaccurate compositions (columns 3 and 5 of rows 3 and 4 in Fig. 1d). This is true even for simple unseen logical compositions of attributes (AND in row 2 of Fig. 1d) or for complex logical compositions (rows 3 and 4 of Fig. 1d involving a NOT operation). Such failure under partial support was also observed by Du et al. (2020). Surprisingly, note that even when all compositions of the attributes are observed, the model fails to generate realistic samples (column 1 of row 2 in Fig. 1d).

These observations naturally raise the following research questions that this paper seeks to answer:

**– (RQ1)** *Why do standard classifier-free conditional diffusion models fail to generate data with arbitrary logical compositions of attributes?* We hypothesize that violating the assumption that the conditional marginal distributions are statistically independent of each other will result in poor image quality, diminished control over the generated image attributes, and, ultimately, failure to adhere to the desired logical composition. We verify and confirm our hypothesis through a case study in § 3.

**– (RQ2)** *How can we explicitly enable conditional diffusion models to generate data with arbitrary logical compositions of attributes?* We adopt the principle of independent causal mechanisms (Peters et al., 2017) to express the conditional data likelihood in terms of the constituent conditional *marginal* distributions to ensure that the model does not learn non-existent statistical dependencies.

**Summary of contributions.**

1. In Section 3, we show that conditional diffusion models trained to maximize the likelihood of the observed data do not learn independent conditional marginal distributions, even when all compositions of the attributes are uniformly (Fig. 1a) observed. Furthermore, this problem is exacerbated in more practical scenarios where we learn from non-uniform (Fig. 1b) or partial (Fig. 1c) support of the compositional attributes. Instead, the models learn non-existent statistical dependencies induced by unknown confounding factors.
2. Through causal modeling, we derive a training objective, COIND, comprising the standard score-matching loss and a conditional independence violation loss required to enforce the COnditional INdependence relations necessary for enabling logical compositions in conditional Diffusion models.
3. Strong inductive biases, in the form of the conditional independence relations in COIND, enable arbitrary logical compositionality in conditional diffusion models with fine-grained control over conditioned attributes and diversity for unconditioned attributes. COIND achieves these goals while being monolithic and is scalable with the number of attributes.

## 2 Logical Compositionality in Diffusion Models

We study the problem of generating data with attributes that satisfy a given logical relation between them. We consider the case where the attributes are statistically independent of each other. However, not all attribute compositions may be observed during training. To study this problem, we first model the underlying data-generation process using a suitable causal model that relates data and their independently varying attributes.

**Notations.** We use bold lowercase and uppercase characters to denote vectors (e.g., $\boldsymbol{a}$) and matrices (e.g., $\boldsymbol{A}$) respectively. Random variables are denoted by uppercase Latin characters (e.g., $X$). The distribution of a random variable $X$ is denoted as $p(X)$, or as $p_{\boldsymbol{\theta}}(X)$ if the distribution is parameterized by a vector $\boldsymbol{\theta}$. We adopt non-standard terminology where *marginals* denote the conditionals $p(\mathbf{X} \mid C_i)$ rather than integrated marginals, $p(C_i)$ emphasizing their functional role as modular components in our compositional framework. Correspondingly, *joint* refers to $p(\mathbf{X} \mid C)$, acknowledging this deliberate departure from probabilistic conventions due to a lack of better terminology.

**Data Generation Process.** The data generation process consists of observed data $\boldsymbol{X}$ (e.g., images) and its attribute variables $C_1, C_2, \ldots, C_n$ (e.g., color, digit, etc.). To have explicit control over these attributes during generation, they should vary independently of each other. In this work, we limit our study to only those causal graphs in which the attributes are not causally related and can hence vary independently, as shown in Fig. 2a. Each $C_i$ assumes values from a set $\mathcal{C}_i$ and their Cartesian product $\mathcal{C} = \mathcal{C}_1 \times \cdots \times \mathcal{C}_n$ is referred to as the *attribute space*. Each attribute $C_i$ generates its own observed component $\boldsymbol{X}_{C_i} = f_{C_i}(C_i)$, which together with unobserved exogenous variables $\boldsymbol{U}_{\boldsymbol{X}}$ form the composite observed data $\boldsymbol{X} = f(\boldsymbol{X}_{C_1}, \ldots, \boldsymbol{X}_{C_i}, \boldsymbol{U}_{\boldsymbol{X}})$ (see Fig. 2a). We do not restrict $f$ much except that it should not obfuscate individual observed components in $\boldsymbol{X}$ (Wiedemer et al., 2024). A simple example of $f$ is the concatenation function. We also assume that all $f_{C_i}$ are invertible and therefore it is possible to estimate $C_1, \ldots, C_n$ from $\boldsymbol{X}$. These assumptions together ensure that $C_1, \ldots, C_n$ are mutually independent given $\boldsymbol{X}$ despite being seemingly d-connected.

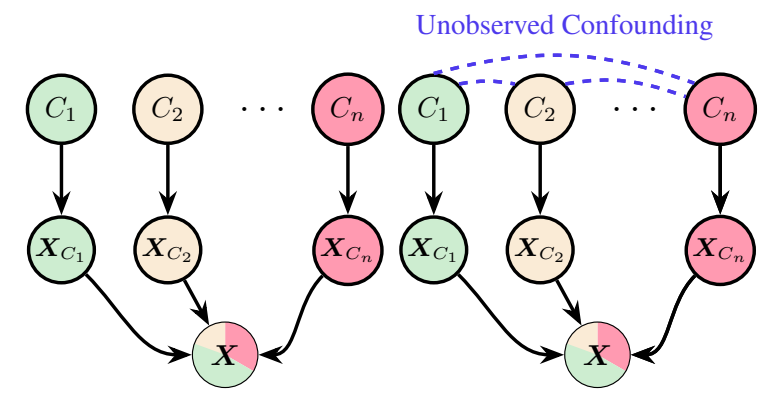

(a) True underlying causal model (b) Causal model during training

Figure 2: (a) $C_1, C_2, \ldots, C_n$ vary freely and independently in the underlying causal graph. (b) However, they become dependent during training due to unknown and unobserved confounding factors.

**Problem Statement.** When the training data is sampled according to the causal graph in Fig. 2a, all attribute compositions are equally likely to be observed. We refer to this scenario as *uniform support* (illustrated in Fig. 1a). However, real-world datasets often deviate from the independence due to unobserved confounders such as sample selection bias (Storkey, 2008), inducing an *attribute shift*. As shown in Fig. 2b, this shift modifies the causal structure during training through unobserved confounding relationships, resulting in *non-uniform support* (Fig. 1b) where attribute compositions exhibit unequal occurrence probabilities. In extreme cases, this dependence could lead to the training samples consisting of only a subset of all attribute compositions (Fig. 1c), i.e., $\mathcal{C}_{\text{train}} \subset \mathcal{C}$. We refer to this scenario as *partial support*. We aim to learn conditional diffusion models under these scenarios to generate samples with attributes that satisfy a given logical compositional relation between them.

The attribute space in our problem statement has the following properties. **(1)** The attribute space observed during training $\mathcal{C}_{\text{train}}$ covers $\mathcal{C}$ in the following sense:

**Definition 1** (Support Cover)**.** *Let $\mathcal{C} = \mathcal{C}_1 \times \cdots \times \mathcal{C}_n$ be the Cartesian product of $n$ finite sets $\mathcal{C}_1, \ldots, \mathcal{C}_n$. Consider a subset $\mathcal{C}_{train} \subset \mathcal{C}$, where $|\mathcal{C}_{train}| = m$. Let $\mathcal{C}_{train} = \{(c_{1j}, \ldots, c_{nj}) : c_{ij} \in \mathcal{C}_i, 1 \leq i \leq n, 1 \leq j \leq m\}$ and $\tilde{\mathcal{C}}_i = \{c_{ij} : 1 \leq j \leq m\}$ for $1 \leq i \leq n$. The Cartesian product of these sets is $\tilde{\mathcal{C}}_{train} = \tilde{\mathcal{C}}_1 \times \cdots \times \tilde{\mathcal{C}}_n$. We say $\mathcal{C}_{train}$ covers $\mathcal{C}$ iff $\mathcal{C} = \tilde{\mathcal{C}}_{train}$.*

Informally, this assumption implies that every possible value that $C_i$ can assume is present in the training set, and open-set attribute compositions do not fall under this definition. For instance, in the Colored MNIST example in Fig. 1, we are not interested in generating a digit with an unobserved $11^{\text{th}}$ color. **(2)** For every ordered tuple $c \in \mathcal{C}_{\text{train}}$, there is another $c' \in \mathcal{C}_{\text{train}}$ such that $c$ and $c'$ differ on only one attribute. Similar assumptions were discussed in (Wiedemer et al., 2024).

**Preliminaries on Score-based Models.** In this work, we train conditional score-based models (Song et al., 2021b) using classifier-free guidance (Ho & Salimans, 2022) to generate data corresponding to a given logical attribute composition. Score-based models learn the score of the observed data distributions $p_{\text{train}}(\boldsymbol{X})$ and $p_{\text{train}}(\boldsymbol{X} \mid C)$ through score matching (Hyvärinen, 2005). Once the score of a distribution is learned, samples can be generated using Langevin dynamics. For logical attribute compositional generation, the given attribute composition is decomposed in terms of two primitive logical compositions: (1) AND operation (e.g., $C_1 = c_1 \wedge C_2 = c_2$ generates data where attributes $C_1$ and $C_2$ takes values $c_1$ and $c_2$ respectively), and (2) NOT operation (e.g., $C_1 = \neg c_1$ generates data where the attribute $C_1$ takes any value except $c_1$). Liu et al. (2022) proposed the following modifications during sampling to enable AND and NOT logical operations between the attributes, assuming that the diffusion model learns the conditional independence relations from the underlying data-generation process, i.e., $p(C_1, \ldots, C_n|\boldsymbol{X}) = \prod_{i=1}^{n} p(C_i|\boldsymbol{X})$.

**Logical AND ($\wedge$) operation:** Since $p_{\boldsymbol{\theta}}(C_1 \wedge C_2 \mid \boldsymbol{X}) = p_{\boldsymbol{\theta}}(C_1 \mid \boldsymbol{X}) p_{\boldsymbol{\theta}}(C_2 \mid \boldsymbol{X})$ samples are generated for the logical composition $C_1 \wedge C_2$ by sampling from the following score:

$$\nabla_{\boldsymbol{X}} \log p_{\boldsymbol{\theta}}(\boldsymbol{X} \mid C_1 \wedge C_2) = \nabla_{\boldsymbol{X}} \log p_{\boldsymbol{\theta}}(\boldsymbol{X} \mid C_1) + \nabla_{\boldsymbol{X}} \log p_{\boldsymbol{\theta}}(\boldsymbol{X} \mid C_2) - \nabla_{\boldsymbol{X}} \log p_{\boldsymbol{\theta}}(\boldsymbol{X}) \quad (1)$$

**Logical NOT ($\neg$) operation:** Following the approximation $p_{\boldsymbol{\theta}}(\neg C_2 \mid \boldsymbol{X}) \propto \frac{1}{p_{\boldsymbol{\theta}}(C_2|\boldsymbol{X})}$, the score to sample data for the logical composition $C_1 \wedge \neg C_2$ can be expressed as,

$$\nabla_{\boldsymbol{X}} \log p_{\boldsymbol{\theta}}(\boldsymbol{X} \mid C_1 \wedge \neg C_2) = \nabla_{\mathbf{X}} \log p_{\boldsymbol{\theta}}(\boldsymbol{X}) + \nabla_{\mathbf{X}} \log p_{\boldsymbol{\theta}}(\boldsymbol{X} \mid C_1) - \nabla_{\mathbf{X}} \log p_{\boldsymbol{\theta}}(\boldsymbol{X} \mid C_2) \quad (2)$$

**Precise Control:** To achieve precise control over attribute composition, the hyperparameter $\gamma$ is used to modulate the relative intensity of attribute $C_2$ with respect to $C_1$. We sample from the distribution, $\nabla_{\boldsymbol{X}} \log p_{\boldsymbol{\theta}}(\boldsymbol{X} \mid C_1 \wedge \uparrow C_2)$, expressed as

$$\nabla_{\boldsymbol{X}} \log p_{\boldsymbol{\theta}}(\boldsymbol{X} \mid C_1) + \gamma \nabla_{\boldsymbol{X}} \log p_{\boldsymbol{\theta}}(\boldsymbol{X} \mid C_2) - \gamma \nabla_{\boldsymbol{X}} \log p_{\boldsymbol{\theta}}(\boldsymbol{X}) \quad (3)$$

**Logical OR ($\vee$) operation:** From the rules of Boolean algebra, $C_1 \vee C_2$ operation can be expressed in terms of $\wedge$ and $\neg$ as $\neg(\neg C_1 \wedge \neg C_2)$. Following the approximation for $\neg$ from above, it follows that $p(\neg(\neg C_1 \wedge \neg C_2)) \approx p(C_1)p(C_2)$.

For example, to generate colored handwritten digits with the "$4 \wedge \neg$[Green $\vee$ Pink]" logical composition, the score of the logical composition can be decomposed into its constituent logical primitive operations and further in terms of the score of marginals, which can be obtained from the trained diffusion models. Therefore, $\nabla_{\mathbf{X}} \log p_\theta(\boldsymbol{X} \mid 4 \wedge \neg\, [\text{G} \vee \text{P}])$ is given by:

$$= \nabla_{\mathbf{X}} \log p_\theta(\boldsymbol{X} \mid C_1 = 4 \wedge C_2 = \neg\text{G}) + \nabla_{\mathbf{X}} \log p_\theta(\boldsymbol{X} \mid C_1 = 4 \wedge C_2 = \neg\text{P}) - \nabla_{\mathbf{X}} \log p_\theta(\boldsymbol{X})$$
$$= 2\nabla_{\mathbf{X}} \log p_\theta(\boldsymbol{X} \mid C_1 = 4) - \nabla_{\mathbf{X}} \log p_\theta(\boldsymbol{X} \mid C_2 = \text{G}) - \nabla_{\mathbf{X}} \log p_\theta(\boldsymbol{X} \mid C_2 = \text{P}) + \nabla_{\mathbf{X}} \log p_\theta(\boldsymbol{X})$$

**Note** that the scores to sample from these primitive logical compositions involve conditional *marginal* likelihood terms $\boldsymbol{X} \mid C_i$. Therefore, to perform logical composition, it is critical to accurately learn the conditional *marginals* of the attributes.

**Evaluation** We evaluate the distributions learned by the model based on their accuracy in generating images with attributes that align with the desired compositions for a logical relation. For example, to evaluate AND ($\wedge$) composition, consider sampling an arbitrary digit and color, represented as $C = (4, \text{Cyan})$. We generate images $\hat{X}$ by sampling from Eq. (1), and subsequently infer attributes, $(\hat{c}_1, \hat{c}_2) = (\phi_{C_1}(\hat{X}), \phi_{C_2}(\hat{X}))$. We then verify if $(\hat{c}_1, \hat{c}_2) \subseteq \{4\} \times \{\text{Cyan}\}$, and this process is averaged over all combinations in $\mathcal{C}$ to obtain CS.

**Conformity Score (CS)** To formally define CS: For a logical relation, $R$. This relation is defined as a boolean function over the attribute space $\mathcal{C}$, such that $R : \mathcal{C} \to \{0, 1\}$. This induces a constrained attribute space given by $\mathcal{R} = \{(c_1, \ldots, c_n) \mid R(c_1, \ldots, c_n) = 1\} \subseteq \mathcal{C}$. The CS is defined as:

$$\text{CS}(R, \theta) := \mathbb{E}_{C \sim p(C)} \mathbb{E}_{U \sim p(U)} \left[ \mathbb{1}_{\mathcal{R}_C} \left( (\phi_{C_i}(g_\theta(R(C), U)))_{i=1}^n \right) \right] \quad (4)$$

where $R(C)$ can represent various logical operations such as $\wedge$, $\neg$, and $\vee$ on the attribute space $\mathcal{C}$. Here, $g_\theta(R(C), U)$ denotes a generative model parameterized by $\theta$, which samples according to the logical relations specified above. The variable $U$ represents exogenous noise in the diffusion model. The functions $\phi_{C_i}$ are attribute-specific classifiers that infer attributes from the generated images. The term $\mathbb{1}_{\mathcal{R}_C}$ , is an indicator function, equals 1 if the inferred attributes $(\phi_{C_i}(g_\theta(R(C), U)))_{i=1}^n \subseteq \mathcal{R}_C$. Further details regarding the Conformity Score can be found in App. D.6.

## 3 Why do conditional diffusion models fail to generate data with arbitrary logical compositions of attributes?

To address **(RQ1)**, we utilize the task of generating synthetic images from the Colored MNIST dataset for *any* given combination of color and digit, as introduced in § 1. To study the effect of data support, we consider the three training distributions of attribute compositions defined in § 2: *(1) uniform support*, where every ordered pair in $\mathcal{C}$ has an equal chance of being observed (Fig. 1a), *(2) non-uniform support*, where every ordered pair in $\mathcal{C}$ appears but with unequal probabilities (Fig. 1b), and *(3) partial support*, where only subset of ordered pairs, $\mathcal{C}_{\text{train}} \subset \mathcal{C}$ are observed (Fig. 1c).

For each support, we train a diffusion model and evaluate the conditional *joint*, $p_{\boldsymbol{\theta}}(X \mid C)$ and *marginal*, $p_{\boldsymbol{\theta}}(X \mid C_i)$ distributions. During inference, the images are separately sampled from the *joint* distribution, $\nabla_{\boldsymbol{X}} \log p_\theta(\boldsymbol{X} \mid C)$, and from the product of the learned *marginals* as shown in Eq. (1). We refer to the former method as *joint sampling* and the latter as *marginal* sampling. To

| Support | Conformity Score | | JSD ↓ |
|---|---|---|---|
| | *Joint* ↑ | *Marginal* ↑ | |
| Uniform | 99.98 | 98.15 | 0.16 |
| Non Uniform | 99.98 | 86.10 | 0.30 |
| Partial | 33.14 | 7.40 | 2.75 |

Table 1: **Conformity Scores** and **Jensen-Shannon divergence** for samples generated from joint and marginal distributions learned by models under various support settings for the Colored MNIST dataset.

measure the accuracy of the attributes in the generated image in accordance to the desired attributes, we use conformity score (CS) defined in § 2. Tab. 1 compares the *joint* and the *marginal* distributions learned by models trained under various training scenarios. We draw the following conclusions.

**Diffusion models struggle to generate unseen attribute compositions.** From the conformity scores of images sampled from the *joint* distribution, we conclude that while the models trained with uniform and non-uniform support generate images with accurate attribute compositions, those trained with partial support struggle to generate images for unseen attribute compositions. The standard training objective of diffusion models is to maximize the likelihood of conditional generation, for every observed attribute composition, the model accurately learns $p_{\text{train}}(\boldsymbol{X} \mid C)$, i.e., $p_{\boldsymbol{\theta}}(\boldsymbol{X} \mid C) \approx p_{\text{train}}(\boldsymbol{X} \mid C)$ However, with partial support, the model does not observe samples for every attribute composition from $p_{\text{train}}(\boldsymbol{X} \mid C)$. Therefore, the model does not accurately learn the density of the unobserved support region.

**Diffusion models violate underlying Conditional Independence relations.** Although the diffusion model is trained on all *marginals* $(X \mid C_i)$, per the support cover assumption, *marginals* samples performs inferior to that of sampling from the *joint* distribution. This further drop in conformity score when sampled from the product of *marginals* ( Eq. (1)) for the models trained under non-uniform and partial support settings is due to the disparity between the *joint* distribution and the product of *marginals*, which points to the violation of independence relations from the underlying data-generation process in the learned model. Refer to App. B.1 for a detailed proof. To further strengthen the claim, we measure this violation as the disparity between the conditional *joint* distribution $p_{\boldsymbol{\theta}}(C \mid \boldsymbol{X})$ and the product of conditional *marginal* distributions $\prod_i^n p_{\boldsymbol{\theta}}(C_i \mid \boldsymbol{X})$ learned by the guidance term in a model using Jensen-Shannon divergence (JSD):

$$\text{JSD} = \mathbb{E}_{C,\boldsymbol{X}\sim p_{\text{data}}}\left[D_{\text{JS}}\left(p_{\boldsymbol{\theta}}(C \mid \boldsymbol{X}) \,||\, \prod_i^n p_{\boldsymbol{\theta}}(C_i \mid \boldsymbol{X})\right)\right] \tag{5}$$

where $D_{\text{JS}}$ is the Jensen-Shannon divergence and following (Li et al., 2023) $p_{\boldsymbol{\theta}}$ is obtained by evaluating the implicit classifier learned by the diffusion model. More details can be found in App. D.7.

A positive JSD value suggests that the model fails to adhere to the independence relations present in the underlying causal model. Our findings (Tab. 1) indicate that as the training distribution of attribute compositions diverges from the true underlying distribution – where attributes vary independently – the trained models increasingly violate independence relations, as reflected by the JSD. These findings demonstrate diffusion models lack inherent compositional bias, instead propagate dependencies as present in their training data.

**Training objective of the diffusion models is not suitable for logical compositionality.** The objective of the diffusion models trained with classifier-free guidance is to maximize the conditional likelihood of power-set of attributes. However, due to confounding induced by the training support (Fig. 2b), the attributes become dependent during training, i.e., $p_{\text{train}}(C_1, \ldots, C_n) \neq \prod_{i=1}^n p_{\text{train}}(C_i)$. As a result, the conditional distribution of *marginals* does not match its true underlying distribution. i.e $p_\theta(\boldsymbol{X} \mid C_i) \approx p_{\text{train}}(\boldsymbol{X} \mid C_i) \neq p_{\text{data}}(\boldsymbol{X} \mid C_i)$ Refer to App. B.2 for formal proof. Therefore, any method (Nie et al., 2021) that relies on training on these incorrect *marginals* or relies on conditional independence (Liu et al., 2022) is bound to fail. Moreover, even when realistic samples of unseen composition are successfully generated, it is by accident rather than design.

⚠ Failure of Logical Compositionality

Standard conditional diffusion models trained with classifier-free guidance struggle to generate data with arbitrary logical compositions of attributes because they violate the independence relations inherent in the causal data-generation process.

Based on these observations, we propose CoInD to train diffusion models that explicitly enforce the conditional independence dictated by the underlying causal data-generation process to encourage the model to learn accurate *marginal* distributions of the attributes.

# 4 COIND: ENFORCING CONDITIONALLY INDEPENDENT MARGINAL TO ENABLE LOGICAL COMPOSITIONALITY

In this section, we propose COIND to answer **(RQ2)** posed in § 1: *How can we explicitly enable conditional diffusion models to generate data with arbitrary logical compositions of attributes?*

In the previous section, we observed that diffusion models do not obey the underlying causal relations, learning incorrect attribute *marginals*, and hence struggling to demonstrate logical compositionally as we showed in Fig. 1. To remedy this, COIND uses a training objective that explicitly enforces the causal factorization to ensure that the trained diffusion models obey the underlying causal relations. From the causal graph Fig. 2a, along with the assumption of $C_1 \perp\!\!\!\perp \ldots \perp\!\!\!\perp C_n \mid \boldsymbol{X}$ mentioned in § 2, we have $p(\boldsymbol{X} \mid C) = \frac{p(\boldsymbol{X})}{p(C)} \prod_i^n \frac{p(\boldsymbol{X}|C_i)p(C_i)}{p(\boldsymbol{X})}$. Note that the invariant $p(\boldsymbol{X} \mid C)$ is now expressed as the product of *marginals* employed for sampling. Therefore, training the diffusion model by maximizing this conditional likelihood is naturally more suited for learning accurate *marginals* for the attributes. We minimize the distance between the true conditional likelihood and the learned conditional likelihood as,

$$\mathcal{L}_{\text{comp}} = \mathcal{W}_2\left(p(\boldsymbol{X} \mid C), \frac{p_{\boldsymbol{\theta}}(\boldsymbol{X})}{p_{\boldsymbol{\theta}}(C)} \prod_i \frac{p_{\boldsymbol{\theta}}(\boldsymbol{X} \mid C_i)p_{\boldsymbol{\theta}}(C_i)}{p_{\boldsymbol{\theta}}(\boldsymbol{X})}\right) \tag{6}$$

where $\mathcal{W}_2$ is 2-Wasserstein distance. Applying the triangle inequality to Eq. (6) we have,

$$\mathcal{L}_{\text{comp}} \leq \underbrace{\mathcal{W}_2\left(p(\boldsymbol{X} \mid C), p_{\boldsymbol{\theta}}(\boldsymbol{X} \mid C)\right)}_{\text{Distribution matching}} + \underbrace{\mathcal{W}_2\left(p_{\boldsymbol{\theta}}(\boldsymbol{X} \mid C), \frac{p_{\boldsymbol{\theta}}(\boldsymbol{X})}{p_{\boldsymbol{\theta}}(C)} \prod_i^n \frac{p_{\boldsymbol{\theta}}(\boldsymbol{X} \mid C_i)p_{\boldsymbol{\theta}}(C_i)}{p_{\boldsymbol{\theta}}(\boldsymbol{X})}\right)}_{\text{Conditional Independence}} \tag{7}$$

(Kwon et al., 2022) showed that the Wasserstein distance between $p_0(\boldsymbol{X}), q_0(\boldsymbol{X})$ is upper bounded by the square root of the score-matching objective.

$$\mathcal{W}_2\left(p_0(\boldsymbol{X}), q_0(\boldsymbol{X})\right) \leq K\sqrt{\mathbb{E}_{p_0(\boldsymbol{X})}\left[||\nabla_{\boldsymbol{X}} \log p_0(\boldsymbol{X}) - \nabla_{\boldsymbol{X}} \log q_0(\boldsymbol{X})||_2^2\right]}$$

**Distribution matching:** Following this result, the first term in Eq. (7) is upper bounded by the standard score-matching objective of diffusion models (Song et al., 2021b),

$$\mathcal{L}_{\text{score}} = \mathbb{E}_{p(\boldsymbol{X},C)}\|\nabla_{\boldsymbol{X}} \log p_{\boldsymbol{\theta}}(\boldsymbol{X} \mid C) - \nabla_{\boldsymbol{X}} \log p(\boldsymbol{X} \mid C)\|_2^2 \tag{8}$$

**Conditional Independence:** Similarly, the second term in Eq. (7) is upper bounded by score-matching between the *joint* and product of *marginals*

$$\mathcal{L}_{\text{CI}} = \mathbb{E}\|\nabla_{\mathbf{X}} \log p_\theta(\mathbf{X} \mid C) - \nabla_{\mathbf{X}} \log p_\theta(\mathbf{X}) - \sum_i \left[\nabla_{\mathbf{X}} \log p_\theta(\mathbf{X} \mid C_i) - \nabla_{\mathbf{X}} \log p_\theta(\mathbf{X})\right]\|_2^2 \tag{9}$$

Substituting Eq. (8), Eq. (9) in Eq. (7) will result in our final learning objective

$$\mathcal{L}_{\text{comp}} \leq K_1\sqrt{\mathcal{L}_{\text{score}}} + K_2\sqrt{\mathcal{L}_{\text{CI}}} \tag{10}$$

where $K_1, K_2$ are positive constants, i.e., the conditional independence objective $\mathcal{L}_{\text{CI}}$ is incorporated alongside the existing score-matching loss $\mathcal{L}_{\text{score}}$.

$\mathcal{L}_{\text{CI}}$, is the Fisher divergence between the joint and the product of marginals. From the properties of Fisher's divergence Sánchez-Moreno et al. (2012). $\mathcal{L}_{\text{CI}} = 0$ iff $p_\theta(\boldsymbol{X} \mid C) = \frac{p_\theta(\boldsymbol{X})}{p_\theta(C)} \prod_i^n \frac{p_\theta(\boldsymbol{X}|C_i)p_\theta(C_i)}{p_\theta(\boldsymbol{X})}$. Detailed derivation of the upper bound can be found in App. B.3.

**Practical Implementation.** A computational burden presented by $\mathcal{L}_{\text{CI}}$ in Eq. (9) is that the required number of model evaluations increases linearly with the number of attributes. To mitigate this burden, we approximate the mutual conditional independence with pairwise conditional independence (Hammond & Sun, 2006). Thus, the modified $\mathcal{L}_{\text{CI}}$ becomes,

$$\mathcal{L}_{\text{CI}} = \mathbb{E}_{p(\boldsymbol{X},C)}\mathbb{E}_{j,k}\|\nabla_{\boldsymbol{X}} \log p_\theta(\boldsymbol{X} \mid C_j, C_k) - \nabla_{\boldsymbol{X}} \log p_\theta(\boldsymbol{X} \mid C_j) - \nabla_{\boldsymbol{X}} \log p_\theta(\boldsymbol{X} \mid C_k) + \nabla_{\boldsymbol{X}} \log p_\theta(\boldsymbol{X})\|_2^2$$

The weighted sum of the square of the terms in Eq. (10) has shown stability. Therefore, COIND's training objective:

$$\mathcal{L}_{\text{final}} = \mathcal{L}_{\text{score}} + \lambda\mathcal{L}_{\text{CI}} \tag{11}$$

where $\lambda$ is the hyper-parameter that controls the strength of conditional independence. The reduction to the practical version of the upper bound (Eq. (10)) is discussed in extensively in App. C. For guidance on selecting hyper-parameters in a principled manner, please refer to App. C.3. Finally, our proposed approach can be implemented with just a few lines of code, as outlined in Algorithm 1.

## 5 DOES COIND IMPROVE THE LOGICAL COMPOSITIONALITY?

COIND encourages diffusion models to learn conditionally independent *marginals* of attributes, and thereby improve their logical compositionality capabilities. In this section, we design experiments to evaluate COIND on two questions: (1) *Does* COIND *effectively train diffusion models that obey the underlying causal model?*, and (2) *Does* COIND *improve the logical compositionality of these models?* We measure the JSD of the trained models to answer the first question. To answer the second question, we use two primitive logical compositional tasks: (a) $\wedge$ (AND) composition and (b) $\neg$ (NOT) composition. In each case, the generative model is provided with a logical relation between the attributes, and the task is to generate images with attributes that satisfy this logical relation. A more detailed description of task construction can be found in App. D.2. Beyond improved logical compositionality, we ask: Does learning conditionally independent *marginals* lead to greater diversity in uncontrolled attributes and enhanced controllability of attributes?

**Datasets.** We use the following image datasets with labeled attributes for our experiments: **(1) Colored MNIST** dataset described in § 1, where the attributes of interest are digit and color, **(2) Shapes3d** dataset (Kim & Mnih, 2018) containing images of 3D objects in various environments where each image is labeled with six attributes of interest. **(3) CelebA** with gender and smile attributes demonstrates effectiveness of COIND on real-world datasets. Refer to App. D.5.

**Observed training distributions.** We evaluate COIND on four scenarios where we observe different distributions of attribute compositions during training: *(1) Uniform support*, *(2) Non-uniform support (3) Diagonal partial support*, as defined in § 2. *(4) Orthogonal partial support* includes only the attribute compositions along the axes originating from a corner of the hypercube $\mathcal{C}$, following (Wiedemer et al., 2024) (Fig. 3). For Colored MNIST experiments, we evaluate with uniform, non-uniform, and diagonal partial support. For Shapes3d experiments, we evaluate with uniform and orthogonal partial support, following the compositional setup in (Schott et al., 2020). We evaluate CelebA on orthogonal partial support, where all compositions except unseen male smiling celebrities are observed.

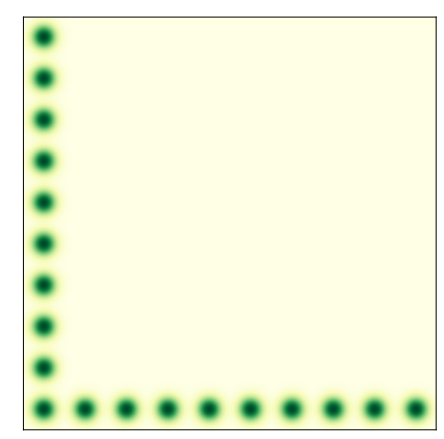

Figure 3: Orthogonal partial support

**Baselines.** **LACE** (Nie et al., 2021) and **Composed GLIDE** (Liu et al., 2022) are our primary baselines. LACE trains distinct energy-based models (EBMs) for each attribute and combines them following the compositional logic described in § 2 during sampling. A similar approach was proposed by (Du et al., 2020). However, in our experimental evaluation for LACE, we train distinct score-based models instead of EBMs. In contrast, Composed GLIDE samples from score-based models by factorizing the joint distribution into marginals, assuming these models had implicitly learned conditionally independent marginals of attributes. Additional details about the baselines are delegated to App. D.3.

**Metrics.** We assess how accurately the models have captured the underlying data generation process using the JSD, defined in § 3. To measure the accuracy of the attributes in the generated image w.r.t. the input logical composition, we use conformity score (CS) from § 2. As a reminder, CS measures the accuracy with which the model adds the desired attributes to the generated image using attribute-specific classifiers. In addition to the conformity score, since the Shapes3d dataset contains unique ground truth images corresponding to the input logical relation, we directly compare generated samples with reference images at the pixel level using the variance-weighted coefficient of determination, $R^2$. Additionally, for CelebA, we measure FID (Seitzer, 2020). We evaluate uniform and non-uniform support on the generations for the input logical relations correspond to attribute compositions that span the attribute space $\mathcal{C}$. In other cases, we evaluate models ability to generate input logical relations corresponding to the unseen compositional support, i.e., $\mathcal{C} \setminus \mathcal{C}_{\text{train}}$.

☞ **Learning Independent Marginals Enables Logical Compositionality**

Fig. 4a compares COIND with baselines on $\wedge$ and $\neg$ composition tasks. The "$\neg$ Color" task generates images with the negation applied on color attribute, while"$\neg$ Digit" applies the negation to the digit attribute. From these results, we make the following observations:

**Conditional diffusion models do not learn accurate marginals even when all attribute compositions are observed during training with equal probability.** This is evident from the positive JSD

| Support | Method | JSD ↓ | ∧ (CS) ↑ | ¬ Color (CS)↑ | ¬ Digit (CS) ↑ |
|---|---|---|---|---|---|
| Uniform | LACE | - | 96.40 | 92.56 | 83.67 |
| | Composed GLIDE | 0.16 | 98.15 | 99.30 | 81.64 |
| | CoInD ($\lambda = 0.2$) | 0.14 | 99.73 | 99.32 | 84.94 |
| | CoInD ($\lambda = 1.0$) | **0.10** | **99.99** | **99.33** | **89.60** |
| Non-uniform | LACE | - | 82.61 | 65.16 | 69.51 |
| | Composed GLIDE | 0.30 | 86.10 | 81.61 | 70.44 |
| | CoInD ($\lambda = 1.0$) | **0.15** | **99.95** | **92.41** | **84.98** |
| Partial | LACE | - | 10.85 | 9.03 | 28.24 |
| | Composed GLIDE | 2.75 | 7.40 | 5.09 | 33.86 |
| | CoInD ($\lambda = 1.0$) | **1.17** | **52.38** | **53.28** | **52.59** |

(a) Results on Colored MNIST Dataset

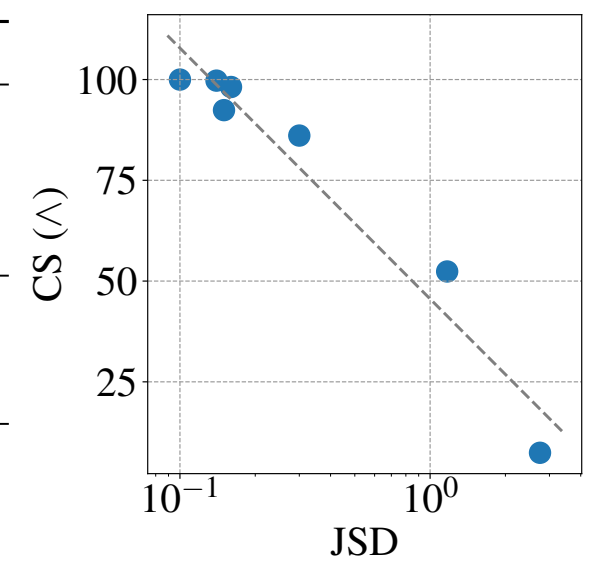

(b) JSD vs CS

Figure 4: **Results on Colored MNIST dataset.** (a) We compare JSD and CS of CoInD against baselines trained under various settings and on different compositional tasks. (b) Plotting CS against JSD in the log scale of the models trained under different settings reveals a negative correlation.

of the methods trained with uniform support. Furthermore, the conformity score (CS) is lower when JSD is higher. This observation has significant ramifications for compositional generative models.

*This result contradicts the intuitive expectation that uniformly observing the whole compositional support during training is sufficient to generate arbitrary logical compositions of attributes.* And, it suggests that even in this ideal yet impractical case, the current objectives for training diffusion models are insufficient for controllable and accurate closed-set, let alone open-set, compositional generation. As such, we conjecture that scaling the datasets without inductive biases (conditional independence of marginals in this case) is insufficient for arbitrary logical compositional generation.

*Even methods like LACE that train separate models for each attribute fail on ¬ composition tasks.* This suggests that softer inductive biases, such as learning separate marginals for each attribute without paying heed to the desired independence relations, are insufficient for logical compositionality.

In the more practical scenarios of non-uniform and partial support, JSD increases with non-uniform support and worsens further with partial support due to incorrect marginals as discussed in § 3. This result suggests that current state-of-the-art models learned on finite datasets likely operate in the non-uniform or partial support scenario and thus may fail to generate accurate and realistic data for arbitrary logical compositions of attributes.

*Logical AND (∧) and NOT (¬) compositionality deteriorates with increasing dependence between the marginals.* The negative correlation between JSD and CS was noted in § 3 and can be observed in Fig. 4b, which shows JSD-vs-CS for ∧ compositions across different methods, and under different settings for observed support. This negative correlation strongly suggests that violation of conditional independence plays a major role in the diminished logical compositionality demonstrated by standard diffusion models.

By enforcing conditional independence between the attributes during training, CoInD achieves lower JSD and improves both ∧ and ¬ compositionality in non-uniform and partial support. Even when trained on non-uniform support, CoInD matches compositionality with the uniform support in terms of compositional score. Under partial support setting, CoInD achieves $\approx 2 - 10\times$ fold improvement over the baselines on ∧ and ¬ compositions. These results demonstrate that **enforcing conditional independence between the marginals is vital for enabling arbitrary logical compositions in conditional diffusion models.**

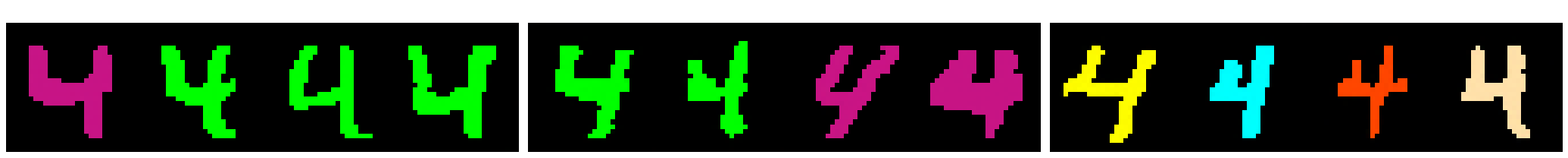

(a) LACE; $H = 1.82$ (b) Composed GLIDE; $H = 1.71$ (c) CoInD; $H = 2.63$

Figure 5: Images generated by CoInD for the logical composition digit $= 4$ under non-uniform scenario are significantly diverse compared to the baselines. $H$ is the Shannon entropy.

☞ **CoInD generates diverse samples.** It is desirable that any attribute not part of the logical composition for generation assumes diverse values in the generated samples to avoid harmful generated content—including stereotypes (Dehdashtian et al., 2025) and biases (Luccioni et al., 2024). In Fig. 5, we observe that although CoInD does not explicitly optimize for diversity, the samples generated by CoInD for the logical relation digit $= 4$ are significantly more diverse compared

to the baselines. We quantitatively measure the diversity of these images using the Shannon entropy $H$ of the color attributes in the generated images. Higher Shannon entropy indicates more diversity. Entropy is maximum for a uniform distribution with $H(\text{uniform}) = \log_2(10) = 3.32$, since there are 10 colors. We observe that $H(\text{CoInD}) = 2.63$, while $H(\text{LACE}) = 1.82$, $H(\text{Composed GLIDE}) = 1.71$. Although CoInD does not explicitly seek diversity, breaking the dependence induced by unknown confounders exhibits diversity in attributes.

| Support | Method | JSD ↓ | ∧ Composition | | ¬ Composition | |
|---|---|---|---|---|---|---|
| | | | $R^2$ ↑ | CS ↑ | $R^2$↑ | CS↑ |
| Uniform | LACE | - | 0.97 | 91.19 | 0.85 | 50.00 |
| | Composed GLIDE | 0.302 | 0.94 | 83.75 | 0.91 | 48.43 |
| | CoInD ($\lambda = 1.0$) | **0.215** | **0.98** | **95.31** | **0.92** | **55.46** |
| Orthogonal | LACE | - | 0.88 | 62.07 | 0.70 | 30.10 |
| | Composed GLIDE | 0.503 | 0.86 | 51.56 | 0.61 | 34.63 |
| | CoInD ($\lambda = 1.0$) | **0.287** | **0.97** | **91.10** | **0.92** | **53.90** |

(a) Quantitative Results on Shapes3D Dataset

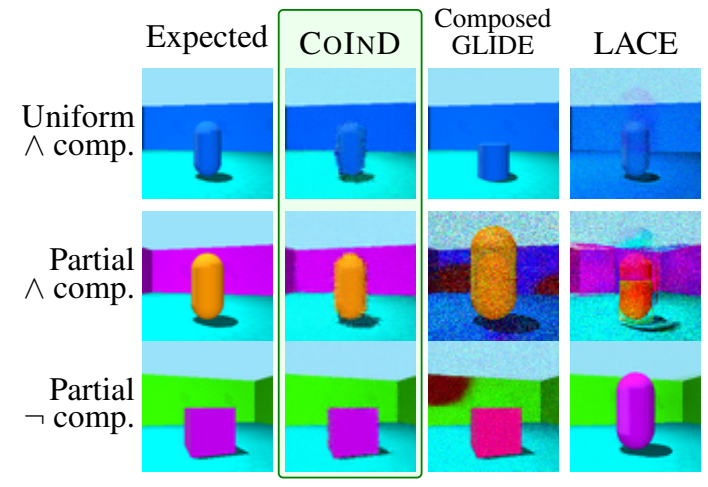

(b) Visual comparison of samples

Figure 6: **Results on Shapes3d dataset.** (a) We compare JSD, $R^2$, and CS of CoInD against the baselines trained with uniform and partial support on the Shapes3d dataset for ∧ and ¬ composition tasks. (b) Samples generated by CoInD match the expected image in all cases.

☞ **CoInD is scalable with attributes.** We use the Shapes3d dataset to evaluate the scalability of CoInD w.r.t. the number of attributes. As a reminder, every image in the Shapes3d dataset is labeled with six attributes of interest. For the negation composition task, the ¬ operator is applied to the shape attribute such that the attribute composition satisfying this logical relation is unique. Detailed descriptions of the composition tasks are provided in App. D.2. Fig. 6a compares CoInD against the baselines for the uniform and orthogonal partial support scenarios. CoInD leads to a significant decrease in JSD and, consequently, a significant increase in the composition score. When trained with orthogonal support, the performance (CS) of both LACE and Composed GLIDE suffers significantly while CoInD matches its performance when trained on uniform support. In conclusion, CoInD affords superior logical compositionality from a single monolithic model in a sample-efficient manner even as the number of attributes increases.

☞ **CoInD generates unseen compositions of real-world face images**

We evaluate ability of CoInD to generate unseen "smiling male celebrities". Diffusion model is trained on all compositions of the CelebA dataset (Liu et al., 2015) except gender = "male" and smiling = "true". This is equivalent to the orthogonal support scenario shown in Fig. 3. During inference, the model is asked to generate images with the unseen attribute combination gender = "male" and smiling = "true" through both *joint* sampling and ∧ composition.

| Method | JSD ↓ | "smiling male" | | "smiling"∧"male" | |
|---|---|---|---|---|---|
| | | CS ↑ | FID ↓ | CS ↑ | FID ↓ |
| LACE | - | - | - | **24.20** | 80.40 |
| Composed GLIDE | 2.44 | 2.51 | 61.21 | 10.55 | 95.41 |
| CoInD ($\lambda = 100$) | **1.82** | **8.63** | **43.97** | 8.79 | **43.76** |

Table 2: **Results on CelebA dataset.** CoInD outperforms the baselines on both CS and FID across various compositionality tasks.

Tab. 2 compares CoInD against baselines in terms of CS and FID. **(1)** CoInD outperforms the baseline by $> 4\times$ in *joint*. **(2)** CoInD generates realistic faces, closer to smiling male celebrities in the held out set, as measured by FID and displayed in Fig. 7c($\gamma = 1$). In App. E.3, we show that CoInD extends to Text-to-Image models by fine-tuning Stable Diffusion (Rombach et al., 2022).

☞ **CoInD provides fine-grained control over attributes.** In addition to merely generating samples with conditioned attributes, CoInD can also control the *amount of attributes* in the sample. For example, in the task of generating face images of smiling male celebrities, we may wish to adjust the amount of smiling without affecting gender-specific attributes. To achieve this, we sample from Eq. (3), where $\gamma$ controls the strength of smile. Fig. 7 shows the result of increasing $\gamma$ to increase the amount of smiling in the generated image. The subjects in the face images generated by CoInD smile more as $\gamma$ increases without any changes to any gender-specific attribute. For instance, the images for $\gamma = 1$ show a soft smile while the subjects in the images for $\gamma = 6$ show teeth. However, those generated by baselines contain gender-specific attributes such as long hair and earrings. These distinctions are quantified in Figs. 7a and 7b. Refer to App. E.2 for more analysis.

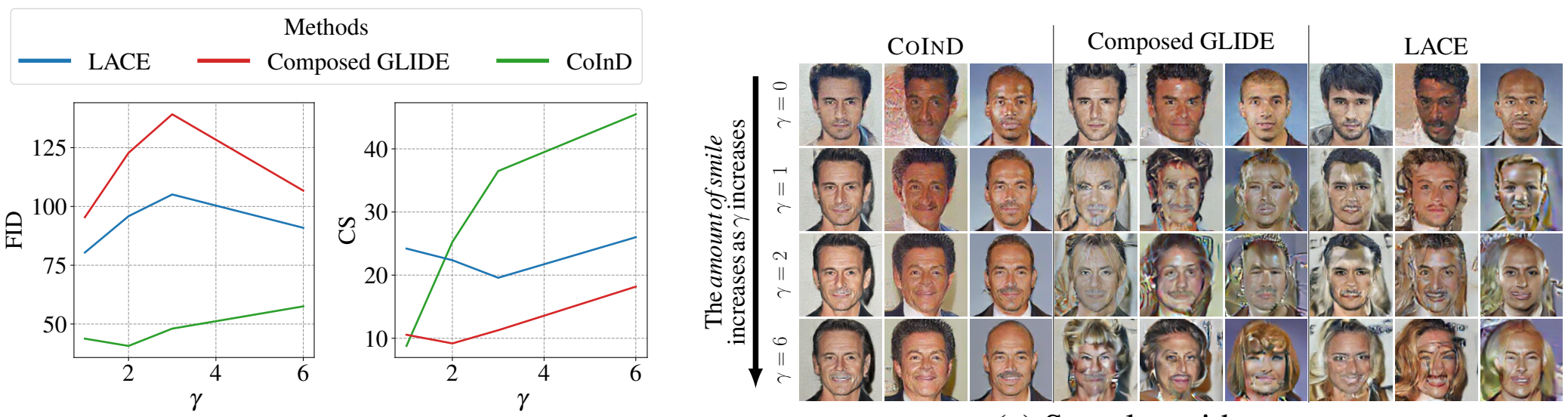

(a) FID with $\gamma$ (b) CS with $\gamma$ (c) Samples with $\gamma$

Figure 7: **Effect of $\gamma$ on FID and CS:** Varying the amount of smile in a generated image through $\gamma$ does not affect the FID of CoInD. However, the smiles in the generated images become more apparent, leading to easier detection by the smile classifier and improved CS.

## 6 Related Work

Our work concerns compositional generalization in generative models, where the goal is to generate data with unseen attribute compositions expressed through logical relations between attributes. One class of approaches achieves logical compositionality by combining distinct models trained for each attribute (Du et al., 2020; Liu et al., 2021; Nie et al., 2021; Du et al., 2023). In contrast, we are interested in monolithic diffusion models that learn logical compositionality. Besides being expensive and scaling linearly with the number of attributes, these models fail under practical partial support scenarios. Liu et al. (2022) studied logical compositionality broadly without differentiating between attribute supports and proposed methods to represent logical compositions in terms of *marginal* probabilities obtained through factorization of the *joint* distribution. However, these factorized sampling methods fail since the underlying generative model learns inaccurate *marginals*. In comparison, CoInD is trained to obey the independence relations from the underlying causal graph. Also, (Cho et al., 2024) note that diffusion models lack the conditional independence needed for controllability and address this with a hyperparameter during sampling. We argue that, even with disentangled features, learning accurate *marginals* tackles the root cause more effectively than such post-hoc adjustments. Encouragingly, Okawa et al. (2023) shows that compositional abilities emerge multiplicatively, and Liang et al. (2024) highlights factorization in diffusion models, suggesting they naturally exhibit compositional capabilities. However, these studies focus on generating from the *joint* distribution—a special case of logical compositionality—and are limited to binary attributes. Our work extends these ideas to more general compositions. Lastly, (Wiedemer et al., 2024) studies compositional generalization for supervised learning and provides sufficient conditions for compositionality. Our empirical observations in generative models are consistent with their theoretical results, suggesting that their findings could perhaps be extended to conditional diffusion models.

## 7 Conclusion

Conditional diffusion models struggle to generate data for arbitrary attribute compositions, even when all attribute compositions are observed during training. Existing methods represent logical relations in terms of the learned *marginal* distributions, assuming that the diffusion model learns the underlying conditional independence relations. We showed that this assumption does not hold in practice and worsens when only a subset of these attribute compositions are observed during training. To mitigate this problem, we proposed CoInD to train diffusion models by maximizing conditional data likelihood in terms of the *marginal* distributions that are obtained from the underlying causal graph. Our causal modeling provides CoInD a natural advantage in logical compositionality by ensuring it learns accurate *marginals*. Our experiments on synthetic and real image datasets highlight the theoretical benefits of CoInD. Unlike existing methods, CoInD is monolithic, easy to implement, and demonstrates superior logical compositionality. CoInD shows that adequate inductive biases such as conditional independence between *marginals* are *necessary* for effective logical compositionality. Refer to Apps. F and G for more discussions, analysis and limitations of CoInD.

**Acknowledgements:** This work was supported by the Office of Naval Research (award #N00014-23-1-2417). Any opinions, findings, and conclusions or recommendations expressed in this material are those of the authors and do not necessarily reflect the views of ONR. We also thank Mashrur Morshed for his insights on improving diffusion model training and for sharing the bare-bones code, Lan Wang for providing the code for fine-tuning Stable Diffusion, and Ramin Akbari for his assistance with the proofs presented in Apps. B and C.

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

# Appendix

## Table of Contents

# A PRELIMINARIES OF SCORE-BASED MODELS

**Score-based models** Score-based models (Song et al., 2021b) learn the score of the observed data distribution, $p_{\text{train}}(\boldsymbol{X})$ through score matching (Hyvärinen, 2005). The score function $s_{\boldsymbol{\theta}}(\mathbf{x}) = \nabla_{\mathbf{x}} \log p_{\boldsymbol{\theta}}(\mathbf{x})$ is learned by a neural network parameterized by $\boldsymbol{\theta}$.

$$L_{\text{score}} = \mathbb{E}_{\mathbf{x}\sim p_{\text{train}}}\left[\|s_{\boldsymbol{\theta}}(\mathbf{x}) - \nabla_{\mathbf{x}} \log p_{\text{train}}(\mathbf{x})\|_2^2\right] \tag{12}$$

During inference, sampling is performed using Langevin dynamics:

$$\mathbf{x}_t = \mathbf{x}_{t-1} + \frac{\eta}{2}\nabla_{\mathbf{x}} \log p_{\boldsymbol{\theta}}(\mathbf{x}_{t-1}) + \sqrt{\eta}\epsilon_t, \quad \epsilon_t \sim \mathcal{N}(0,1) \tag{13}$$

where $\eta > 0$ is the step size. As $\eta \to 0$ and $T \to \infty$, the samples $\mathbf{x}_t$ converge to $p_{\boldsymbol{\theta}}(\boldsymbol{X})$ under certain regularity conditions (Welling & Teh, 2011).

**Diffusion models** Song & Ermon proposed a scalable variant that involves adding noise to the data. Ho et al. has shown its equivalence to Diffusion models. Diffusion models are trained by adding noise to the image $\mathbf{x}$ according to a noise schedule, and then neural network, $\epsilon_\theta$ is used to predict the noise from the noisy image, $\mathbf{x_t}$. The training objective of the diffusion models is given by:

$$L_{\text{score}} = \mathbb{E}_{\mathbf{x}\sim p_{\text{train}}}\mathbb{E}_{t\sim[0,T]}\left\|\epsilon - \epsilon_\theta\left(\mathbf{x}_t, t\right)\right\|^2 \tag{14}$$

Here, the perturbed data $\mathbf{x}_t$ is expressed as: $\mathbf{x}_t = \sqrt{\bar{\alpha}_t}\mathbf{x} + \sqrt{1-\bar{\alpha}_t}\epsilon$ where $\bar{\alpha}_t = \prod_{i=1}^{T}\alpha_i$, for a pre-specified noise schedule $\alpha_t$. The score can be obtained using,

$$s_\theta(\mathbf{x}_t, t) \approx -\frac{\epsilon_\theta(\mathbf{x}_t, t)}{\sqrt{1-\bar{\alpha}_t}} \tag{15}$$

Langevin dynamics can be used to sample from the $s_\theta(\mathbf{x}_t, t)$ to generate samples from $p(\boldsymbol{X})$. The conditional score (Dhariwal & Nichol, 2021) is used to obtain samples from the conditional distribution $p_{\boldsymbol{\theta}}(\boldsymbol{X} \mid C)$ as:

$$\nabla_{\boldsymbol{X}_t} \log p(\boldsymbol{X}_t \mid C) = \underbrace{\nabla_{\boldsymbol{X}_t} \log p_{\boldsymbol{\theta}}(\boldsymbol{X}_t)}_{\text{Unconditional score}} + \gamma\nabla_{\boldsymbol{X}_t} \log \underbrace{p_{\boldsymbol{\theta}}(C \mid \boldsymbol{X}_t)}_{\text{noisy classifier}}$$

where $\gamma$ is the classifier strength. Instead of training a separate noisy classifier, Ho & Salimans have extended to conditional generation by training $\nabla_{\boldsymbol{X}_t} \log p_{\boldsymbol{\theta}}(\boldsymbol{X}_t \mid C) = s_\theta(\boldsymbol{X}_t, t, C)$. The sampling can be performed using the following equation:

$$\nabla_{\boldsymbol{X}_t} \log p(\boldsymbol{X}_t \mid C) = (1-\gamma)\nabla_{\boldsymbol{X}_t} \log p_{\boldsymbol{\theta}}(\boldsymbol{X}_t) + \gamma\nabla_{\boldsymbol{X}_t} \log p_{\boldsymbol{\theta}}(\boldsymbol{X}_t \mid C) \tag{16}$$

However, the sampling needs access to unconditional scores as well. Instead of modelling $\nabla_{\boldsymbol{X}_t} \log p_{\boldsymbol{\theta}}(\boldsymbol{X}_t)$, $\nabla_{\boldsymbol{X}_t} \log p_{\boldsymbol{\theta}}(\boldsymbol{X}_t|C)$ as two different models Ho & Salimans have amortize training a separate classifier training a conditional model $s_\theta(\mathbf{x}_t, t, \mathbf{c})$ jointly with unconditional model trained by setting $c = \varnothing$.

In the general case of classifier-free guidance, a single model can be effectively trained to accommodate all subsets of attribute distributions. During the training phase, each attribute $c_i$ is randomly set to $\varnothing$ with a probability $p_{\text{uncond}}$. This approach ensures that the model learns to match all possible subsets of attribute distributions. Essentially, through this formulation, we use the same network to model all the possible subsets of conditional probability.

Once trained, the model can generate samples conditioned on specific attributes, such as $c_i$ and $c_j$, by setting all other conditions to $\varnothing$. The conditional score is then computed as, $\nabla_{\boldsymbol{X}_t} \log p_{\boldsymbol{\theta}}(\boldsymbol{X}_t|c_i, c_j) = \mathbf{x}_t, \mathbf{c}^{i,j})$, where $\mathbf{c}^{i,j}$ represents the condition vector with all values other than $i$ and $j$ set to $\varnothing$. This method allows for flexible and efficient sampling across various attribute combinations.

**Estimating Guidance** Once the diffusion model is trained, we investigate the implicit classifier, $p_\theta(C|\boldsymbol{X})$, learned by the model. This will give us insights into the learning process of the diffusion models. (Li et al., 2023) have shown a way to calculate $p_\theta(C_i = c_i \mid \boldsymbol{X} = \mathbf{x})$, borrowing equation (5), (6) from (Li et al., 2023).

$$p_\theta(C_i = c_i \mid \mathbf{x}) = \frac{p(c_i)\, p_\theta(\mathbf{x} \mid c_i)}{\sum_k p(c_k)\, p_\theta(\mathbf{x} \mid c_k)}$$

$$p_\theta(C_i = c_i \mid \mathbf{x}) = \frac{\exp\{-\mathbb{E}_{t,\epsilon}[\|\epsilon - \epsilon_\theta(\mathbf{x}_t, t, \mathbf{c}^i)\|^2]\}}{\mathbb{E}_{C_i}\left[\exp\{-\mathbb{E}_{t,\epsilon}[\|\epsilon - \epsilon_\theta(\mathbf{x}_t, t, \mathbf{c}^i)\|^2]\}\right]} \tag{17}$$

Likewise, we can extend it to joint distribution by

$$p_\theta(C_i = c_i, C_j = c_j \mid \mathbf{x}) = \frac{\exp\{-\mathbb{E}_{t,\epsilon}[\|\epsilon - \epsilon_\theta(\mathbf{x}_t, t, \mathbf{c}^{i,j})\|^2]\}}{\mathbb{E}_{C_i,C_j}\left[\exp\{-\mathbb{E}_{t,\epsilon}[\|\epsilon - \epsilon_\theta(\mathbf{x}_t, t, \mathbf{c}^{i,j})\|^2]\}\right]} \tag{18}$$

**Practical Implementation** The authors Li et al.. have showed many axproximations to compute $\mathbb{E}_{t,\epsilon}$. However, we use a different approximation inspired by Kynkäänniemi et al. (2024), where we sample 5 time-steps between [300,600] instead of these time-steps spread over the [0, T].

# B Proofs for Claims

In this section, we detail the mathematical derivations for case study from § 3 in App. B.1, relate the origin of the conditional independence violation to the unsuitable loss function of vanilla diffusion models in App. B.2, and then derive the final loss function of CoInD in App. B.3.

## B.1 Proof for the case study in § 3

In this section, we prove that failure of compositionality in diffusion models is due to the violation of conditional independence.

Following conditional independence relation:

$$p(C \mid X) = \prod_i p(C_i \mid X) \tag{CI relation}$$

This CI relation is used by several works (Liu et al., 2022; Nie et al., 2021), including ours, to derive the expression for the joint distribution $p(X \mid C)$ in terms of the marginals $p(X \mid C_i)$ for logical compositionality. As a reminder, logical compositionality is preferred over simple conditional generation as it (1) provides fine-grained control over the attributes, (2) facilitates NOT relations on attributes, and (3) is more interpretable. The joint likelihood is written in terms of the marginals using the CI relation and the causal factorization as,

$$p(X \mid C) = \frac{p(X)}{p(C)} \prod_i \left( \frac{p(X \mid C_i) p(C_i)}{p(X)} \right) \tag{JM relation}$$

Note that CI relation is crucial for JM relation to hold. We sample from joint likelihood using the score of LHS of JM relation, referred to as joint sampling in § 3. Similarly, we sample using the score of RHS of JM relation, referred to as marginal sampling in § 3. If the learned generative model satisfies the JM relation, then there should not be any difference in the CS between joint sampling and marginal sampling. However, in Tab. 1, we see a drop in CS, implying JM relation is not satisfied in the learned model.

JM relation must hold in the learned generative model if CI relation is true in the learned generative model. Therefore, we check if the CI relation holds in the generative model by measuring JSD between LHS and RHS of CI relation as shown in Eq. (5) in the main paper. The results Tab. 1 confirm that the CI relation does not hold in the learned model. This is a significant finding since existing works (Liu et al., 2022; Nie et al., 2021) blindly trust the model to satisfy CI relation, leading to severe performance drop when the training support is non-uniform or partial.

The CI relation is violated in the learned model because the standard training objective is not suitable for compositionality, as it does not account for the incorrect $p_{\text{train}}(X \mid C_i)$. The proof is detailed in the next section App. B.2. Therefore, we proposed CoInD to ensure the JM relation was satisfied by explicitly learning the marginal likelihood according to the causal factorization.

## B.2 Standard diffusion model objective is not suitable for logical compositionality

This section proves that the violation in conditional independence in diffusion models is due to learning incorrect marginals, $p_{\text{train}}(\boldsymbol{X} \mid C_i)$ under $C_i \not\perp\!\!\!\perp C_j$. We leverage the causal invariance

property: $p_{\text{train}}(\boldsymbol{X} \mid C) = p_{\text{true}}(\boldsymbol{X} \mid C)$, where $p_{\text{train}}$ is the training distribution and $p_{\text{true}}$ is the true underlying distribution.

Consider the training objective of the score-based models in classifier free formulation Eq. (12). For the classifier-free guidance, a single model $s_{\boldsymbol{\theta}}(\mathbf{x}, C)$ is effectively trained to match the score of all subsets of attribute distributions. Therefore, the effective formulation for classifier-free guidance can be written as,

$$L_{\text{score}} = \mathbb{E}_{\mathbf{x}\sim p_{\text{train}}}\mathbb{E}_S\left[\|\nabla_{\mathbf{x}} \log p_{\boldsymbol{\theta}}(\mathbf{x} \mid c_S) - \nabla_{\mathbf{x}} \log p_{\text{train}}(\mathbf{x} \mid c_S)\|_2^2\right] \tag{19}$$

where $S$ is the power set of attributes.

From the properties of Fisher divergence, $L_{\text{score}} = 0$ iff $p_{\boldsymbol{\theta}}(\boldsymbol{X} \mid c_S) = p_{\text{train}}(\boldsymbol{X} \mid c_S), \forall S$. In the case of marginals, $p_{\boldsymbol{\theta}}(\boldsymbol{X} \mid C_i)$ i.e. $S = \{C_i\}$ for some $1 \leq i \leq n$,

$$\begin{aligned}
p_{\boldsymbol{\theta}}(\boldsymbol{X} \mid C_i) &= p_{\text{train}}(\boldsymbol{X} \mid C_i) \\
&= \sum_{C_{-i}} p_{\text{train}}(\boldsymbol{X} \mid C_i, C_{-i}) p_{\text{train}}(C_{-i} \mid C_i) \\
&= \sum_{C_{-i}} p_{\text{true}}(\boldsymbol{X} \mid C_i, C_{-i}) p_{\text{train}}(C_{-i} \mid C_i) \\
&\neq \sum_{C_{-i}} p_{\text{true}}(\boldsymbol{X} \mid C_i, C_{-i}) p_{\text{true}}(C_{-i}) = p_{\text{true}}(\boldsymbol{X} \mid C_i) \\
\implies p_{\boldsymbol{\theta}}(\boldsymbol{X} \mid C_i) &\neq p_{\text{true}}(\boldsymbol{X} \mid C_i)
\end{aligned} \tag{20}$$

Where $C_{-i} = \prod_{\substack{j=1 \\ j\neq i}}^{n} C_j$, which is every attribute except $C_i$. Therefore, the objective of the score-based models is to maximize the likelihood of the marginals of training data and not the true marginal distribution, which is different from the training distribution when $C_i \not\perp C_j$.

### B.3 Step-by-step derivation of CoInD in § 4

The objective is to train the model by explicitly modeling the joint likelihood following the causal factorization from Eq. (JM relation). The minimization for this objective can be written as,

$$\mathcal{L}_{\text{comp}} = \mathcal{W}_2\left(p(\boldsymbol{X} \mid C), \frac{p_{\boldsymbol{\theta}}(\boldsymbol{X})}{p_{\boldsymbol{\theta}}(C)} \prod_i \frac{p_{\boldsymbol{\theta}}(\boldsymbol{X} \mid C_i) p_{\boldsymbol{\theta}}(C_i)}{p_{\boldsymbol{\theta}}(\boldsymbol{X})}\right) \tag{21}$$

where $\mathcal{W}_2$ is 2-Wasserstein distance. Applying the triangle inequality to Eq. (21) we have,

$$\mathcal{L}_{\text{comp}} \leq \underbrace{\mathcal{W}_2\left(p(\boldsymbol{X} \mid C), p_{\boldsymbol{\theta}}(\boldsymbol{X} \mid C)\right)}_{\text{Distribution matching}} + \underbrace{\mathcal{W}_2\left(p_{\boldsymbol{\theta}}(\boldsymbol{X} \mid C), \frac{p_{\boldsymbol{\theta}}(\boldsymbol{X})}{p_{\boldsymbol{\theta}}(C)} \prod_i^n \frac{p_{\boldsymbol{\theta}}(\boldsymbol{X} \mid C_i) p_{\boldsymbol{\theta}}(C_i)}{p_{\boldsymbol{\theta}}(\boldsymbol{X})}\right)}_{\text{Conditional Independence}} \tag{22}$$

(Kwon et al., 2022) showed that under some conditions, the Wasserstein distance between $p_0(\boldsymbol{X}), q_0(\boldsymbol{X})$ is upper bounded by the square root of the score-matching objective. Rewriting Equation 16 from (Kwon et al., 2022)

$$\mathcal{W}_2\left(p_0(\boldsymbol{X}), q_0(\boldsymbol{X})\right) \leq K\sqrt{\mathbb{E}_{p_0(\boldsymbol{X})}\left[||\nabla_{\boldsymbol{X}} \log p_0(\boldsymbol{X}) - \nabla_{\boldsymbol{X}} \log q_0(\boldsymbol{X})||_2^2\right]} \tag{23}$$

**Distribution matching** Following Eq. (23) result, the first term in Eq. (22), replacing $p_0$ as $p$ and $q_0$ as $p_\theta$ will result in

$$\begin{aligned}
\mathcal{W}_2\left(p(\boldsymbol{X} \mid C), p_{\boldsymbol{\theta}}(\boldsymbol{X} \mid C)\right) &\leq K_1\sqrt{\mathbb{E}_{p_0(\boldsymbol{X})}\left[||\nabla_{\boldsymbol{X}} \log p(\boldsymbol{X} \mid C) - \nabla_{\boldsymbol{X}} \log p_\theta(\boldsymbol{X})||_2^2\right]} \\
&= K_1\sqrt{\mathcal{L}_{\text{score}}}
\end{aligned} \tag{24}$$

**Conditional Independence** Following Eq. (23) result, the second term in Eq. (22), replacing $p_0$ as $p_\theta$ and $q_0(\boldsymbol{X})$ as $\frac{p_{\boldsymbol{\theta}}(\boldsymbol{X})}{p_{\boldsymbol{\theta}}(C)} \prod_i^n \frac{p_{\boldsymbol{\theta}}(\boldsymbol{X}|C_i)p_{\boldsymbol{\theta}}(C_i)}{p_{\boldsymbol{\theta}}(\boldsymbol{X})}$

$$\mathcal{W}_2\left(p_{\boldsymbol{\theta}}(\boldsymbol{X} \mid C), \frac{p_{\boldsymbol{\theta}}(\boldsymbol{X})}{p_{\boldsymbol{\theta}}(C)} \prod_i^n \frac{p_{\boldsymbol{\theta}}(\boldsymbol{X} \mid C_i)p_{\boldsymbol{\theta}}(C_i)}{p_{\boldsymbol{\theta}}(\boldsymbol{X})}\right)$$
$$\leq \sqrt{\mathbb{E}\|\nabla_{\mathbf{X}} \log p_\theta(\boldsymbol{X} \mid C) - \nabla_{\mathbf{X}} \log \frac{p_{\boldsymbol{\theta}}(\boldsymbol{X})}{p_{\boldsymbol{\theta}}(C)} \prod_i^n \frac{p_{\boldsymbol{\theta}}(\boldsymbol{X} \mid C_i)p_{\boldsymbol{\theta}}(C_i)}{p_{\boldsymbol{\theta}}(\boldsymbol{X})}\|_2^2}$$

Further simplifying and incorporating $\nabla_{\mathbf{X}} \log p_\theta(C_i) = 0$ and $\nabla_{\mathbf{X}} \log p_\theta(C) = 0$ will result in

$$\mathcal{W}_2\left(p_{\boldsymbol{\theta}}(\boldsymbol{X} \mid C), \frac{p_{\boldsymbol{\theta}}(\boldsymbol{X})}{p_{\boldsymbol{\theta}}(C)} \prod_i^n \frac{p_{\boldsymbol{\theta}}(\boldsymbol{X} \mid C_i)p_{\boldsymbol{\theta}}(C_i)}{p_{\boldsymbol{\theta}}(\boldsymbol{X})}\right)$$
$$\leq K_2 \sqrt{\underbrace{\mathbb{E}\|\nabla_{\mathbf{X}} \log p_\theta(\boldsymbol{X} \mid C) - \nabla_{\mathbf{X}} \log p_\theta(\boldsymbol{X}) - \sum_i \left[\nabla_{\mathbf{X}} \log p_\theta(\boldsymbol{X} \mid C_i) - \nabla_{\mathbf{X}} \log p_\theta(\boldsymbol{X})\right] \|_2^2}_{\mathcal{L}_{\text{CI}}}}$$
$$= K_2 \sqrt{\mathcal{L}_{\text{CI}}} \tag{25}$$

Substituting Eq. (24), Eq. (25) in Eq. (22) will result in our final learning objective

$$\mathcal{L}_{\text{comp}} \leq K_1 \sqrt{\mathcal{L}_{\text{score}}} + K_2 \sqrt{\mathcal{L}_{\text{CI}}} \tag{26}$$

where $K_1, K_2$ are positive constants, i.e., the conditional independence objective $\mathcal{L}_{\text{CI}}$ is incorporated alongside the existing score-matching loss $\mathcal{L}_{\text{score}}$.

Note that Eq. (25) is the Fisher divergence between the joint $p_\theta(\boldsymbol{X} \mid C)$ and the causal factorization $\frac{p_\theta(X)}{p_\theta(C)} \prod_i \frac{p_\theta(\boldsymbol{X}|C_i)p_\theta(C_i)}{p_\theta(\boldsymbol{X})}$ from Eq. (JM relation). From the properties of Fisher divergence (Sánchez-Moreno et al., 2012), $\mathcal{L}_{\text{CI}} = 0$ iff $p_\theta(\boldsymbol{X} \mid C) = \frac{p_\theta(\boldsymbol{X})}{p_\theta(C)} \prod_i^n \frac{p_\theta(\boldsymbol{X}|C_i)p_\theta(C_i)}{p_\theta(\boldsymbol{X})}$ and further implying, $\prod_i p_\theta(C_i \mid \boldsymbol{X}) = p_{\text{train}}(C \mid \boldsymbol{X})$

When $L_{\text{comp}} = 0$: $P_\theta(\boldsymbol{X} \mid C) = P_{\text{train}}(\boldsymbol{X} \mid C) = P(\boldsymbol{X} \mid C)$, and $\prod_i p_\theta(C_i \mid \boldsymbol{X}) = p_{\text{train}}(C \mid \boldsymbol{X})$. This implies that the learned marginals obey the causal independence relations from the data-generation process, leading to more accurate marginals.

## C Practical Considerations

To facilitate scalability and numerical stability for optimization, we introduce two approximations to the upper bound of our objective function Eq. (10).

### C.1 Scalability of $\mathcal{L}_{\text{CI}}$

A key computational challenge posed by Eq. (9) is that the number of model evaluations grows linearly with the number of attributes. The Eq. (9) is derived from conditional independence formulation as follows:

$$p_\theta(C \mid X) = \prod_i p_\theta(C_i \mid X). \tag{27}$$

By applying Bayes' theorem to all terms, we obtain,

$$\frac{p_\theta(\boldsymbol{X} \mid C)p_\theta(C)}{p_\theta(X)} = \prod_i \frac{p_\theta(\boldsymbol{X} \mid C_i)p_\theta(C_i)}{p_\theta(\boldsymbol{X})} \tag{28}$$

Note that this formulation is equal to the causal factorization. From this, by applying logarithm and differentiating w.r.t. $\boldsymbol{X}$, we derive the score formulation.

$$\nabla_{\boldsymbol{X}} \log p_\theta(\boldsymbol{X} \mid C) = \nabla_{\boldsymbol{X}} \log \sum_i p_\theta(\boldsymbol{X} \mid C_i) - \nabla_{\boldsymbol{X}} \log p_\theta(\boldsymbol{X}) \tag{29}$$

The $L_2$ norm of the difference between LHS and RHS of the objective in Eq. (29) is given by, which forms our $\mathcal{L}_{\text{CI}}$ objective.

$$\mathcal{L}_{\text{CI}} = \|\nabla_{\boldsymbol{X}} \log p_\theta(\boldsymbol{X} \mid C) - \left(\nabla_{\boldsymbol{X}} \log \sum_i p_\theta(\boldsymbol{X} \mid C_i) - \nabla_{\boldsymbol{X}} \log p_\theta(\boldsymbol{X})\right)\|_2^2 \tag{30}$$

Due to the $\sum_i$, in the equation, the number of model evaluations grows linearly with the number of attributes ($n$). This $\mathcal{O}(n)$ computational complexity hinders the approach's applicability at scale. To address this, we leverage the results of (Hammond & Sun, 2006), which shows conditional independence is equivalent to pairwise independence under large $n$ to reduce the complexity to $\mathcal{O}(1)$ in expectation. This allows for a significant improvement in scalability while maintaining computational efficiency. Using this result, we modify Eq. (27) to:

$$p_\theta(C_i, C_j \mid X) = p_\theta(C_i \mid X) p_\theta(C_j \mid X). \quad \forall i, j$$

Accordingly, we can simplify the loss function for conditional independence as follows:

$$\mathcal{L}_{\text{CI}} = \mathbb{E}_{p(\boldsymbol{X},C)} \mathbb{E}_{j,k} \|\nabla_X [\log p_\theta(X|C_j, C_k) - \log p_\theta(X|C_j) - \log p_\theta(X|C_k) + \log p_\theta(X)]\|_2^2. \tag{31}$$

In score-based models, which are typically neural networks, the final objective is given as:

$$\mathcal{L}_{\text{CI}} = \mathbb{E}_{p(\boldsymbol{X},C)} \mathbb{E}_{j,k} \|s_{\boldsymbol{\theta}}(\boldsymbol{X}, C_j, C_k) - s_{\boldsymbol{\theta}}(\boldsymbol{X}, C_j) - s_{\boldsymbol{\theta}}(\boldsymbol{X}, C_k) + s_{\boldsymbol{\theta}}(\boldsymbol{X}, \varnothing)\|_2^2 \tag{32}$$

where $s_{\boldsymbol{\theta}}(\cdot) := \nabla_{\boldsymbol{X}} \log p_\theta(\cdot)$ is the score of the distribution modeled by the neural network. We leverage classifier-free guidance to train the conditional score $s_{\boldsymbol{\theta}}(\boldsymbol{X}, C_i)$ by setting $C_k = \varnothing$ for all $k \neq i$, and likewise for $s_{\boldsymbol{\theta}}(\boldsymbol{X}, C_i, C_j)$, we set $C_k = \varnothing$ for all $k \notin \{i, j\}$.

### C.2 Simplification of Theoretical Loss

In Eq. (10), we showed that the 2-Wassertein distance between the true joint distribution $p(\boldsymbol{X} \mid C)$ and the causal factorization in terms of the marginals $p(\boldsymbol{X} \mid C_i)$ is upper bounded by the weighted sum of the square roots of $\mathcal{L}_{\text{score}}$ and $\mathcal{L}_{\text{CI}}$ as $\mathcal{L}_{\text{comp}} \leq K_1\sqrt{\mathcal{L}_{\text{score}}} + K_2\sqrt{\mathcal{L}_{\text{CI}}}$. In practice, however, we minimized a simple weighted sum of $\mathcal{L}_{\text{score}}$ and $\mathcal{L}_{\text{CI}}$, given by $\mathcal{L}_{\text{final}} = \mathcal{L}_{\text{score}} + \lambda\mathcal{L}_{\text{CI}}$ as shown in Eq. (11) instead of Eq. (10). We used Eq. (11) to avoid the instability caused by larger gradient magnitudes (due to the square root). Eq. (11) also provided the following practical advantages: **(1)** the simplicity of the loss function that made hyperparameter tuning easier, and **(2)** the similarity of Eq. (11) to the loss functions of pre-trained diffusion models allowing us to reuse existing hyperparameter settings from these models. We did not observe any significant difference in conclusion between the models trained on Eq. (10) and Eq. (11) as shown in Tabs. 3 and 4. Both approaches significantly outperformed the baselines.

| Support | Method | JSD ↓ | ∧ (CS) ↑ | ¬ Color (CS)↑ | ¬ Digit (CS) ↑ |
|---|---|---|---|---|---|
| | LACE | - | 96.40 | 92.56 | 83.67 |
| | Composed GLIDE | 0.16 | 98.15 | 99.30 | 81.64 |
| Uniform | Theoretical CoInD Eq. (10) | 0.12 | 98.44 | **100.00** | 81.25 |
| | CoInD ($\lambda = 0.2$) | 0.14 | 99.73 | 99.32 | 84.94 |
| | CoInD ($\lambda = 1.0$) | **0.10** | **99.99** | 99.33 | **89.60** |
| | LACE | - | 82.61 | 65.16 | 69.51 |
| | Composed GLIDE | 0.30 | 86.10 | 81.61 | 70.44 |
| Non-uniform | Theoretical CoInD Eq. (10) | 0.17 | 96.88 | **93.75** | 72.66 |
| | CoInD ($\lambda = 1.0$) | **0.15** | **99.95** | 92.41 | **84.98** |
| | LACE | - | 10.85 | 9.03 | 28.24 |
| | Composed GLIDE | 2.75 | 7.40 | 5.09 | 33.86 |
| Partial | Theoretical CoInD Eq. (10) | **1.11** | 23.44 | **64.84** | **53.12** |
| | CoInD ($\lambda = 1.0$) | 1.17 | **52.38** | 53.28 | 52.59 |

Table 3: Results on Colored MNIST to directly minimize the upper bound ($K_1 = 1, K_2 = 0.1$)

| Support | Method | JSD ↓ | ∧ Composition | | ¬ Composition | |
|---|---|---|---|---|---|---|
| | | | $R^2$ ↑ | CS ↑ | $R^2$↑ | CS↑ |
| | LACE | - | 0.97 | 91.19 | 0.85 | 50.00 |
| | Composed GLIDE | 0.302 | 0.94 | 83.75 | 0.91 | 48.43 |
| Uniform | Theoretical CoInD Eq. (10) | 0.270 | 0.98 | 92.19 | 0.92 | **64.06** |
| | CoInD ($\lambda = 1.0$) | **0.215** | 0.98 | **95.31** | 0.92 | 55.46 |
| | LACE | - | 0.88 | 62.07 | 0.70 | 30.10 |
| | Composed GLIDE | 0.503 | 0.86 | 51.56 | 0.61 | 34.63 |
| Partial | Theoretical CoInD Eq. (10) | 0.450 | 0.93 | 78.13 | 0.88 | 51.56 |
| | CoInD ($\lambda = 1.0$) | **0.287** | **0.97** | **91.10** | **0.92** | **53.90** |

Table 4: Results on Shapes3D with the objective of directly minimizing the upper bound Eq. (10) ($K_1 = 1$, $K_2 = 0.1$)

### C.3 Choice of Hyperparameter $\lambda$

**Effect of $\lambda$ on the Learned Conditional Independence.** CoInD enforces conditional independence between the marginals of the attributes learned by the model by minimizing $\mathcal{L}_{\text{CI}}$ defined in Eq. (32). Here, we investigate the effect of $\mathcal{L}_{\text{CI}}$ on the effectiveness of logical compositionality by varying its strength through $\lambda$ in Eq. (11). Figure 8 plots JSD and CS (∧) as functions of $\lambda$ for models trained on the Colored MNIST dataset under the diagonal partial support setting. When $\lambda = 0$, training relies solely on the score matching loss, resulting in higher conditional dependence between $C_i \mid \boldsymbol{X}$. As $\lambda$ increases, CS improves since ensuring conditional independence between the marginals also encourages more accurate learning of the true marginals. However, when $\lambda$ takes large values, the model learns truly independent conditional distribution $C \mid \boldsymbol{X}$ but effectively ignores the input compositions and generates samples based solely on the prior distribution $p_{\boldsymbol{\theta}}(\boldsymbol{X})$. As a result, CS drops.

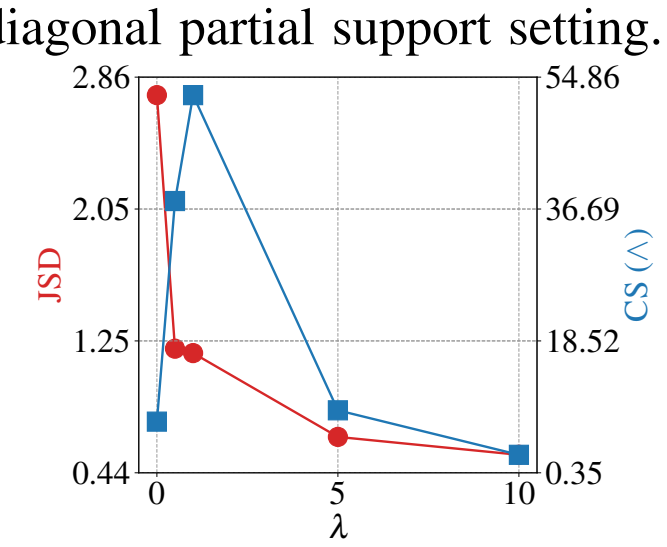

Figure 8: Effect of $\lambda$ on logical compositionality under diagonal partial support on the Colored MNIST dataset.

The value for the hyperparameter $\lambda$ is chosen such that the gradients from the score-matching objective $L_{\text{score}}$ and the conditional independence objective $L_{\text{CI}}$ are balanced in magnitude. One way to choose $\lambda$ is by training a vanilla diffusion model and setting $\lambda = \frac{L_{score}}{L_{CI}}$. We used two values for $\lambda$ in our experiments and noticed that they gave similar results, indicating that the approach was stable for various values of $\lambda$.

## D Experiment Details

In this section, we outline the high-level design choices of our approach. We provide full implementation details in our publicly available code and checkpoints at https://github.com/sachit3022/compositional-generation/.

### D.1 CoInD Algorithm

To compute pairwise independence in a scalable fashion, we randomly select two attributes, $i$ and $j$, for a sample in the batch and enforce independence between them. As the score in Eq. (15) is given by $\frac{\epsilon_\theta(\mathbf{x}_t,t)}{\sqrt{1-\bar{\alpha}_t}}$. The final equation for enforcing $\mathcal{L}_{\text{CI}}$ will be:

$$L_{CI} = \frac{1}{1-\bar{\alpha}_t} \left\| \boldsymbol{\epsilon}_\theta(\mathbf{x}_t, t, \mathbf{c}^i) + \boldsymbol{\epsilon}_\theta(\mathbf{x}_t, t, \mathbf{c}^j) - \boldsymbol{\epsilon}_\theta(\mathbf{x}_t, t, \mathbf{c}^{i,j}) - \boldsymbol{\epsilon}_\theta(\mathbf{x}_t, t, \mathbf{c}^{\varnothing}) \right\|_2^2$$

We follow Ho et al. (2020) to weight the term by $1 - \bar{\alpha}_t$. This results in an algorithm for CoInD, requiring only a few modifications of lines from (Ho & Salimans, 2022), highlighted below.

**Algorithm 1** CoInD Training

**repeat**
  $(\mathbf{c}, \mathbf{x}_0) \sim p_{\text{train}}(\mathbf{c}, x)$
  $c_k \leftarrow \varnothing$ with probability $p_{uncond}$ ▷ Set element of index,$k$ i.e, $c_k$ to $\varnothing$ with $p_{uncond} \forall k \in [0, N]$ probability
  $i \sim \text{Uniform}(\{0, \dots, N\}), j \sim \text{Uniform}(\{0, \dots, N\} \setminus \{i\})$ ▷ Select two random attribute indices
  $t \sim \text{Uniform}(\{1, \dots, T\})$
  $\boldsymbol{\epsilon} \sim \mathcal{N}(\mathbf{0}, \mathbf{I})$
  $x_t = \sqrt{\bar{\alpha}_t}\mathbf{x}_0 + \sqrt{1 - \bar{\alpha}_t}\boldsymbol{\epsilon}$
  $\mathbf{c}^i, \mathbf{c}^j, \mathbf{c}^{i,j} \leftarrow \mathbf{c}$
  $\mathbf{c}^i \leftarrow \{c_k = \varnothing \mid k \neq i\}, \mathbf{c}^j \leftarrow \{c_k = \varnothing \mid k \neq j\}, \mathbf{c}^{i,j} \leftarrow \{c_k = \varnothing \mid k \notin \{i, j\}\}, \mathbf{c}^{\varnothing} \leftarrow \varnothing$
  $L_{CI} = ||\boldsymbol{\epsilon}_\theta(\mathbf{x}_t, t, \mathbf{c}^i) + \boldsymbol{\epsilon}_\theta(\mathbf{x}_t, t, \mathbf{c}^j) - \boldsymbol{\epsilon}_\theta(\mathbf{x}_t, t, \mathbf{c}^{i,j}) - \boldsymbol{\epsilon}_\theta(\mathbf{x}_t, t, \mathbf{c}^{\varnothing})||_2^2$
  Take gradient descent step one
    $\nabla_\theta[\|\boldsymbol{\epsilon} - \boldsymbol{\epsilon}_\theta(\mathbf{x}_t, t, \mathbf{c})\|^2 \; + \lambda L_{CI} \; ]$
**until** converged

**Practical Implementation** In our experiments, we have used $p_{\text{uncond}} = 0.2$ and for Shapes3D instead of enforcing $C_i \perp\!\!\!\perp C_j \mid X$, for all $i, j$ enforcing $C_i \perp\!\!\!\perp C_{-i} \mid X$ for all $i$ have led to slightly better results.

### D.2 Details of Logical Compositionality Task

We designed the following task to evaluate two primitive logical compositions. **(1)** AND Composition $\wedge$, **(2)** NOT Composition $\neg$

**AND Composition** To evaluate the $\wedge$ composition, we apply the $\wedge$ operation over all the attributes to generate a respective image. Consider an image from the Shapes3D dataset (see Figure Fig. 9). The image is generated by some function, $f$, with the input $c = [\; 6 \quad 8 \quad 4 \quad 6 \quad 2 \quad 11 \;]$. The following image can be queried using the logical expression $C_1 = 6 \wedge \dots \wedge C_6 = 11$. We follow Equation Eq. (1) to sample from the above logical composition. To reiterate, for the $\wedge$ composition task on Shapes3D, the sampling equation is given by $\nabla_{\boldsymbol{X}} p_{\boldsymbol{\theta}}(\boldsymbol{X} \mid C_1 = 6 \wedge \dots \wedge C_6 = 11)$:

$$\nabla_{\mathbf{X}} \log p_{\boldsymbol{\theta}}(\boldsymbol{X}) + \sum_i \left[\nabla_{\mathbf{X}} \log p_{\boldsymbol{\theta}}(\boldsymbol{X} \mid C_i) - \nabla_{\mathbf{X}} \log p_{\boldsymbol{\theta}}(\boldsymbol{X})\right] \tag{33}$$

Similarly, to evaluate the AND composition for the Colored MNIST dataset, we perform the $\wedge$ operation over digit $C_1$ and color $C_2$.

**NOT Composition** To evaluate the $\neg$ compositions, the image is queried as an AND on all the attributes except the object attribute, which is queried by its negation. For example, consider the same image from Figure Fig. 9, where the object sphere ($C_5 = 2$) can be expressed as $C_5 = \neg[0 \vee 1 \vee 3]$, because the object class can only take four possible values. Therefore, the same image can be described as $C_1 = 6 \wedge \dots \wedge C_5 = \neg[0 \vee 1 \vee 3] \dots \wedge C_6 = 11$. The only possible generation that meets these criteria is the image (Fig. 9) displayed as expected.

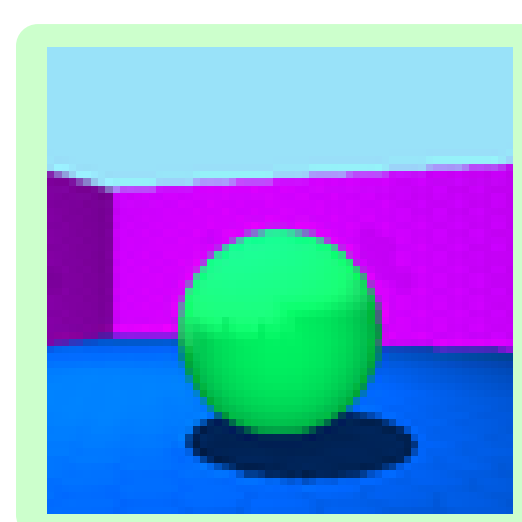

Figure 9: Image from Shapes3d with attributes $c = [6, 8, 4, 6, 2, 11]$

The sampling equation for a test image with attributes $C_1, C_2, C_3, C_4, C_5, C_6$ can be written as $C_1 = 6 \wedge C_2 = 8 \wedge C_3 = 4 \wedge C_4 = 6 \wedge C_5 = \neg[0 \vee 1 \vee 3] \wedge C_6 = 11$. Following Eq. (2), the sampling equation is written as follows:

$$\begin{aligned} &\nabla_{\mathbf{X}} \log p_{\boldsymbol{\theta}}(\boldsymbol{X}|C_1 = 6) + \nabla_{\mathbf{X}} \log p_{\boldsymbol{\theta}}(\boldsymbol{X}|C_2 = 8) + \nabla_{\mathbf{X}} \log p_{\boldsymbol{\theta}}(\boldsymbol{X}|C_3 = 4) \\ &\quad + \nabla_{\mathbf{X}} \log p_{\boldsymbol{\theta}}(\boldsymbol{X}|C_4 = 6) + \nabla_{\mathbf{X}} \log p_{\boldsymbol{\theta}}(\boldsymbol{X}|C_6 = 11) - \nabla_{\mathbf{X}} \log p_{\boldsymbol{\theta}}(\boldsymbol{X}|C_5 = 0) \\ &\quad\quad - \nabla_{\mathbf{X}} \log p_{\boldsymbol{\theta}}(\boldsymbol{X}|C_5 = 1) - \nabla_{\mathbf{X}} \log p_{\boldsymbol{\theta}}(\boldsymbol{X}|C_5 = 3) - \nabla_{\mathbf{X}} \log p_{\boldsymbol{\theta}}(\boldsymbol{X}) \end{aligned}$$

Similarly, for Colored MNIST, we perform two kinds of negation operations: one on digit and another on color. In Section § 2, we have shown negation on color $4 \wedge \neg[\text{Green} \vee \text{Pink}]$, along with

its sampling equation. A similar logic can be followed for negation on color; an example of negation on digit is $\neg[3 \vee 4] \wedge$ Pink.

For $\wedge$ and $\neg$, evaluations are strictly restricted to unseen compositions under orthogonal partial support for Shapes3D and under diagonal partial support for Colored MNIST. This approach allows us to explore how effectively the model handles logical operations through unseen image generation. Additionally, we evaluate compositions observed during training with less frequency under non-uniform support.

### D.3 Training details, Architecture, and Sampling

**Training Composed GLIDE & CoInD** We train the diffusion model using the DDPM noise scheduler. The model architecture and hyperparameters used for all experiments are detailed in Tab. 5.

**Training LACE** The LACE method involves training multiple energy-based models for each attribute and sampling according to logical compositional equations. However, we use score-based models instead. We follow the architecture outlined in Tab. 5 for each attribute to train multiple score-based models. For Colored MNIST, which has two attributes, we create two models—one for each attribute—using the same architecture as other methods, effectively doubling the model size. Similarly, for Shapes3D with six attributes, we develop six models. We reduce the Block Out Channels for each attribute model to fit these into memory while keeping all other hyperparameters consistent. Since we train a single model per attribute, we do not match the joint distribution, preventing us from evaluating it and measuring the JSD.

**Sampling** To generate samples for a given logical composition, we sample from equations from App. D.2 using DDIM (Song et al., 2021a) with 100 steps.

| Hyperparameter | Colored MNIST | | Shapes3D | |
|---|---|---|---|---|
| | CoInD & Composed GLIDE | LACE | CoInD & Composed GLIDE | LACE |
| Optimizer | AdamW | AdamW | AdamW | AdamW |
| Learning Rate | $2.0 \times 10^{-4}$ | $2.0 \times 10^{-4}$ | $2.0 \times 10^{-4}$ | $2.0 \times 10^{-4}$ |
| Num Training Steps | 50000 | 100000 | 100000 | 100000 |
| Train Noise Scheduler | DDPM | DDPM | DDPM | DDPM |
| Train Noise Schedule | Linear | Linear | Linear | Linear |
| Train Noise Steps | 1000 | 1000 | 1000 | 1000 |
| Sampling Noise Schedule | DDIM | DDIM | DDIM | DDIM |
| Sampling Steps | 150 | 150 | 100 | 100 |
| Model | U-Net | U-Net | U-Net | U-Net |
| Layers per block | 2 | 2 | 2 | 2 |
| Beta Schedule | Linear | Linear | Linear | Linear |
| Sample Size | 28x3x3 | 28x3x3 | 64x3x3 | 64x3x3 |
| Block Out Channels | [56,112,168] | [56,112,168] | [56,112,168,224] | **[56,112,168]** |
| Dropout Rate | 0.1 | 0.1 | 0.1 | 0.1 |
| Attention Head Dimension | 8 | 8 | 8 | 8 |
| Norm Num Groups | 8 | 8 | 8 | 8 |
| Number of Parameters | $8.2M$ | $8.2M \times 2$ | $17.2M$ | $8.2M \times 6$ |

Table 5: Hyperparameters for Colored MNIST and Shapes3D used by CoInD, Composed GLIDE, and LACE

**CelebA** To generate CelebA images, we scale the image size to $128 \times 128$. We use the latent encoder of Stable Diffusion 3 (SD3) to encode the images to a latent space and perform diffusion in the latent space. The architecture is similar to the Colored MNIST and Shapes3D, except that Block out Channels are scaled as [224, 448, 672, 896]. We use a learning rate of $1.0 \times 10^{-4}$ and train the model for 500,000 steps on one A6000 GPU.

**FID Measure** To evaluate both the generation quality and how well the generated samples align with the natural distribution of 'smiling male celebrities', we use the FID metric (Seitzer, 2020). Notably, we calculate the FID score specifically on the subset of 'smiling male celebrities,' as our primary objective is to assess the model's ability to generate these unseen compositions. We generate 10000 samples to evaluate FID.

**T2I: Finetuning SDv1.5** We finetune SDv1.5 with the data constructed from CelebA, where the labels are converted to text. For example, a label of (male=1, smiling=1) is converted to a “photo of a smiling male celebrity."

### D.4 Analytical Forms of Support Settings

Below are the analytical expressions for the densities under the various support settings that we considered in the paper. Let $n_i$ be the number of categories for the attribute $C_i$. For non-uniform and diagonal partial support settings, we assume that $n_i = n_j = n, \forall i, j, i \neq j$.

- **Uniform setting:** $p(C_i = c_1) = \frac{1}{n_i}$ and $p(C_i = c_1, C_j = c_2) = p(C_i = c_1)p(C_j = c_2) = \frac{1}{n_i n_j}$.
- **Orthogonal support setting:** $p(C_i = c_1, C_j = c_2) = \begin{cases} \frac{1}{n_i+n_j-1}, & c_1 = 0 \text{ or } c_2 = 0 \\ 0, & \text{otherwise} \end{cases}$
- **Non-uniform setting:** $p(C_i = c_1, C_j = c_2) = \begin{cases} a, & c_2 \leq c_1 \leq c_2 + 1 \\ b, & \text{otherwise} \end{cases}$. where $\frac{1}{n^2} \leq b \leq a \leq \frac{1}{2n-1}$
- **Diagonal partial support setting:** $p(C_i = c_1, C_j = c_2) = \begin{cases} \frac{1}{2n-1}, & c_2 \leq c_1 \leq c_2 + 1 \\ 0, & \text{otherwise} \end{cases}$.

### D.5 Datasets

**Colored MNIST Dataset** In Section § 1, we introduced the Colored MNIST dataset. Here, we will detail the dataset generation process. We selected 10 visually distinct colors [1], taking the value $C_2 \in [0, 9]$. The dataset is constructed by coloring the grayscale images from MNIST by converting them into three channels and applying one of the ten colors to non-zero grayscale values.

The training data is composed of three types of support:

- **Uniform Support**: A digit and a color are randomly selected to create an image.
- **Diagonal Partial Support**: A digit is selected, and during training, it is only assigned one of two colors, $C_2 \in \{d, d + 1\}$, except for 9, which only takes one color. This creates a dataset where compositions observed during training are along the diagonal of the $\mathcal{C}$ space, meaning each digit is seen only with its corresponding colors.
- **Non-uniform Support**: All compositions are observed, but combining a digit and its corresponding colors occurs with a higher probability (0.5). The remaining color space is distributed evenly among other colors, resulting in approximately a 0.25 probability for each corresponding color and a 0.0625 probability for each remaining color.

**Shapes3D** Full support for Shapes3D consists of all samples from the dataset. For orthogonal support, we use the composition split of Shapes3D as described by Schott et al.., whose code is publicly available [2].

**CelebA** CelebA consists of 40 attributes, from which we select the "smiling" and "male" attributes. We train generative models on all combinations of these attributes except (smiling=1, male=1), resulting in an orthogonal partial support.

### D.6 Conformity Score (CS)

In Section 2, we described the Conformity Score (CS) to quantify the accuracy of the generation per the prompt. To measure the CS, we train a single ResNet-18 (He et al., 2016) classifier with multiple classification heads, one corresponding to each attribute, and trained on the full support. This classifier estimates the attributes in the generated image, $\mathbf{x}$, and extracts these attributes as

[1] `https://mokole.com/palette.html`
[2] https://github.com/bethgelab/InDomainGeneralizationBenchmark

$\phi(\mathbf{x}) = [\hat{c_1}, \ldots, \hat{c_n}]$. These attributes are matched against the input prompt that generated the image to obtain accuracy.

To explain further, for example, if the prompt is to generate "$4 \wedge \neg$[Green $\vee$ Pink]", the generated sample will have a CS of 1 if $\hat{c_1} = 4$ and $\hat{c_2} \notin \{\text{Green}, \text{Pink}\}$. We average this across all the prompts in the test set, which determines the CS for a given task.

The effectiveness of the classifier in predicting the attributes is reported in Table 10.

| Feature | Attributes | Possible Values | Accuracy |
|---|---|---|---|
| $C_1$ | Digit | 0-9 | 98.93 |
| $C_2$ | color | 10 values | 100 |

(a) Colored MNIST Dataset

| Feature | Attributes | Possible Values | Accuracy |
|---|---|---|---|
| $C_1$ | Gender | {0,1} | 98.2 |
| $C_2$ | Smile | {0,1} | 92.1 |

(b) CelebA Dataset

| Feature | Attributes | Possible Values | Accuracy |
|---|---|---|---|
| $C_1$ | floor hue | 10 values in [0, 1] | 100 |
| $C_2$ | wall hue | 10 values in [0, 1] | 100 |
| $C_3$ | object hue | 10 values in [0, 1] | 100 |
| $C_4$ | scale | 8 values in [0, 1] | 100 |
| $C_5$ | shape | 4 values in [0-3] | 100 |
| $C_6$ | orientation | 15 values in [-30, 30] | 100 |

(c) Shapes3D Dataset

Figure 10: Independent attribute, their possible values, and the classifier accuracy in estimating them for different datasets

### D.7 Computing JSD

We are interested in understanding the causal structure learned by diffusion models. Specifically, we aim to determine whether the learned model captures the conditional independence between attributes, allowing them to vary independently. This raises the question: *Do diffusion models learn the conditional independence between attributes?* The conditional independence is defined by:

$$p_{\boldsymbol{\theta}}(C_i, C_j \mid \boldsymbol{X}) = p_{\boldsymbol{\theta}}(C_i \mid \boldsymbol{X}) p_{\boldsymbol{\theta}}(C_j \mid \boldsymbol{X}) \tag{34}$$

We aim to measure the violation of this equality using the Jensen-Shannon divergence (JSD) to quantify the divergence between two probability distributions:

$$\text{JSD} = \mathbb{E}_{p_{\text{data}}} \left[ D_{\text{JS}} \left( p_{\boldsymbol{\theta}}(C \mid \boldsymbol{X}) \,||\, p_{\boldsymbol{\theta}}(C_i \mid \boldsymbol{X}) p_{\boldsymbol{\theta}}(C_j \mid \boldsymbol{X}) \right) \right] \tag{35}$$

The joint distribution, $p_{\boldsymbol{\theta}}(C_i, C_j \mid \boldsymbol{X})$, and the marginal distributions, $p_{\boldsymbol{\theta}}(C_i \mid \boldsymbol{X})$ and $p_{\boldsymbol{\theta}}(C_j \mid \boldsymbol{X})$, are evaluated at all possible values that $C_i$ and $C_j$ can take to obtain the probability mass function (pmf). The probability for each value is calculated using Equation Eq. (18) for the joint distribution and Equation Eq. (17) for the marginals.

**Practical Implementation** For the diffusion model with multiple attributes, the violation in conditional *mutual* independence should be calculated using all subset distributions. However, we focus on pairwise independence. We further approximate this in our experiments by computing JSD between the first two attributes, $C_1$ and $C_2$. We have observed that computing JSD between any attribute pair does not change our examples' conclusion.

### D.8 Measuring Diversity in Attributes

To achieve explicit control over certain attributes during the generation process, these attributes must vary independently. Therefore, an ideal generative model must be able to produce samples where all except the controlled attributes take diverse values. This diversity can be measured by the entropy of the uncontrolled attributes in the generated samples, where higher entropy suggests greater diversity. Therefore, the accurate generation of controlled and diverse uncontrolled attributes for the given the underlying data distribution has uniform attributes indicates that the model has successfully learned when the underlying attribute distribution is uniform. In contrast, for non-uniform

distributions—such as the Gaussian example discussed in App. G.1—a simple diversity argument no longer applies, and minimum KL divergence between the model and the true distribution becomes the appropriate measure. Under a uniform attribute assumption, however, the KL divergence essentially reduces to maximum entropy.

For example, consider the generation of colored MNIST digits. In this case, controllability means that the model has learned that digit and color attributes are independent. When prompted to generate a specific digit (controlled attribute), the model should generate this digit in all possible colors (uncontrolled attribute) with equal likelihood, implying maximum entropy for the color attribute and diverse generation. We measure this entropy by generating samples $x^i \sim p_\theta(X \mid c_1 = 4)$ and passing them through a near-perfect classifier to obtain the color predictions $p(\hat{C}_2) = p(\phi_2(x^i))$. The diversity is then quantified as: $H = \mathbb{E}_{\hat{c_2} \sim p(\hat{C_2})} [\log_2 p(\hat{c_2})]$

Ensuring diversity through explicit control has applications in bias detection and mitigation in generative models. For example, a biased model may generate images of predominantly male doctors when asked to generate images of "doctors". Ensuring diversity in uncontrolled attributes like gender or race can limit such biases.

# E CoInD for Face Image Generation

In § 5, we demonstrated that CoInD outperforms baseline methods on the unseen logical compositionality task using synthetic datasets. In App. E.1, we showcase the success of CoInD in generating face images from the CelebA dataset (Liu et al., 2015), where CoInD demonstrates superior control over attributes compared to the baseline. CoInD also allows us to adjust the strength of various attributes and thus provides more fine-grained control over the compositional attributes, as shown in App. E.2. Finally, in App. E.3, we extend CoInD to text-to-image (T2I) models widely used in practice to generate face images by providing the desired attributes as logical expressions of text prompts.

**Problem Setup** We choose the CelebA dataset to evaluate CoInD's ability to generate real-world images. We choose the binary attributes "smiling" and "gender" as the attributes we wish to control. During training, all combinations of these attributes except gender = "male" and smiling = "true" are observed, similar to the orthogonal support shown in Fig. 3. During inference, the model is tasked to generate images with the attribute combination gender = "male" and smiling = "true", which was not observed during training.

**Metrics** Similar to the experiments on the synthetic image datasets in § 5, we assess the accuracy of the generation w.r.t. the input desired attribute combination CS (conformity score). We also measure the violation of the learned conditional independence using JSD. In addition to CS, we compute FID (Fréchet inception distance) between the generated images and the real samples in the CelebA dataset where gender = "male" and smiling = "true". A lower FID implies that the distribution of generated samples is closer to the real distribution of the images in the validation dataset.

## E.1 CoInD can successfully generate real-world face images

Tab. 2 shows the quantitative results of CoInD and Composed GLIDE trained from scratch in the tasks of joint sampling and $\wedge$ composition. Similar to our observations from previous experiments, CoInD achieves better CS in both tasks by learning accurate marginals as demonstrated by lower JSD. When sampled from the joint likelihood, CoInD achieves a nearly $4\times$ improvement in CS over the baseline.

## E.2 CoInD provides fine-grained control over attributes

So far, we studied the capabilities of CoInD to dictate the presence and absence of attributes in the task of controllable image generation. However, there are applications where we desire fine-grained control over the attributes. Specifically, we may want to control the *amount of each attribute* in the generated sample. We can mathematically formulate this task by revisiting the formulation of logical expressions of attributes in terms of the score functions of marginal likelihood. As an example, the

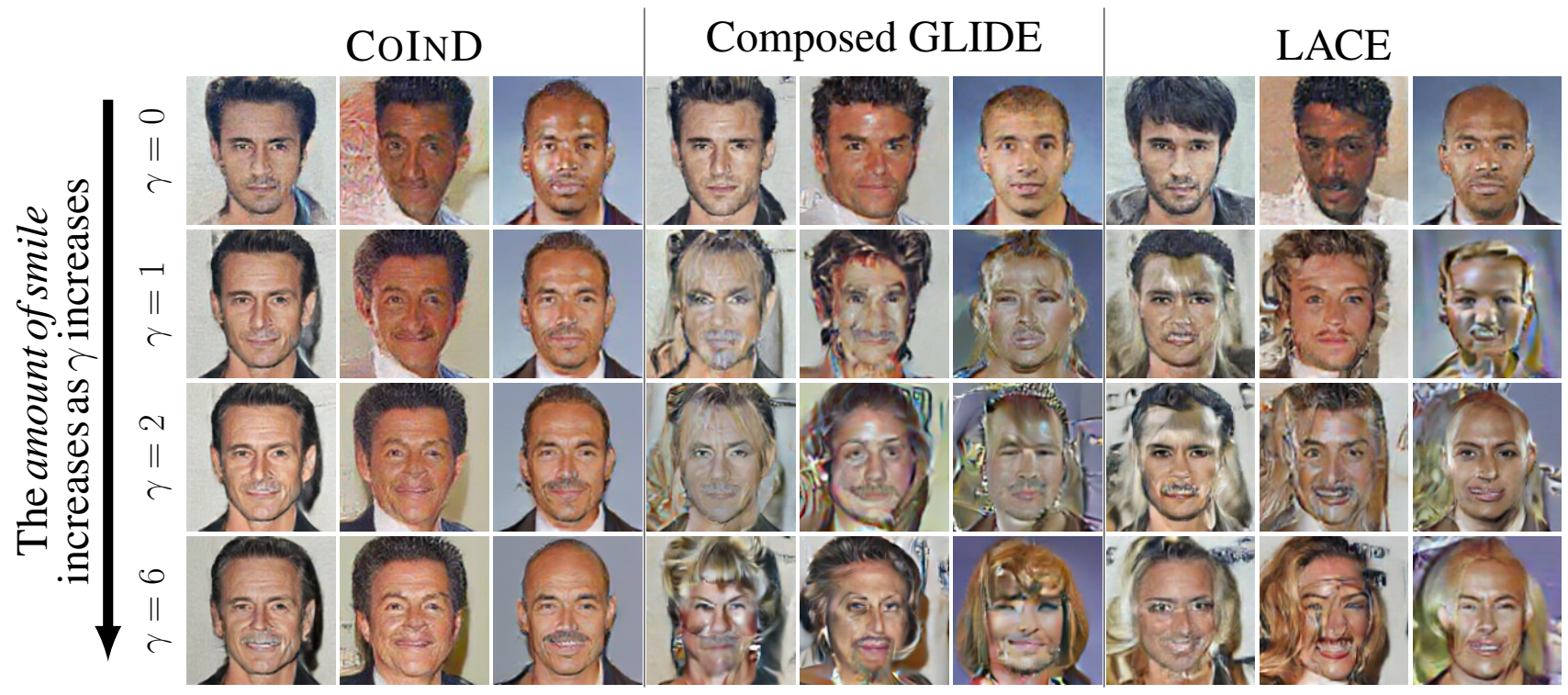

Figure 11: By adjusting $\gamma$, CoInD allows us to the vary the *amount of "smile"* in the generated images. However, Composed GLIDE associates the smile attribute with the gender attribute due to their association in the training data. Hence, the images generated by Composed GLIDE contain gender-specific attributes such as long hair and earrings.

$\wedge$ operation can be written as,

$$\nabla_{\boldsymbol{X}} \log p_{\boldsymbol{\theta}}(\boldsymbol{X} \mid C_1 \wedge C_2) = \nabla_{\boldsymbol{X}} \log p_{\boldsymbol{\theta}}(\boldsymbol{X} \mid C_1) + \nabla_{\boldsymbol{X}} \log p_{\boldsymbol{\theta}}(\boldsymbol{X} \mid C_2) - \nabla_{\boldsymbol{X}} \log p_{\boldsymbol{\theta}}(\boldsymbol{X})$$

Here, to adjust the amount of attribute added to the generated sample, we can weigh the score functions using some scalar $\gamma$, as follows,

$$\nabla_{\boldsymbol{X}} \log p_{\boldsymbol{\theta}}(\boldsymbol{X} \mid C_1) + \gamma \nabla_{\boldsymbol{X}} \log p_{\boldsymbol{\theta}}(\boldsymbol{X} \mid C_2) - \gamma \nabla_{\boldsymbol{X}} \log p_{\boldsymbol{\theta}}(\boldsymbol{X}) \tag{36}$$

where $\gamma$ controls for the amount of $C_2$ attribute.

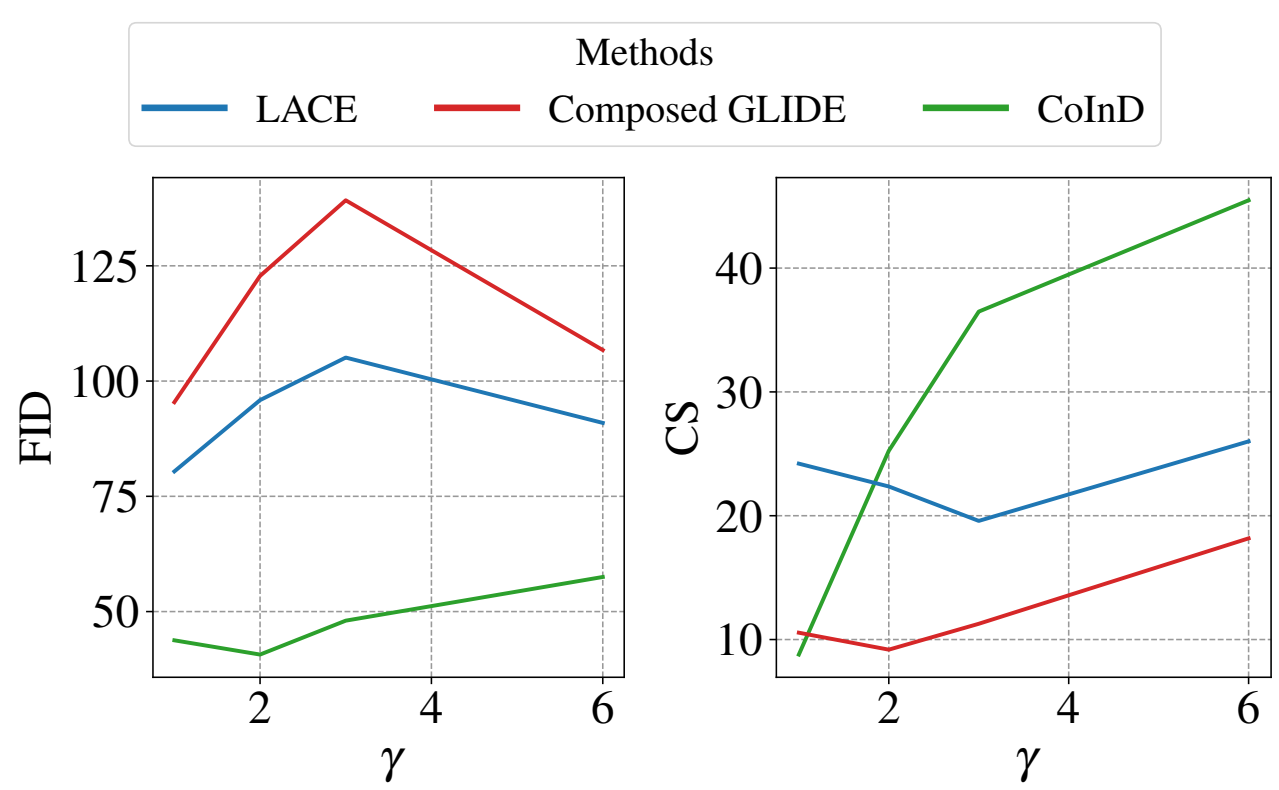

(a) Variation of FID with $\gamma$ (b) Variation of CS with $\gamma$

Figure 12: **Effect of $\gamma$ on FID and CS:** Varying the amount of smile in a generated image through $\gamma$ does not affect the FID of CoInD. However, the smiles in the generated images become more apparent, leading to easier detection by the smile classifier and improved CS.

Fig. 11 shows the effect of increasing $\gamma$ to adjust the amount of smiling in the generated image. Ideally, we expect increasing $\gamma$ to increase the amount of smiling without affecting the gender attribute. When $\gamma = 0$ (top row), both CoInD and Composed GLIDE generate images of men who are not smiling. As $\gamma$ increases, we notice that the samples generated by CoInD show an increase in the amount of smiling, going from a short smile to a wider smile to one where teeth are visible. Note that the training dataset did not include any images of smiling men or fine-grained annotations for the amount of smiling in each image. This conclusion is strengthened by Fig. 12b that shows an increase in CS when $\gamma$ increases. CS increases when it is easier for the smile classifier to detect the smile. CoInD provides this fine-grained control over the smiling attribute without any effect on the realism of the images, as shown by the minimal changes in FID in Fig. 12a.

In contrast, the images generated by Composed GLIDE show an increase in the amount of smiling while adding gender-specific attributes such as long hair and makeup. We conclude that, by strictly enforcing a conditional independence loss between the attributes, CoInD provides fine-grained control over the attributes, allowing us to adjust the intensity of the attribute in the image without additional training. As shown in Tab. 2, CoInD outperforms the baselines for generating unseen compositions. Tuning $\gamma$ further improves the generation.

### E.3 Finetuning T2I models with CoInD improves logical compositionality

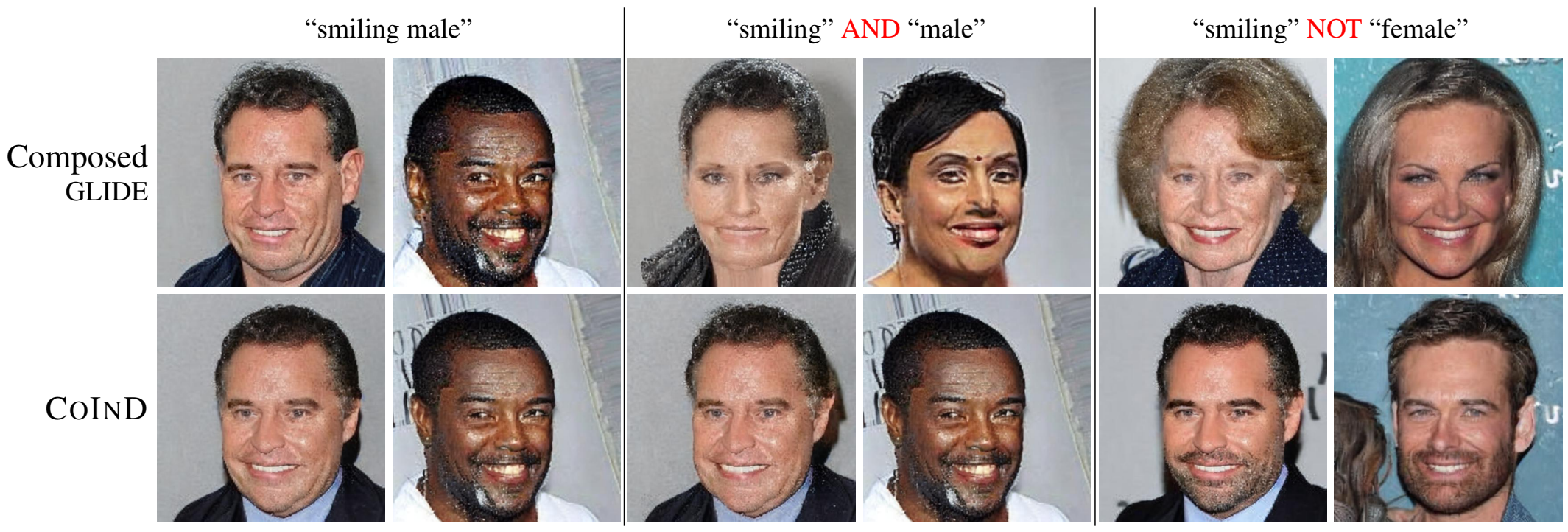

Figure 13: Samples generated after fine-tuning SDv1.5 on CelebA. The first row shows images generated by SDv1.5 fine-tuned on CelebA, while the second row shows images generated by SDv1.5 fine-tuned with CoInD. Columns indicate samples generated from the respective prompts indicated above.

We proposed CoInD to improve control over the attributes in an image through logical expressions of these attributes. Since larger pre-trained diffusion models such as Stable Diffusion (Rombach et al., 2022) have become more accessible, we seek to incorporate the benefits of CoInD in these models. This section shows that text-to-image (T2I) models can be fine-tuned to generate images using logical expressions of text prompts. Specifically, we use Stable Diffusion v1.5 (SDv1.5) to generate face images from the CelebA dataset where smiling and gender attributes can be controlled. We consider both joint and marginal sampling, similar to our case study in § 3. For joint sampling, we provide SDv1.5 with the prompt "Photo of a smiling male celebrity". In the marginal sampling, we provide the values for smiling and gender attributes using separate prompts – "Photo of a smiling celebrity" $\wedge$ "Photo of a male celebrity". Then, we sample from these marginal likelihoods resulting from these prompts following Eq. (1). To evaluate $\neg$ capabilities, we use the prompts "Photo of a smiling celebrity" $\neg$ "Photo of a female celebrity" and follow Eq. (2).

| Support | Method | JSD $\downarrow$ | Joint | | $\wedge$ Composition | | $\neg$ Composition | |
|---|---|---|---|---|---|---|---|---|
| | | | CS $\uparrow$ | FID $\downarrow$ | CS $\uparrow$ | FID $\downarrow$ | CS $\uparrow$ | FID $\downarrow$ |
| Orthogonal | Composed GLIDE | 0.57 | 56.57 | 58.31 | 14.19 | 73.53 | 11.02 | 115.95 |
| | CoInD ($\lambda = 1.0$) | **0.37** | **58.57** | **58.19** | **49.15** | **61.16** | **18.80** | **86.31** |

Table 6: **Results on SDv1.5 fine-tuning.** CoInD outperforms the baseline on all the metrics.

**Discussion**

1. In Tab. 6, CoInD improves performance across all metrics – achieving $3.46\times$ and $2\times$ improvement in CS over Composed GLIDE in $\wedge$ and $\neg$ composition tasks. The images generated by CoInD have better FID than those from the baseline.
2. Visual inspection of the generated samples for the same random seed provides insights into how Composed GLIDE and CoInD perceive the prompts. Images in columns 1, 3, and 5 of Fig. 13 were generated with the same random seed. Similarly, those in columns 2 and 4 share the random seed. We note the following observations:

- Both Composed GLIDE and CoInD generated images with the desired attributes when sampled from the joint likelihood using “photo of a smiling male celebrity”. The images generated by these models from the same random seed were also visually similar. This shows that both models can aptly set attributes in the generated images and have identical stochastic profiles, leading to unspecified attributes that assume similar values.
- When the attributes were passed as the $\wedge$ expression “smiling” $\wedge$ “male”, CoInD generated images that were visually similar to those with matching random seeds generated from joint sampling. This implies that CoInD learned accurate marginals that help it to correctly model the joint likelihood.
- When tasked with generating images for “smiling” $\wedge$ “male”, Composed GLIDE generated images of smiling persons with gender-specific attributes such as thinner eyebrows, commonly seen in photos of female celebrities. These gender-specific features increase when the task is to generate images of “smiling” $\neg$ “female”. In contrast, CoInD generates images of smiling celebrities while adding attributes such as a beard. Thus, we conclude that CoInD offers better control over the desired attributes without affecting correlated attributes.

# F Discussion on CoInD

## F.1 Connection to compositional generation from first principles

Compositional generation from first principles Wiedemer et al. (2024) have shown that restricting the function to a certain compositional form will perform better than a single large model. In this section, we show that, by enforcing conditional independence, we restrict the function to encourage compositionality.

Let $c_1, c_2, \ldots, c_n$ be independent components such that $c_1, c_2, \ldots, c_n \in \mathbb{R}$. Consider an injective function $f : \mathbb{R}^n \to \mathbb{R}^d$ defined by $f(c) = x$. If the components, $c$ are conditionally independent given $x$ the cumulative functions, $F$ must satisfy the following constraint:

$$F_{C_i, C_j, \ldots, C_n | X=x}(c_i, c_j, \ldots, c_n) = \prod_i F_{C_i|X=x}(c_i) \tag{37}$$

$F^{-1}_{C_i, C_j, \ldots, C_n | X=x}(x) = \inf\{c_i, c_j, \ldots, c_n \mid F(c_i, c_j, \ldots, c_n) \geq x\}$, where $F^{-1}_{c_i, c_j, \ldots, C_n | X=x}$ is a generalized inverse distribution function.

$$\begin{aligned} f(c_i, c_j, \ldots, c_n) &= (f \circ F^{-1}_{c_i, c_j, \ldots, C_n | X=x})(\prod_i F_{C_i|X=x}(c_i)) \\ &= (f \circ F^{-1}_{c_i, c_j, \ldots, C_n | X=x} \circ e)(\sum_i \log F_{C_i|X=x}(c_i)) \\ &= g(\sum_i \phi_i(c_i)) \end{aligned}$$

Therefore, we are restricting $f$ to take a certain functional form. However, it is difficult to show that the data generating process, $f$, meets the rank condition on the Jacobian for the sufficient support assumption Wiedemer et al. (2024), which is also the limitation discussed in their approach. Therefore, we cannot provide guarantees. However, this section provides a functional perspective of CoInD.

## F.2 2D Gaussian: Closed-form Analysis of CoInD

In this section, we derive closed-form expressions for the score functions underlying our method and demonstrate how CoInD leverages conditional independence constraints to generate the true data distribution.

**Data Generation Process** We consider data generated from two independent attributes, $C_1$ and $C_2$, which are binary variables taking values in $\{-1, +1\}$. The observed variable $\mathbf{X}$ is defined as:

$$\mathbf{X} = f(C_1) + f(C_2), \tag{38}$$

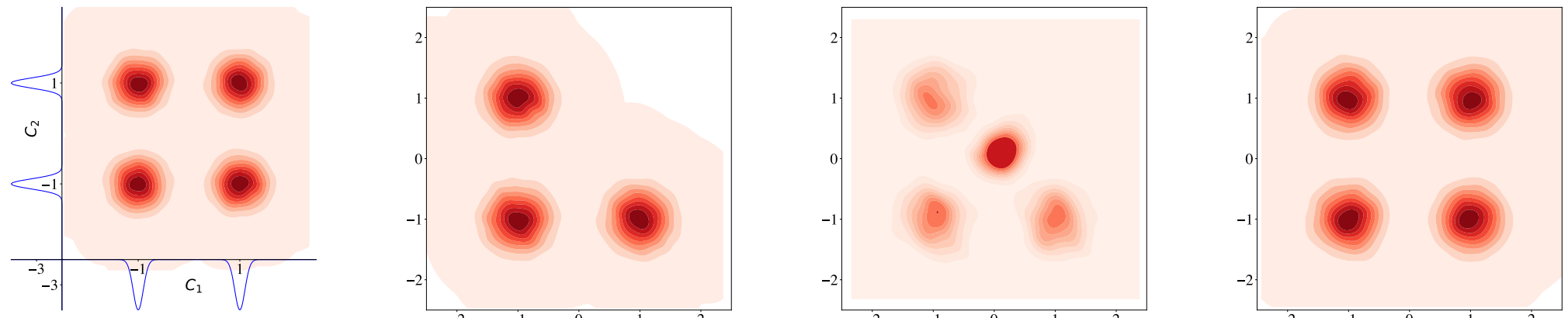

(a) True underlying data distribution (b) Training data Orthogonal Support (c) Conditional distribution learned by vanilla diffusion objective (d) Conditional distribution learned by CoInD

Figure 14: CoInD respects underlying independence conditions thereby generating true data distribution (d).

where

$$f(c) = c + \sigma\epsilon, \quad \epsilon \sim \mathcal{N}(0, I).$$

Thus, $f(C_1)$ produces a Gaussian mixture along the $x$-axis with means at $-1$ and $+1$, and similarly $f(C_2)$ produces a mixture along the $y$-axis with means at $-1$ and $+1$ (see blue plot on the axis of Figure 14a). The combination yields a two-dimensional Gaussian mixture (Figure 14a).

**Training Setup and Orthogonal Support** For training, we assume *orthogonal support* where only the following combinations of $(C_1, C_2)$ are observed: $\{(-1,-1), (-1,+1), (+1,-1)\}$. The model is then tasked with generating samples from the unseen composition $(+1,+1)$. Recall that our assumptions (see Section 2) are satisfied: $C_1$ and $C_2$ independently generate $\mathbf{X}$, and all possible values for each attribute are observed at least once during training.

**Score Function Decomposition** Let $s_{+1,+1}(x)$ denote the score corresponding to $p(x \mid C_1 = +1, C_2 = +1)$, and let $s_{1,\varnothing}(x)$ denote the marginal score $p(x \mid C_1 = +1)$ (with a similar definition for $s_{\varnothing,1}(x)$). Leveraging Eq. (1) $s_{+1,+1}(x)$ is decomposed as follows:

$$s_{+1,+1}(x) = s_{1,\varnothing}(x) + s_{\varnothing,1}(x) - s_{\varnothing,\varnothing}(x), \tag{39}$$

where $s_{\varnothing,\varnothing}(x)$ is the score of the training data and not full data.

For example, when training on the observed combination $(+1,-1)$, the score function of the $s_{1,\varnothing}, s_{\varnothing,1}$, is a Gaussian, and written in closed form as

$$\begin{aligned} s_{1,\varnothing}(x) &= \frac{\mu_{+1,-1} - x}{\sigma^2}, \\ s_{\varnothing,1}(x) &= \frac{\mu_{+1,-1} - x}{\sigma^2}. \end{aligned} \tag{40}$$

In contrast, the score of $s_{\varnothing,\varnothing}(x)$, is the mixture (over the three training components) given as:

$$s_{\varnothing,\varnothing}(x) = \frac{\sum_i \mathcal{N}(x; \mu_i, \sigma^2 I)\left(\frac{\mu_i - x}{\sigma^2}\right)}{\sum_i \mathcal{N}(x; \mu_i, \sigma^2 I)}. \tag{41}$$

However, when using Langevin dynamics for sampling (see Eq. (13)), the naive combination in Eq. (39) produces an incorrect conditional distribution (Figure 14c). Specifically, the generated distribution shows a spurious red blob between the $(+1,-1)$ and $(-1,+1)$ modes rather than a proper Gaussian centered at $(+1,+1)$. This shows that Diffusion models interpolate between the modes, rather than following underlying conditional independence and generalizing to unseen modes.

**Correcting with Conditional Independence Constraints** Instead of modeling $s_{1,\varnothing}(x)$ directly, CoInD learns the joint scores for the three observed combinations:

$$s_{-1,-1}(x), \quad s_{+1,-1}(x), \quad s_{-1,+1}(x).$$

These are then combined under the assumption of pairwise conditional independence to infer the score for the unseen composition:

$$\begin{aligned} s_{-1,-1}(x) &= s_{+1,\varnothing}(x) + s_{\varnothing,+1}(x) - s_{\varnothing,\varnothing}(x), \\ s_{+1,-1}(x) &= s_{+1,\varnothing}(x) + s_{\varnothing,-1}(x) - s_{\varnothing,\varnothing}(x), \\ s_{-1,+1}(x) &= s_{-1,\varnothing}(x) + s_{\varnothing,+1}(x) - s_{\varnothing,\varnothing}(x), \end{aligned} \tag{42}$$

which leads to the following expression for the unseen $(+1,+1)$ composition:

$$\begin{aligned} s_{+1,+1}(x) &= s_{+1,\varnothing}(x) + s_{\varnothing,+1}(x) - s_{\varnothing,\varnothing}(x) \\ &= s_{+1,-1}(x) + s_{-1,+1}(x) - s_{-1,-1}(x) \\ &= \frac{[\mu_{+1,-1} + \mu_{-1,+1} - \mu_{-1,-1}] - x}{\sigma^2}. \end{aligned} \tag{43}$$

The derivation above shows that CoInD effectively enforces conditional independence constraints to generate the unseen data distribution. This analysis underscores the necessity of incorporating conditional independence constraints into diffusion models to faithfully reproduce the target distribution, particularly when extrapolating to unseen compositions.

### F.3 Extension to Gaussian source flow models

Diffusion models can be viewed as a specific case of flow-based models where: (1) the source distribution is Gaussian, and (2) the forward process follows a predetermined noise schedule (Lipman et al., 2024). Can we reformulate CoInD in terms of velocity rather than score, thereby generalizing it to accommodate arbitrary source distributions and schedules? When the source distribution is gaussian, score and velocity are related by affine transformation as detailed in Tab. 1 of (Lipman et al., 2024).

$$s^t_\theta(x, C_1, C_2) = a_t x + b_t u^t_\theta(x, C_1, C_2) \tag{44}$$

replacing $s^t_\theta(\cdot)$ into Eq. (32)

$$\begin{aligned} \mathcal{L}_{\text{CI}} &= \mathbb{E}_{p(\boldsymbol{X},C),t\sim U[0,1]}\mathbb{E}_{j,k}\|s^t_{\boldsymbol{\theta}}(x, C_j, C_k) - s^t_{\boldsymbol{\theta}}(x, C_j) - s^t_{\boldsymbol{\theta}}(x, C_k) + s^t_{\boldsymbol{\theta}}(x)\|^2_2 \\ &= \mathbb{E}_{p(\boldsymbol{X},C),t\sim U[0,1]}\mathbb{E}_{j,k}\left[b^2_t\|u^t_{\boldsymbol{\theta}}(x, C_j, C_k) - s^t_{\boldsymbol{\theta}}(x, C_j) - u^t_{\boldsymbol{\theta}}(x, C_k) + u^t_{\boldsymbol{\theta}}(x)\|^2_2\right] \end{aligned}$$

However we can ignore $b^2_t$, weighting for the time step $t$.

$$\mathcal{L}_{\text{CI}} = \mathbb{E}_{p(\boldsymbol{X},C),t\sim U[0,1]}\mathbb{E}_{j,k}\left[\|u^t_{\boldsymbol{\theta}}(x, C_j, C_k) - u^t_{\boldsymbol{\theta}}(x, C_j) - u^t_{\boldsymbol{\theta}}(x, C_k) + u^t_{\boldsymbol{\theta}}(x)\|^2_2\right] \tag{45}$$

Therefore, if the source distribution is gaussian and for any arbitrary noise schedule, constraint in score translates directly to velocity constraint as given as Eq. (45).

### F.4 Compositional vs Monolithic models

Our findings echo the prior observations (Du & Kaelbling, 2024) that composite models consisting of separate diffusion models trained on individual factors (e.g., LACE) demonstrate better $\wedge$ compositionality under partial support than sampling from factorized distributions learned by monolithic models (e.g., Composed GLIDE). However, we found that monolithic models can be significantly improved by enforcing the conditional independence constraints necessary for enabling logical compositionality. For instance, CoInD achieved a $2.4\times$ better CS on Colored MNIST with diagonal partial support and a $1.4\times$ improvement on orthogonal partial support on Shapes3D compared to LACE.

### F.5 Limitations

This paper considered compositions of a closed set of attributes. As such, CoInD requires pre-defined attributes and access to data labeled with the corresponding attributes. Moreover, CoInD must be enforced during training, which requires retraining the model whenever the attribute space changes to include additional values. Instead, state-of-the-art generative models seek to operate without pre-defined attributes or labeled data and generate open-set compositions. Despite the seemingly restricted setting of our work, our findings provide valuable insights into a critical limitation of current generative models, namely their failure to generalize for unseen compositions, by identifying the source of this limitation and proposing an effective solution to mitigate it.

# G Additional Results and Discussion on CoInD

## G.1 Learning under non-uniform $p(C_i)$

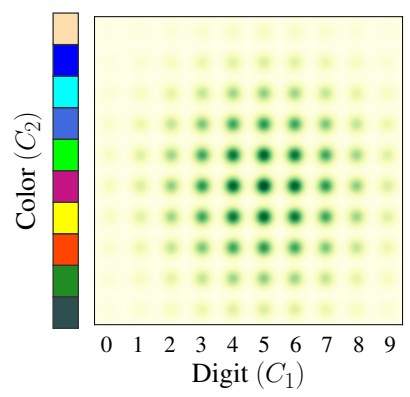

(a) Gaussian support

| Method | JSD ↓ | ∧ (CS) ↑ | ¬ Color (CS)↑ | ¬ Digit (CS) ↑ |
|---|---|---|---|---|
| LACE | - | 89.22 | 58.59 | 57.81 |
| Composed GLIDE | 0.27 | 91.74 | 88.91 | 78.39 |
| CoInD ($\lambda = 1.0$) | **0.16** | **99.61** | **98.51** | **83.03** |

(b) Quantitative results for Gaussian support

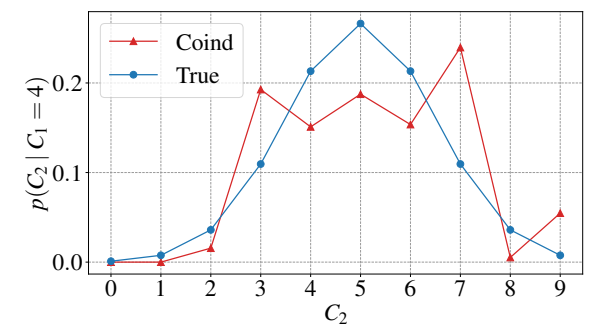

(c) $p(C_2 \mid C_1 = 4)$ vs $p_{\boldsymbol{\theta}}(C_2 \mid C_1 = 4)$

Figure 15: **Results on Gaussian support:** When the independent attributes have non-uniform categorical distributions, the joint distribution of attribute combinations is not uniform. Even in this case, CoInD learns $p_{\boldsymbol{\theta}}(C_i \mid C_j)$ accurately.

In our experiments, we considered the uniform support setting as an example where the attribute variables are independent of each other in the training data, i.e., $C_1 \perp\!\!\!\perp C_2 \mid \boldsymbol{X}$ during training. However, uniform support is not the only scenario that can arise from independent attribute variables. In this section, we show that CoInD can learn accurate marginals irrespective of the distribution of $C_i$.

We designed an experiment using the Colored MNIST images where the attributes $C_1$ and $C_2$ assume values from a non-uniform categorical distribution that resembles a discrete Gaussian distribution. The resulting joint distribution of the attributes, which we refer to as *Gaussian support*, is illustrated in Fig. 15a. We trained CoInD and baselines on this dataset and evaluated on $\wedge$ and $\neg$ compositionality tasks. Apart from comparing the CS of baselines and CoInD on these compositionality tasks, we also evaluate if CoInD accurately learns $p(C_i)$ by comparing the learned $p_{\boldsymbol{\theta}}(C_i \mid C_j)$ against the true $p(C_i \mid C_j)$. Intuitively, this verifies if CoInD generates images with uncontrolled attributes matching their distribution in the training dataset.

Fig. 15b quantitatively compares CoInD against Composed GLIDE on CS in both $\wedge$ and $\neg$ compositionality tasks. Like our previous experiments, CoInD outperforms Composed GLIDE w.r.t. CS in all tasks. In Fig. 15c, we verify if CoInD has learned $p_{\boldsymbol{\theta}}(C_2 \mid C_1)$ accurately by comparing it against the true distribution $p(C_2 \mid C_1)$. $p_{\boldsymbol{\theta}}(C_2 \mid C_1 = c^*) = p_{\boldsymbol{\phi}}(C_2 \mid X)p_\theta(X \mid C_1 = c^*)$ is obtained as the histogram density of the attributes that appear in the generated images when $C_1 = c^*$. We observe that the learned distribution $p_{\boldsymbol{\theta}}(C_2 \mid C_1 = 4)$ is close to the true distribution, forming a bell shape.

## G.2 Failure examples of CoInD

Here, we examine some samples generated by CoInD where it failed to include the desired attributes. We show these failure cases from each dataset, i.e., Colored MNIST, Shapes3d, and CelebA datasets. Samples from Colored MNIST and Shapes3d datasets are taken from the partial support setting, while the ones from the CelebA dataset are taken from the orthogonal support setting. Fig. 16a shows some failure samples from the Colored MNIST dataset. The images in the first row contain digits with colors leaking from the nearby seen attribute combination. Those in the second row correspond to $\neg$ approximation and have wrong attributes due to the approximation in the probabilistic formulation in Eq. (2). Some images, like those in the third row, are unrealistic, although they may contain the desired attributes. We observe similar failures in Shapes3d samples shown in Fig. 16b where the CoInD deviates from the desired compositions (column 1). Some failed samples from the CelebA dataset are shown in Fig. 16c. The samples correspond to the task of "smile" $\wedge$ "male". In the top image, it is hard to distinguish if the subject is smiling or laughing. In some samples, we observed only a weak or soft smile. This could be because a smile is difficult to control due to its limited spatial presence in an image.

## G.3 Conformity score for each attribute combination

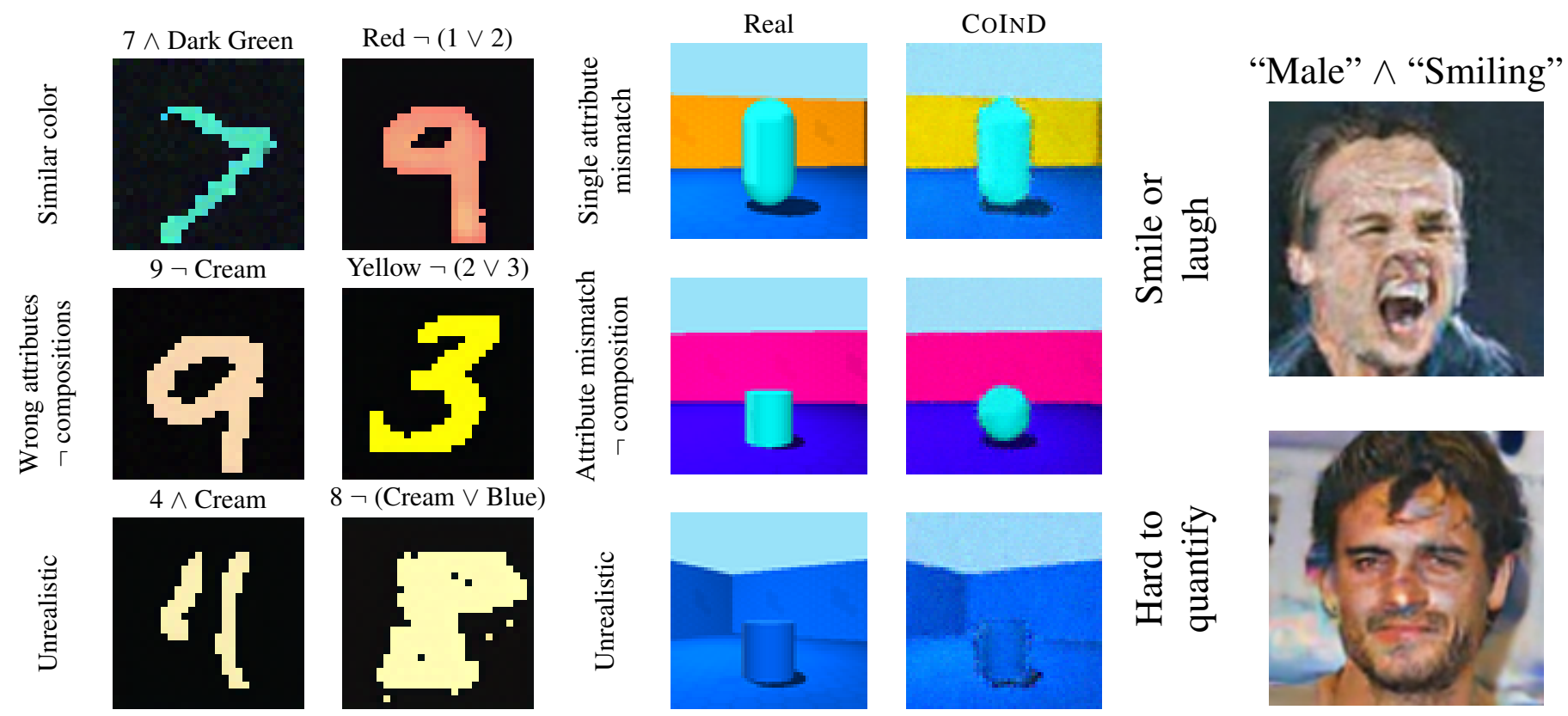

(a) Failure samples from Colored MNIST dataset (b) Failure samples from Shapes3d dataset (c) Failure samples from CelebA dataset

Figure 16: Some samples generated by CoInD where it could not enforce the desired attributes.

In all our experiments, we report CS as the primary metric to evaluate if the generative model produced images with accurate attributes. However, CS is the average accuracy across all unseen attribute combinations. Not all attribute combinations may be generated with equal accuracy.

For instance, Fig. 17 shows the CS for each attribute combination in the $\wedge$ compositionality task in Colored MNIST image generation with partial support setting (row 10 in Fig. 4a). As a reminder, CoInD achieved 52.38% CS on unseen attribute combinations in this task.

We can see that CoInD can successfully generate all seen attribute combinations that appear on the diagonal. Some unseen attribute combinations achieve $> 90\%$ CS, while others have nearly $0\%$ CS. We do not observe the model struggling to generate images with any specific attribute or digit, although some colors have a generally lower CS than others. For example, colors 2 and 3 have zero CS with more digits than others. On the other hand, colors 4, 5, and 6 have high CS with all digits. We hypothesize that this disparity in CS could depend on the nature of attributes and the similarity between the values they can take.

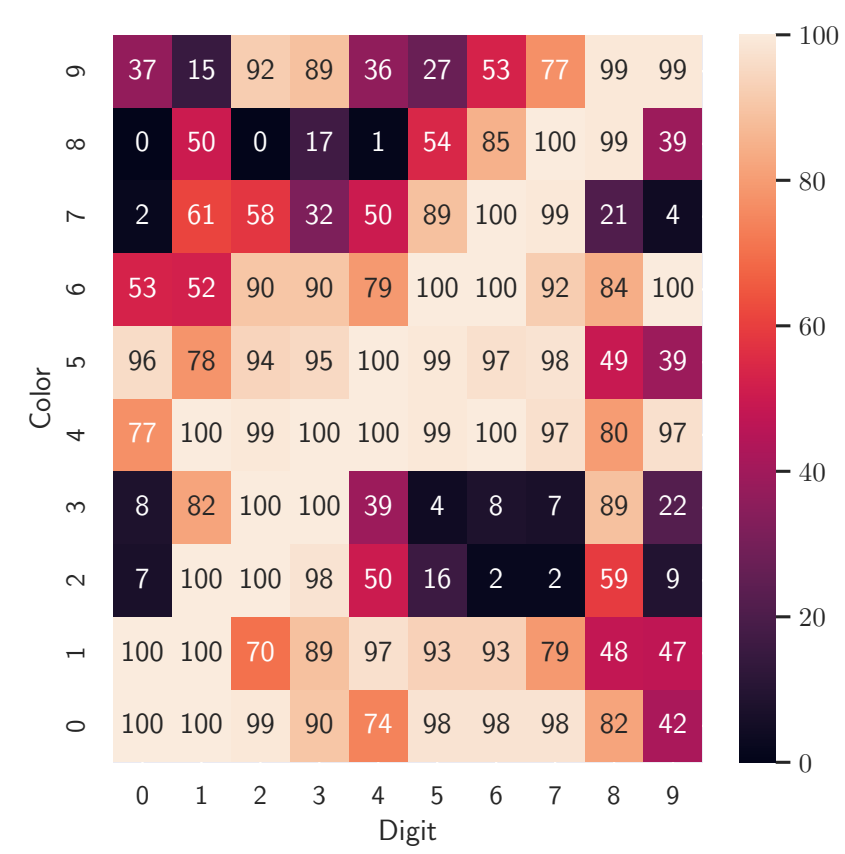

Figure 17: Heatmap showing CS for each attribute combination in the $\wedge$ compositionality task in Colored MNIST generation with partial support (row 10 in Fig. 4a)

### G.4 CoInD also improves conditional generation

| Support | Configuration | CS |
|---|---|---|
| Uniform | Vanilla | 99.98 |
| Uniform | CoInD($\lambda = 1$) | 100 |
| Non-uniform | Vanilla | 99.98 |
| Non-uniform | CoInD($\lambda = 1$) | 99.98 |
| Diagonal partial | Vanilla | 33.14 |
| Diagonal partial | CoInD($\lambda = 0.5$) | 68.82 |

(a) Colored MNIST

| Support | Configuration | $R^2$ | CS |
|---|---|---|---|
| Uniform | Vanilla | 0.99 | 100 |
| Uniform | CoInD($\lambda = 1$) | 0.99 | 100 |
| Orthogonal partial | Vanilla | 0.97 | 95.88 |
| Orthogonal partial | CoInD($\lambda = 1$) | 0.99 | 99.57 |

(b) Shapes3D

Table 7: Overall Performance Metrics for Conditional generation

Given an ordered $n$-tuple from the attribute space not observed during training, can CoInD generate images corresponding to this sampled from joint distribution, $P_\theta(X|C)$? To answer this question, we train CoInD and the baselines on Colored MNIST and Shapes3d datasets. Tab. 7 shows the results. As expected, the vanilla model, under full support, generates samples corresponding to the joint distribution. However, as demonstrated in § 3, models trained on partial support fail to generate samples for unseen attribute compositions. In addition to the improved performance on logical compositionality, enforcing conditional independence explicitly improves conditional generation as well and produces better results on partial support compared to vanilla diffusion models for both Colored MNIST and Shapes3D datasets.

### G.5 CoInD can interpolate between discrete attributes

In some cases, it may be necessary to have control over continuous-valued attributes such as height or thickness. However, the datasets with continuous annotations may not be available to train such models. Or we may be interested in using a pre-trained model that was trained to generate images with discrete attributes. In such cases, *can we generate samples where attributes take arbitrary values that do not belong to the set of training annotations?* We show that CoInD can interpolate between the discrete values of an attribute on which it was originally trained and thus essentially produce images with continuous-valued attributes.

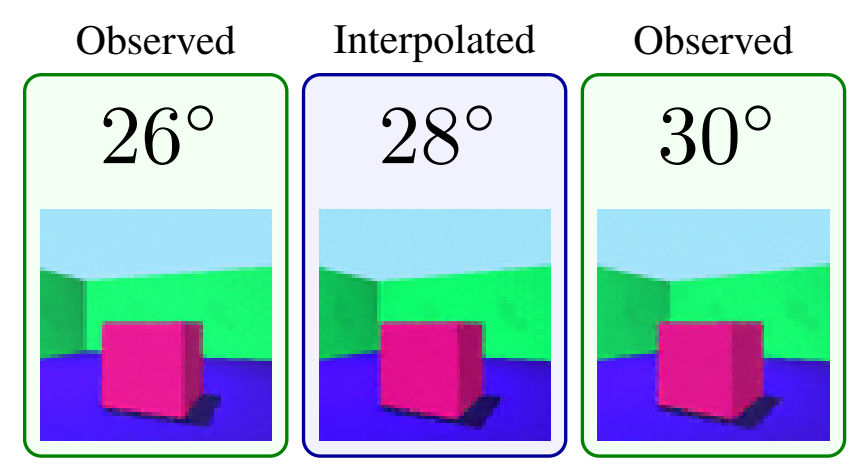

Figure 18: Although CoInD was only trained to generate images with orientations 26° and 30°, it successfully generated a sample with 28° orientation.

As mentioned in the main paper, we trained CoInD to generate images from the Shapes3d dataset using the labels provided in (Kim & Mnih, 2018). The labels provided for the orientation attribute were discrete, although orientation itself is continuous.

In Fig. 18, we highlight the images generated by CoInD where the subject has orientations 26° and 30°. We interpolate between observed discrete values linearly and generate the samples shown in the second column of Fig. 18. By carefully observing the variation in the gap between the corner of the cube and the corner of the room, we notice that CoInD generated an image where the orientation of the cube is midway between those of 26° and 30°. This demonstrates that CoInD offers a promising direction where training on datasets with discrete annotations is sufficient to generate samples with continuous-valued attributes.