# OpenReview forum: "CoInD: Enabling Logical Compositions in Diffusion Models"
_ICLR.cc/2025/Conference — ICLR 2025 Poster_

### Official Review · Reviewer_jrRB · 2024-11-02

**Soundness:** 3
**Presentation:** 3
**Contribution:** 2
**Rating:** 8
**Confidence:** 3

**Summary:**

The authors introduce COIND, a novel approach that enforces conditional independence between marginal distributions by minimizing Fisher's divergence between joint and marginal distributions. Traditional models struggle to handle arbitrary compositions, especially when only partial data compositions are available during training. In comparison, COIND leverages causal independence to improve the generation of images that meet desired attribute compositions. The experiments show COIND’s performance in generating images with diverse and accurate compositions.

**Strengths:**

1. This paper introduces conditional independence analysis into the training of conditional diffusion models, integrating insights from both causal structure learning and generative modeling. By uncovering latent causal structures, it enhances the performance of traditional conditional diffusion models, which is quite innovative.

2. The derivation of error decomposition in Section 4 is particularly impressive. By decomposing the Wasserstein distance, the paper introduces a novel loss function, providing a strong theoretical justification for incorporating causal structures like conditional independence, which is very intriguing.

3. On the selected datasets, the proposed method consistently outperforms comparative approaches across various settings, demonstrating a significant improvement. Moreover, the presentation of experimental results is comprehensive and meticulous. Notably, Appendix A.7 provides an exhaustive comparison of experimental settings across methods, which is commendable.

**Weaknesses:**

1. While the paper uses illustrative images to present the experimental settings for Non-uniform and Partial support scenarios, the lack of mathematical descriptions limits the rigor of this section.

2. The paper’s exploration of causal structures introduced during training remains relatively simple, focusing primarily on conditional independence, and lacks investigation into more complex causal graphs and mechanisms.

3. The study only examines the model's performance on the Colored MNIST and Shapes3D datasets, which, compared to papers like Liu et al. (2023), leaves it without results on more complex, real-world datasets to substantiate the research conclusions.

**Questions:**

1. As pointed out by (Karras et al. (2022), Fu et al. (2024)), generative diffusion models often introduce an early stopping time rather than reverting completely to time zero in the generation process to avoid numerical instability. Did you consider the impact of setting an early stopping time in your experiments? If so, could you elaborate on it? If not, could you explain why this setting might not significantly affect the results?

2. The selection of the hyperparameter λ in the loss function is crucial. Could you share the principles you followed when choosing it (e.g., Bayesian optimization, cross-validation)? The analysis in Section 5.2 does not seem to fully address this aspect.

3. In the experimental scenarios considered in this paper, the features are all  discretized. Do you believe it is possible to extend this method to continuous features, or potentially adapt it with certain techniques to handle continuous feature scenarios?

4. The paper uses Jensen-Shannon Divergence (JSD) to measure conditional independence, though there are many metrics for conditional independence. For instance, Conditional Mutual Information (CMI) (Mukherjee et al., 2020; Li et al., 2023) has shown strong performance for continuous feature scenarios mentioned in Q3. Could you explain the advantages of using the JSD metric in this context?

Reference:
1.	Liu, N., Li, S., Du, Y., Torralba, A., & Tenenbaum, J. B. (2022, October). Compositional visual generation with composable diffusion models. In European Conference on Computer Vision (pp. 423-439). Cham: Springer Nature Switzerland.

2.	Karras, T., Aittala, M., Aila, T., & Laine, S. (2022). Elucidating the design space of diffusion-based generative models. Advances in neural information processing systems, 35, 26565-26577.

3.	Fu, H., Yang, Z., Wang, M., & Chen, M. (2024). Unveil conditional diffusion models with classifier-free guidance: A sharp statistical theory. arxiv preprint arxiv:2403.11968.

4.	Mukherjee, S., Asnani, H., & Kannan, S. (2020, August). CCMI: Classifier based conditional mutual information estimation. In Uncertainty in artificial intelligence (pp. 1083-1093). PMLR.

5. Li, S., Zhang, Y., Zhu, H., Wang, C., Shu, H., Chen, Z., ... & Yang, Y. (2023). K-nearest-neighbor local sampling based conditional independence testing. Advances in Neural Information Processing Systems, 36.

---

> ### Author Response · Authors · 2024-11-22
> **Response**
>
> **W1: Formal Definition of Non-Uniform and Partial Support:** We defined both non-uniform and partial support scenarios in Appendix D.4.
>
> **W2: More Complex Causal Graphs:** There are two causal graphs, the true underlying one and the observed one.
>
> Underlying Causal Graph: Compositionality is formally defined over independently identifiable attributes (see Wiedemer et al., 2024 and references therein). As such, the causal graph we use in the paper satisfies the requirements for compositional generation, which are $C_i\rightarrow X$ and $C_i$ and $C_j$ affect $X$ separately (L120-124). The dependent attribute scenario alluded to by the reviewer does not satisfy the formal definition of compositionality. In other words, one cannot achieve the defined (full) compositionality if attributes are dependent.
>
> Having said that, CoInD can be trivially adapted to any *known* causal graphs between the attributes. To achieve logical compositionality while *respecting* the underlying causal relations, Eq. (4) can be modified to preserve the *known* causal relations. For example, for a three-attribute ($C_1$, $C_2$, and $C_3$) case, if $C_1$ and $C_2$ are causally related to each other and $C_3$ is independent of other attributes, Eq. (4) will be,
>
> $L_{\text{comp}} = W_2\left(p(X\mid C), \frac{p_\theta(X\mid C_1, C_2)p_\theta(X\mid C_3)p_\theta(C_1, C_2)p_\theta(C_3)}{p_\theta(C)p_\theta(X)}\right)$.
>
> One can apply CoInD as we did in the main paper to achieve logical compositionality while *breaking* the underlying causal relations.
>
> Observed Causal Graph: In all experiments, the observed causal graph is more complex, with dependencies between attributes. For example, in the Shapes3d dataset, we considered six attributes that could be varied. These six attributes are correlated in partial and orthogonal support settings, resulting in a complex causal graph.
>
> **W3: Experiments on Complex Real-World Dataset:** We applied CoInD to generate face images with logical compositions on two attributes, “gender” and “smiling,” in two settings: (1) training the model from scratch and (2) fine-tuning an existing text-to-image like SD1.5. We used the CelebA face dataset. We added experimental details in Sec. 5.2 and results in Appendix E.
>
> Additionally, we demonstrated how CoInD can be employed to explicitly and independently control the strength of each attribute in the generated image (Appendix E.2). This is feasible only since CoInD explicitly learns the disentangled scores of $p(X|C_i$).
>
> Also, please see the common response “1. Experiments on Real-world Face Dataset”.
>
> **Q1: Early Stopping:** Thank you for the suggestion. We tried early stopping by Karras et al. (2022) for the partial support scenario on the Colored MNIST dataset. However, the model performance is poor, as we infer from our preliminary results in the table below.
>
> | Method | CS w/o early stopping | CS w/ early stopping |
> | --- | --- | --- |
> | Composed GLIDE | 17.90 | 0.0 |
> | CoInD | 55.16 | 17.19 |
>
> Both Composed GLIDE and CoInD show performance deterioration when early stopping is used. We hypothesize that for unseen compositions, guidance throughout the learning is required for accurate compositions. Karras et al. (2022), unlike our problem of interest, focus on generating seen attribute combinations, so the early stopping idea may not be relevant for compositional generation. We believe a more careful study is warranted.
>
> **Q2: Choice of Hyperparameter $\lambda$:** We added a detailed discussion on choosing $\lambda$ in Appendix C.3. $\lambda$ is selected such that the gradients from $L_\text{score}$ and $L_\text{CI}$ are balanced in magnitude. We did not use cross-validation or Bayesian optimization as they are prohibitively expensive while training diffusion models. Please refer to the common response “5. What is the criterion used to select the hyperparameter λ?” for more details.
>
> **Q3: Continuous Attributes:** Yes, CoInD can natively handle continuous features, simply by using a continuous feature as the conditional input to the learned score function. In fact, in the Shapes3D experiment, we treat the discrete orientation labels as a continuous label in our experiments. In Appendix G.5, we demonstrate CoInD’s ability to generate images corresponding to the interpolation between two samples of continuous attributes observed during training.
>
> **Q4: CMI instead of JSD:** We use JSD as a diagnostic metric for evaluation rather than as an objective to optimize.  In scenarios where the labels are discrete, we employ JSD to measure the distance between the joint likelihood with the product of the marginals and not to learn to enforce conditional independence. The choice of JSD is motivated by the fact that it is symmetric. We will explore if CMI can be used as a metric for Shapes3D where the attributes are continuous. In summary, the choice between JSD and CMI only affects the evaluation of CoInD but not its training.

---

### Official Review · Reviewer_D8GZ · 2024-11-04

**Soundness:** 3
**Presentation:** 3
**Contribution:** 3
**Rating:** 5
**Confidence:** 4

**Summary:**

This paper addresses the limitations of standard diffusion models in sampling data with arbitrary logical compositions of statistically independent attributes. It reformulates the problem as minimizing the Fisher divergence between the joint distribution and the product of marginals, demonstrating the effectiveness of this approach.

**Strengths:**

This paper provides a clear and well-articulated statement of the problem, supported by intuitive examples that help the reader understand the limitations of current conditional diffusion models. It effectively highlights the insufficiencies in these models, particularly in capturing accurate marginal distributions, and offers a novel solution to address these gaps. The authors ensure that the proposed model is designed to learn precise marginal distributions, an essential improvement over existing methods. The paper also includes rigorous experimental validation, utilizing a range of extensive datasets, which adds credibility to the findings and demonstrates the model’s effectiveness across diverse scenarios.

**Weaknesses:**

(1) Some claims in the paper need additional rigor. For instance, the assertion that the drop in conformity scores is due to the violation of independence relations in the learned model lacks clear justification. Providing stronger reasoning or citing relevant references would help substantiate this point.

(2) The proof process could be clarified for readers. For example, the derivation of Equation 6 from previous equations is not immediately clear, and similar issues appear in other parts of the theoretical development. Additionally, some expected theoretical support is missing from the appendix, which could help make the proofs more accessible.

(3) In the experiments section, the model is only evaluated on two real-world datasets, with no simulation results to provide more quantitative insights into the model’s effectiveness. Including simulation studies would strengthen the empirical validation by offering additional quantitative measures.

**Questions:**

(1) How is the failure of standard diffusion models linked to the violation of independence assumptions? Could you provide evidence or proof supporting this claim?

(2) How does the model perform on more diverse datasets? Could additional results on simulated data be included to further validate its effectiveness?

---

> ### Author Response · Authors · 2024-11-22
> **Response**
>
> **W1/Q1: Link between failure of diffusion and violation of independence:** We apologize for any misunderstanding created in our case study in Sec. 3. In the case study, we noted the CS drop between joint $p_\theta(X \mid C)$ and marginal sampling $\frac{p_\theta(X)}{p_\theta(C)}\prod_i \left(\frac{p_\theta(X\mid C_i)p_\theta(C_i)}{p_\theta(X)}\right)$. This drop indicated the violation of conditional independence, i.e., $C_i \not\perp\not\perp C_j\mid X$. This, in turn, is due to learning incorrect marginals $p_\text{train}(X|C_i)$.
>
> Please also refer to the common response for a detailed explanation “3. How is the failure of standard diffusion models linked to the violation of independence assumptions?” We have also restructured Sec. 3 to clarify this and included detailed proofs in Appendices B.1 and B.2.
>
> **W2: Making proofs more accessible:** Thank you for the suggestion. To make the proofs more accessible, we added detailed derivations with intermediate steps in Appendices B (Proofs for Claims) and C (Practical Considerations). To summarize, the proof involves the following steps:
>
> **Step 1:**  We start with 2-Wasserstein distance between the joint likelihood $p_\text{train}(X\mid C)$ and the product of marginals $\frac{p_\theta(X)}{p_\theta(C)} \prod_i \frac{p_\theta(X \mid C_i)p_\theta(C_i)}{p_\theta(X)}$.
>
> **Step 2:** We apply triangle inequality on this distance w.r.t. the learned joint likelihood $p_\theta(X\mid C)$.
>
> **Step 3:** Kwon et al. (2022) showed that the 2-Wasserstein distance between two distributions is upper bounded by the expectation of the $L_2$ norm of the difference in their scores (Eq. 16 in Kwon et al., (2022)). We apply this result to each term to get their corresponding upper bounds.
>
> **W3/Q2: Diverse Datasets and More Quantitative Insights:** The submitted paper already provides quantitative insights on two synthetic datasets under four different support settings through 2 metrics (CS and JSD).
>
> In the revised paper, we added an experiment with a more realistic CelebA dataset (Tab. 2) and compared the effectiveness with CS, FID, and Shannon entropy metrics.
>
> In summary, the revised paper provides the following quantitative insights:
>
> 1. **Conditional independence:** JSD in Figs. 4a, 6a, and Tab 2 demonstrate that baseline methods have higher JSD than CoInD. We conclude that CoInD respects conditional independence, $C_i\not\perp\not\perp C_j\mid X$, better than the baselines.
> 2. **Fine-grained attribute control:** Fig 11 shows that CoInD also provides fine-grained control compared to baselines (Sec. 5.2 and App. E.2).
> 3. **Diversity of unconditioned attributes:** Shannon entropy in Fig. 5 confirms that CoInD is more diverse in unconditioned attributes.
>
> If the reviewer has further specific suggestions for quantitative insights, we will happily include them in the paper.
>
> If our responses addressed your initial comments, please consider raising the score.

---

### Official Review · Reviewer_BxVf · 2024-11-05

**Soundness:** 3
**Presentation:** 4
**Contribution:** 3
**Rating:** 6
**Confidence:** 4

**Summary:**

This paper performs compositional image generation with and, or, and not operators. The proposed method trains the diffusion models with a score function that enforces the attributes in the generated images to be independent. Finally, the authors show performance on two synthetic datasets.

**Strengths:**

The paper is very well written. Every detail is provided in a clear manner. The description is intuitive. The authors also provided insightful observations.

**Weaknesses:**

### Minor weakness:
* Some motivational examples of the utility of the proposed method in the introduction would be nice.
* The conformity score with notation should be defined/discussed briefly in the experiment section again.
* An initial look at Figure 3 might give readers the impression that the proposed approach is effective for such scenarios as well. This should be made clearer.
* The numbers in line 417 can be written in a more detailed way.
* The authors should give some examples of open and closed set attribute compositions.

### Major weakness:
* The only major issue with this paper is that there is no experiment on a real-world image dataset. One of the baselines, LACE, presents results for human faces with independent attributes, e.g., smile, eyeglasses, etc. Can the authors show performance on such datasets?

**Questions:**

### Questions:

* What happens if $\lambda$ is set to a very high value? Would that give perfect conditional independence? Will that be an issue while minimizing $L_{\text{score}}$?
* What issues will the proposed algorithm face if the attributes are actually dependent in the true data-generating process?
* How many attributes can be composed together? How does the proposed approach behave when you increase the number of attributes?
* Based on Figure 3: if the training dataset has no [color: green, digit: 9], can the model generate such images? The proposed approach has a 55% conformity score (CS) for partial support. Does that mean that, among the generated images with the attributes [color: green, digit: 9], 55% contain these attributes correctly?
* Can the authors provide images with minimal attribute combinations where the proposed method, along with all baselines, fails?

---

> ### Author Response · Authors · 2024-11-22
> **Response (part 1/2): Weakness**
>
> **W1: Experiments on Real-World Image Dataset:** We applied CoInD to generate face images with logical compositions on two attributes, “gender” and “smiling,” in two settings: (1) training the model from scratch and (2) fine-tuning an existing text-to-image like SD1.5 (Appendix E.3). We used the CelebA face dataset. We added experimental details in Sec. 5.2 and results in Appendix E.
>
> Additionally, we demonstrated how CoInD can be employed to explicitly and independently control the strength of each attribute in the generated image (Appendix E.2). This is feasible only since CoInD explicitly learns the disentangled scores of $p(X|C_i$).
>
> Also, please see the common response “1. Experiments on Real-world Face Dataset”.
>
> **W2: Motivation Examples for CoInD:** Thank you for your suggestion. We have added image editing as an application that could benefit from logical compositionality in Sec. 1 (Introduction). More applications and an example are provided in the common response “2. What are the practical applications of composition generation?”.
>
> **W3: Conformity Score Definition:** We added a discussion of conformity score in the experimental section (Sec. 4). We have also made a minor correction to the definition of conformity score at L203:
>
> $\text{CS}(g) = E_{p(C)p(U)}\left[\prod_i 1(C_i, \phi_{C_i}(g(C, U))\right]$
>
> where $C$ is the desired attribute combination at test time, $U$ denotes the stochastic factors in the generative model $g$, and $\phi_{C_i}$ denotes the attribute classifier that predicts the value assumed by $C_i$ from a generated image. $1(\cdot, \cdot)$ is the indicator function. $p(C)$ distribution corresponds to that of the seen combinations for the uniform scenario and of unseen combinations for other scenarios. We updated the PDF accordingly in both Sec. 3 (case study) and 4 (experimental evaluation) to describe what CS truly measures.
>
> **W4: Effectiveness of CoInD for Orthogonal Support (Fig. 3):** Yes, the proposed approach is effective for the orthogonal support scenario shown in Figure 3. Fig. 6(a) and Tab. 2 provide quantitative evidence that CoInD outperforms the baselines in this scenario.
>
> **W5: Numbers in Line 417:** We modified the lines (L424-425 in the revised paper) to emphasize CoInD's improvement over the baselines.
>
> **W6: Examples of open and closed set compositions:** By open-set composition, we believe the reviewer meant generating images where attributes take values never seen in the training set. For example, in the Colored MNIST experiment, this corresponds to generating a digit in a new 11th color that has never been observed in the training set. We have added this example to L147-149 in Sec. 2. As we mentioned in Sec. 2, the open-set composition is beyond the current scope of our paper and remains an exciting avenue for future work. Also, please see the common response “6. Open set vs closed set composition”.

---

> ### Author Response · Authors · 2024-11-22
> **Response (part 2/2): Questions**
>
> **Q1: Effect of Large $\lambda$:** Yes, a high λ value gives perfect conditional independence and also affects how well $\mathcal{L}_\text{score}$ is optimized. Fig. 8 shows that CS of CoInD on Colored MNIST with partial support drops for $\lambda > 1$. A model trained with a very large $\lambda$ value, thus, ignores the attribute conditioning and instead generates unconditional images which is a trivial, yet undesirable, solution to enforce conditional independence. Also, please see the common response “5. What is the criterion used to select the hyperparameter λ?” for more details on our hyperparameter selection.
>
> **Q2: Dependent Attributes in Data Generating Process:** CoInD is flexible to the underlying causal relations between the attributes and can be modified according to the goal, i.e. whether to break known dependence for conditional generation or to improve logical compositionality while maintaining the known dependence relations. Please see to the common response “4. What issues will the proposed algorithm face if the attributes are actually dependent on the true data-generating process?” for a more detailed answer.
>
> **Q3: Scalability of CoInD w.r.t. Number of Attributes:** Our experiments show composition with 6 attributes in Shapes3D. CoInD achieves a high CS even in this case, demonstrating its scalability. Increasing the number of attributes to investigate the scaling law of CoInD requires retraining the diffusion model, which we could not finish in the limited review time. We will add this to the next revision of the submission.
>
> **Q4: Orthogonal Support and CS Measurement:**
>
> *Orthogonal support (Fig. 3):* Yes, CoInD can generate such unseen images. We show one such example of an unseen combination in Fig. 1(d) [color: cyan digit: 4].
>
> *CS measurement:* Yes, on average, 55% of the images contain the desired attributes correctly.  Conformity score (CS) is the average across all *unseen* attribute combinations. We added the CS for each attribute combination for the partial support scenario in Colored MNIST in Fig. 16 in App. G.3.
>
> Overall, Fig 4a of the paper shows that CoInD outperforms the baselines by a huge margin on the unseen attributes. We reproduce this table from the main paper to emphasize the performance gains.
>
> | Method | ∧ (CS) ↑  | ¬ Color (CS)↑ | ¬ Digit (CS) ↑ |
> | --- | --- | --- | --- |
> | LACE | 22.94 | 15.41 | 7.39 |
> | Composed GLIDE | 17.90 | 25.00 | 4.13 |
> | CoIND | **55.16** | **55.47** | **53.34** |
>
> While the baselines achieve a mere 22% CS score, CoInD achieves 55%, representing a significant improvement.
>
> **Q5: Failed Image Generations:** Thank you for the suggestion. We added failed generated images in Appendix G.2 for Colored MNIST, Shapes3d, and CelebA datasets. CoInD sometimes leaks attributes from the closest seen attribute combination.
>
> If our responses addressed your initial comments, please consider raising the score.

---

### Official Review · Reviewer_oaF6 · 2024-11-07

**Soundness:** 2
**Presentation:** 2
**Contribution:** 3
**Rating:** 6
**Confidence:** 3

**Summary:**

This paper shows that standard conditional diffusion models violate this assumption, even when all attribute compositions are observed during training. The authors propose COIND to address this problem. It explicitly enforces statistical independence between the conditional marginal distributions. Quantitative experiments demonstrate a significantly more faithful and controlled generation of samples for arbitrary logical compositions of attributes. The benefit is more pronounced for scenarios that current solutions relying on the assumption of conditionally independent marginals struggle with, namely, logical compositions involving the NOT operation and when only a subset of compositions are observed during training.

**Strengths:**

The problem is interesting and the authors proposed a solution to the problem. The writing of the paper is clear and easy to follow. The method is simple and effective. The authors provide experiments to validate the proposed method.

**Weaknesses:**

1. The datasets in the paper are simple MNIST and synthetic 3D shapes. Experiments on real-world images could strengthen the paper. Experiments on natural images could provide more convincing evidence for the proposed method.

2. What are the practical applications of NOT operation in the composition generation? What are the motivations for applying logic operations on the attributes with multiple values, e.g. c_i  in  [0, 1, 2, 3, 4, 5]?

3. How does the model process logic expressions? How is the model implemented to process logic expressions? If not implemented, what are the motivations for applying logic operations? Are AND operations sufficient for composition generation with different attributes?

**Questions:**

See weaknesses.

---

> ### Author Response · Authors · 2024-11-22
> **Response**
>
> **W1: Experiments on Real-World Images:** Thanks for a very great suggestion! We applied CoInD to generate face images with logical compositions on two attributes, “gender” and “smiling,” in two settings: (1) training the model from scratch (Sec 5.2), and (2) fine-tuning an existing text-to-image like SD1.5 (Appendix E.3). We used the CelebA face dataset. We added experimental details in Sec. 5.2 and results in Appendix E.
>
> Additionally, we demonstrated how CoInD can be employed to explicitly and independently control the strength of each attribute in the generated image (Appendix E.2). This is feasible only since CoInD explicitly learns the disentangled scores of $p(X|C_i$).
>
> Also, please see the common response “1. Experiments on Real-world Face Dataset”.
>
> **W2: Practical Applications of NOT Operation:** NOT composition is a valuable tool in controllable image generation. For example, suppose one wants to generate an image of a cow. The resulting image is that of a cow in a pastoral background. NOT composition allows the user to have more control over the background attribute. For creative purposes, the user may ask to generate an image with the attributes “cow, NOT pasture”.
>
> Please also refer to the common response “2. What are the practical applications of composition generation?” for applications and examples of logical compositionality.
>
> **W3: Why Logic Operations on Attributes with Multiple Values:** Our work aims to facilitate controllable image generation. Restricting the model’s choice in some attributes is valuable for controllable image generation. For instance, when generating thousands of images, it is desirable to have diversity in attribute values. However, through this logical formulation, we can afford users explicit control over this diversity.
>
> A real-world example is “generate an image of a butterfly with [blue OR orange OR white] color.” Since butterflies come in several colors, using a logical composition with multiple choices is better than excluding several other possible colors to get limited diversity.
>
> **W4: Motivation for and processing of Logical Expressions:**  *Motivation for Logical Expressions:* Logical operations are preferred for controllable generation since they (1) offer fine-grained control over the desired attributes and (2) can be interpreted directly. Also, see “2. What are the practical applications of composition generation?” in our common response.
>
> *Processing Logical Expressions:* We use the probability formulation of the logical expression provided by Li et al. (2023) and Du et al. (2020). Here, logical expressions are expressed in terms of score functions of likelihood. Eqs. (1) and (2) in the preliminary section show these formulae for AND and NOT operations. L162-172 in Sec.1 in our paper discusses these in detail.
>
> The outputs of diffusion models directly give the score functions of likelihood terms. To obtain images, we substitute these outputs in Eq. (1) and (2) and sample according to Langevin dynamics given in Eq. (11). Please refer to Appendix A for a detailed description of the preliminaries of diffusion models.
>
> **W5: Is AND Enough For Logical Compositions:** No, AND is not sufficient. One must combine primitive operations: OR, NOT, AND to support arbitrarily complex logical operations. For example, the NAND operation is NAND(X, Y) = NOT (X AND Y).
>
> If our responses addressed your initial comments, please consider raising the score.

---

> > ### Comment · Reviewer_oaF6 · 2024-11-25
> >
> > Thank you for your response. Most of my concerns have been addressed. I will keep the score unchanged.

---

> > > ### Author Response · Authors · 2024-11-27
> > > **Response from the authors**
> > >
> > > Thank you for your reply to our rebuttal. If any part of our response to your initial comments was unclear, please let us know.
> > >
> > > We also welcome any additional feedback or suggestions you may have to further improve our manuscript.

---

### Official Review · Reviewer_5mnE · 2024-11-09

**Soundness:** 3
**Presentation:** 3
**Contribution:** 3
**Rating:** 6
**Confidence:** 3

**Summary:**

This paper considers the problem of conditional generation for logical compositions of attributes. The authors claim that existing conditional diffusion models violate the conditional marginal independence assumption of the attributes. This makes them struggle in generating samples given arbitrary logical compositions of attributes, even if trained with uniform data over all possible combinations of attributes, and even worse if there is a statistical correlation or only some attribute combinations are seen. The paper then tackles this problem by training score-based models (with their diffusion model interpretation) with an additional loss term to enforce this conditional independence assumption, motivated by Fisher’s divergence between the joint and product of marginals of attributes. The assumption about marginal independence empirically validated by observing that Jensen-Shannon divergence of joint distribution and the product of conditional marginal distributions has a negative correlation with an accuracy metric called conformity score (CS).

**Strengths:**

The paper addresses the problem of generating samples for logical compositions of attributes, including combinations previously not seen in data, which is highly relevant.

The proposed approach trains a single model to handle arbitrary compositions of attributes, and thus scales with the number of attributes. The proposed loss function is derived from a theoretically sound objective based on the Wasserstein distance between the true conditional likelihood and the learned likelihood satisfying the conditional independence assumption.

Experiments are thorough, including different support settings (uniform, non-uniform, partial/orthogonal) for two benchmark datasets, and the results support the claims about conditional independence assumption.

The paper is overall well structured, although some details could be clarified (see below).

**Weaknesses:**

I have a couple of concerns about soundness and clarity. While the proposed loss function (Eq 6 and 7) has a clear theoretical justification of bounding Wasserstein distance, this is somewhat lost in the actual training objective of CoInD (Eq 8 and 9) due to simplifications for practical implementation. It was also not very clear how the modified L_{CI} in Eq 9 is derived.

In several places (including the problem statement), the authors seem to conflate the distribution of attributes being uniform vs. conditionally independent. The conditional distribution of attributes can be factorized without necessarily being uniform. Another issue is that even if each attribute had uniform probability, after conditioning on a logical formula in general, each marginal probability may no longer be uniform.

Moreover, the paper mentions “incorrect marginals” multiple times as being a limitation of current approaches, but I believe the main claim is about conditional independence (factorizability). It would be clearer to refer to independence directly because it’s possible to have the correct marginal probability of each attribute while still violating the conditional independence assumption.

As the authors also discuss, the proposed method is restricted to a composition of a closet set of attributes, and cannot easily add compositional attributes without retraining from scratch. However, while a limitation, open-set compositions are outside of the scope of this paper and not a major weakness.

**Questions:**

I found the definition of conformity score to be not very clear. First, I assume results are reported in percentage rather than as defined (between 0 and 1). My main confusion is about the distribution p(X,C) in the definition (line 199). For the generated samples, is the score measuring whether all consistent combinations of attributes appear following a certain distribution, or is it only measuring whether the attributes (given by the classifier) satisfy the logical composition?

Can this method be used for compositions using general logical formulas beyond single AND and NOT operations?

Can CoInD generate attributes having non-uniform probability (while still being conditionally independent)? For instance, to generate shapes in a scene following a certain distribution, or images for demographic groups following a certain population distribution.

The causal graph in Fig 2a implies independence among the attributes, but not necessarily conditional independence given X. It wasn’t clear whether (around line 121) the claim is that conditional independence is implied by this causal graph or it is an additional assumption made by the authors.

---

> ### Author Response · Authors · 2024-11-22
> **Response (part 1/2): Weaknesses 1-3**
>
> **W1: Soundness and Clarity of Equations:** First, we list our loss function and the variations we consider. The equation numbers correspond to the revised PDF.
>
> Eq. (7): $L_{\text{CI}} = E_{p(X, C)}\| \nabla_X\log p_\theta(X\mid C) - \nabla_X\log p_\theta(X) - \sum_i \left[\nabla_X\log p_\theta(X\mid C_i) - \nabla_X\log p_\theta(X)\right]\|_2^2$      (CI loss)
>
> $L_{\text{CI}} = E_{p(X, C)}E_{j,k\sim\mathcal{U}(n)}\| \nabla_X\log p_\theta(X\mid C_j, C_k) - \nabla_X\log p_\theta(X\mid C_j) - \nabla_X\log p_\theta(X\mid C_k) + \nabla_X\log p_\theta(X)\|_2^2$              (scalable CI loss)
>
> Eq. (8): $L_{\text{comp}} \leq K_1\sqrt{L_{\text{score}}} + K_2\sqrt{L_{\text{CI}}}$        (theoretical bounds)
>
> Eq. (9): $L_{\text{final}} = L_{\text{score}} + \lambda L_{\text{CI}}$                         (CoInD loss)
>
> The purpose of Eq. (8) was to show that an upper bound of the distance between the true distribution $p(X\mid C)$ and its parameterized causal factorization $\frac{p_\theta(X)}{p_\theta(C)} \prod_i \frac{p_\theta(X\mid C_i)p_\theta(C_i)}{p_\theta(X)}$ consisted of the score term $L_{\text{score}}$ and the conditional independence term $L_{\text{CI}}$.
>
> We found directly optimizing Eq. 8 to suffer from some instability due to larger gradient magnitudes and needs more careful learning rate scheduling. With such tuning, optimizing Eq. 8 gives similar conclusions as Tables 3 and 4 in Appendix C.2. For larger scale experiments we preferred Eq. 9 since we could leverage optimization hyperparameters (learning rate, learning rate schedule, etc.) of existing well-tuned implementations of diffusion models.
>
> The conditional independence in Eq. (7) is linear w.r.t. the number of attributes $n$. However, due to Hammon and Sun, (2006), conditional independence can be approximated with pairwise conditional independence for large $n$. Therefore, using our scalable CI loss, we enforce conditional independence between a pair of attributes $(C_j, C_k)$, randomly sampled every batch. Refer to Appendix C for a detailed derivation.
>
> **W2: Uniform Distribution and Conditional Independence:** Yes, the attributes being uniform and being independently conditioned on $X$ are two different ideas. CoInD’s objective is to enforce conditional independence between the attributes irrespective of the nature of their distributions. We used four scenarios — uniform, non-uniform, partial, and orthogonal supports —  to derive and verify the effect of the learned conditional independence on logical compositionality in diffusion models. These scenarios were used because they (1) clearly depicted the underlying causal model that we wished to study, and (2) are widely seen in practice. Similar scenarios were considered in Wiedemer et al., (2024) (Sec. 3.1, Fig. 3) and Schott et al., (2020) (Sec. 2, Fig. 2).
>
> CoInD does not require $p(C_i)$ to be uniform. We have added the following scenario to the Colored MNIST experiment to demonstrate this point:
>
> Let $C_1$ and $C_2$ be color and digit in Colored MNIST where both $C_1$ and $C_2$ follow a non-uniform categorical distribution (similar to discrete Gaussian). During inference, we task the models with generating images corresponding to arbitrary attribute combinations. Refer to Appendix G.1. The following table compares the performance of CoInD against the baselines.
>
> | Method | JSD ↓ | ∧ (CS) ↑ | ¬ Color (CS) ↑ | ¬ Digit (CS) ↑ |
> | --- | --- | --- | --- | --- |
> | LACE | - | 89.22 | 58.59 | 57.81 |
> | Composed GLIDE | 0.27 | 91.74 | 88.91 | 78.39 |
> | CoInD | **0.16** | **99.61** | **98.51** | **83.03** |
>
> **Conclusions:**
>
> 1. Similar to our previous results on Colored MNIST, CoInD achieves better CS on all tasks with around 5-10% and 10-40% improvement in CS over Composed GLIDE and LACE respectively.
> 2. The learned $p_\theta(C_i\mid C_j)$ is similar to the true distribution $p_\text{true}(C_i\mid C_j)$ as shown in Figure 14(c) in Appendix G.1.
>
> **W3: Not Uniform After Logical Conditioning:** Yes, it is true that when conditioned on a logical expression, the marginal probability $p(C_i\mid\text{logical expression})$ will not be uniform even if the true marginal distribution of the attributes $p(C_i)$ is uniform. This is expected since the purpose of conditioning is to restrict the space of generated images, and thus by definition, it will no longer be uniform. It is not clear to us why and in what form this represents an “issue”.

---

> ### Author Response · Authors · 2024-11-22
> **Response (part 2/2): Weaknesses 4-6, Questions 1-3**
>
> **W4: Generate Attributes with Non-Uniform Probability:** Yes, CoInD can generate attributes with non-uniform probability as demonstrated by the above Gaussian support experiment. If only some attributes are conditioned in the logical expression, as we showed in the discrete Gaussian example above, the unconditioned attributes take their true conditional distribution $p_\text{true}(C_{\text{uncond}} \mid C_{\text{cond}})$.
>
> **W5: Incorrect Marginals and Violation of Conditional Independence:** Yes, the limitation we refer to in the paper is about the violation of conditional independence, but this is caused since the marginals are learned incorrectly. Please refer to the common response “3. How is the failure of standard diffusion models linked to the violation of independence assumptions?” for a detailed answer. We have also updated Sec. 3 in the main paper accordingly and have added Appendices B.1 and B.2 to explain the same.
>
> **W6: Open-Set Composition:** Yes, open-set composition is beyond the scope of this paper as we aim to achieve controllable generation with known attributes. Please refer to the common response “6. Open set vs closed set composition”.
>
> However, we showed that logical compositionality can be achieved by fine-tuning an existing model (i.e., without training from scratch). In Appendix E.3, we fine-tuned Stable Diffusion v1.5 to generate face images where we controlled gender and smile attributes.
>
> **Q1 Definition of conformity score:** CS measures how many of generated images have attributes (given by the classifier) that satisfy the desired logical composition. Formally, it is defined as,
>
> $\text{CS}(g) = E_{p(C)p(U)}\left[\prod_i 1(C_i, \phi_{C_i}(g(C, U))\right]$
>
> where $C$ is the desired attribute combination at test-time, $U$ denotes the stochastic factors in the generative model $g$, and $\phi_{C_i}$ denotes the attribute classifier that predicts the value assumed by $C_i$ from a generated image. $1(\cdot, \cdot)$ is the indicator function. $p(C)$ distribution corresponds to that of the seen combinations for the uniform setting and of unseen combinations for other settings. These images are generated by passing the expected attributes in the form of logical compositions. We have updated Sec. 3 (case study) of the PDF accordingly.
>
> **Q2: General Logical Formulae:** Yes, CoInD is applicable for general logical formulae. We demonstrated the capability of CoInD for AND, NOT, and OR compositions since any logical composition can be expressed through compositions of these primitives. See examples of general logical formulae in Fig 1d.
>
> **Q3: Implied Claim or Assumption:** Yes, this is an additional assumption made in L123. We have clarified this in L120. Apart from the independence of the attributes $C_i$’s, we also assume that they affect the image $X$ separately, i.e. their effects are disentangled. As a result, $C_i$’s are conditionally independent given $X$. This assumption of an invertible mixing function is commonly used in causal representation learning.
>
> If our responses addressed your initial comments, please consider raising the score.

---

### Author Response · Authors · 2024-11-22
**Common Response (part 1/4)**

We sincerely thank all the reviewers for their thoughtful comments and valuable feedback.We are encouraged that the reviewers found our paper insightful (**BxVf**) and novel (**D8GZ, jrRB**). We are also happy that the reviewers noted our proposed method for controllable image generation as simple and effective (**oaF6, D8GZ, jrRB**), supported by theory (**5mnE, jrRB**) and that the paper is well-written (**oaF6, BxVf, D8GZ**). Your insights have significantly improved our method, particularly in extending its applicability beyond synthetic data.

Incorporating the feedback from the reviews, we have made the following updates to the paper:

- **Experiments on Real Datasets (Appendix E)**: We demonstrate that CoInD can successfully generate real-world face images satisfying logical expressions. For this experiment, we used gender and smiling attributes.
- **Practical Application of Logical Compositionality (Appendix E.2)**: Our approach allows for fine control over the amount of specific attributes in generated images. This result opens up a new direction for controlled generation in diffusion models, which we believe will be of significant interest to the community.
- **Detailed Proofs (Appendix B)**: We have included a detailed step-by-step walkthrough of the proofs mentioned in the paper. This section clarifies and explains the theoretical foundations of our work.
- **Hyperparameter Choices (Appendix C.3)**: We added a principled explanation of our algorithm's choice for our hyperparameter $\lambda$.
- **Title change**: We shortened the title to “CoInD: Enabling Logical Compositions in Diffusion Models” to avoid confusion caused by the lack of sufficient context for the words “conditionally independent marginals” in our previous title. We hope the reviewers agree that the new title is more transparent and still accurately captures the scope of the paper.


**1. [**oaF6**, **BxVf**, **D8GZ**, **jrRB**]: Experiments on Real-world Face Dataset**

We applied CoInD to generate real 128x128 face images. We used the CelebA dataset, with “smiling” and “gender” as the attributes for logical compositions. we induce a dependence between these attributes such that all images except smiling males (smiling = true, gender = male) are seen. During generation, the goal is to sample images of unseen “smiling males”. The evaluation results are provided in Appendix E. We conducted three experiments:

1. Train the model from scratch.
2. Fine-grained and independent control of each attribute’s strength in the generated image.
3. Fine-tune a pre-trained text-to-image model like SD1.5.

**Train the model from scratch. (Sec. 5.2, App. E.1)**

| Method | JSD ↓ | CS (smiling male) ↑ | FID (smiling male) ↓ |  ∧ CS (smiling ∧ male) ↑ |  ∧ FID (smiling ∧ male) ↓ |
| --- | --- | --- | --- | --- | --- |
| Composed GLIDE | 0.394 | 6.50 | 60.36 | 19.53 | 98.70 |
| CoInD | **0.165** | **23.10** | **42.23** | **30.40** | **39.58** |

Table: Evaluation of CoInD on face image generation when trained from scratch

Conclusions:

1. CoInD outperforms the baseline by >16% and >10% CS in joint sampling (smiling male) and AND ( ∧ ) compositionality respectively.
2. CoInD does so while generating images whose distribution is closer to the real image distribution, as measured using FID.

As a reminder, JSD is the Jensen-Shannon divergence between the joint likelihood and the product of marginals and measures the violation of conditional independence. CS is the conformity score measuring the accuracy with which the model generates the desired attributes in the image. FID is Frechet Inception distance that measures the distance between the distribution of the generated images and the real images (smiling = true, gender = male).

---

> ### Author Response · Authors · 2024-11-22
> **Common response (part 2/4)**
>
> **Fine-grained Adjustment of Attributes in Generated Images using CoInD (Sec. 5.2, App. E.2)**
>
> We evaluate CoInD and Composed GLIDE on the task of adjusting the amount of “smiling” in the image by adjusting a scaling factor $\gamma$ that varies the amount of “smiling” in an image. Refer to Eq. (34) for the exact formulation.
>
> | **γ** | **CS** ↑ **CoInD** | **CS** ↑  **Composed GLIDE**  | **FID**↓  **CoInD** |  **FID** ↓ **Composed GLIDE**  |
> | --- | --- | --- | --- | --- |
> | 1 | **30.40** | 19.53 | **39.58** | 98.70 |
> | 2 | **69.55** | 23.01 | **36.04** | 119.21 |
> | 6 | **91.40** | 45.31 | **42.60** | 96.76 |
>
> Conclusions:
>
> 1. CoInD achieves nearly $2\times$ the CS of Composed GLIDE on increasing $\gamma$.
> 2. CoInD achieves better (lower) FID than Composed GLIDE.
> 3. Fig. 11 in the Appendix qualitatively compares the samples generated by CoInD and the baseline for varying $\gamma$. CoInD allowed us to vary the amount of “smiling” in the generated images without affecting the gender-specific attributes. However, Composed GLIDE associated the smile attribute with the gender attribute due to their association in the training data. Hence, it generated images with gender-specific attributes such as long hair, which explains Composed GLIDE's worse FID scores.
>
> **Real-world face image generation by fine-tuning a T2I model  (App. E.3)**
>
> Table: Evaluation of CoInD on face image generation when T2I is fine-tuned
>
> | Method | JSD ↓ | CS (Joint) ↑ | FID (Joint) ↓ | CS (∧ Composition) ↑ | FID (∧ Composition) ↓ |
> | --- | --- | --- | --- | --- | --- |
> | Composed GLIDE | 0.57 | 56.57 | 58.31 | 14.19 | 73.53 |
> | CoInD | **0.37** | **58.57** | **58.19** | **49.15** | **61.16** |
>
> Conclusion:
>
> 1. CoInD outperforms Composed GLIDE in CS and FID on joint sampling and AND composition tasks. The quantitative results are given below. For more results and details, refer to Appendix E.3.
>
> **2. [**BxVf**, **oaF6**] What are the practical applications of composition generation?**
>
> Many practical scenarios desire explicit control over many attributes of the generated images. For example, suppose you wish to generate a face image of an old man without a beard. This requirement can be expressed as a logical relation in terms of known attributes as “man” AND “old” AND NOT “beard.” Furthermore, logical compositionality is a generalization of the widely studied conditional generation.
>
> Apart from the use cases mentioned above and those discussed in prior work (Liu et.al. (2023), Du et al., (2020), Nie et al., (2021)), we demonstrate that explicitly learning the disentangled scores of $p(X|C_i$) enables diffusion models to afford fine-grained control over the attribute’s strength. For example, we control the amount of smiling in the generated face images through the expression in Eq. (34) in Appendix E.2. These results offer a new direction for fine-grained attribute control in diffusion models.

---

> > ### Author Response · Authors · 2024-11-22
> > **Common Response (part 3/4)**
> >
> > **3. [**5mnE**, **D8GZ**] How is the failure of standard diffusion models linked to the violation of independence assumptions?**
> >
> > We want to address and clarify any misunderstandings regarding the failures of compositionality in diffusion models.
> >
> > The standard diffusion model objective is not suitable for logical compositions as summarized below where $\rightarrow$ means “leads to”.
> >
> > Incorrect marginals $\rightarrow$ Violation of conditional independence $\rightarrow$ Inaccurate compositions (low conformity score).
> >
> > We give a step-by-step description of how we arrived at this reasoning below.
> >
> > **Step 1:** The causal graph shown in Figure 2(a) provides us with the following conditional independence relation:
> >
> > $p(C\mid X) = \prod_i p(C_i\mid X)$                                (CI relation)
> >
> > This CI relation is used by several works (Liu et al., 2023; Nie et al., 2021), including ours, to derive the expression for the joint distribution $p(X\mid C)$ in terms of the marginals $p(X\mid C_i)$ for logical compositionality.
> >
> > **Step 2:** The joint likelihood is written as the product of marginals using the CI relation and the causal factorization as,
> >
> > $p(X\mid C) = \frac{p(X)}{p(C)}\prod_i \left(\frac{p(X\mid C_i)p(C_i)}{p(X)}\right)$                  (JM relation)
> >
> > Note that CI relation is crucial for JM relation to hold. We sample from joint likelihood using the score of LHS of JM relation, referred to as joint sampling in Sec. 3. Similarly, we also sample using the score of RHS of JM relation, referred to as marginal sampling in Sec. 3.
> >
> > **Step 3:** Theoretically, if the learned generative model satisfies the JM relation, there should not be any difference in the CS between joint and marginal sampling. However, we see a drop in CS (Tab. 1 in the main paper), implying that the JM relation is violated in the learned model.
> >
> > **Step 4:** If the model has learned the CI relation, the JM relation must hold in the learned generative model. Therefore, we check if the CI relation holds in the model by measuring JSD between LHS and RHS of CI relation. The results in Tab. 1 confirm that the CI relation **does not** hold. This is a significant finding since existing works blindly trust the model to satisfy CI relation. This leads to severe performance drop (lower CS) when the training support is non-uniform or partial.
> >
> > **Step 5:** Vanilla diffusion models do not respect the underlying CI relation because the training objective maximizes $p_\text{train}(X\mid C_i)$ instead of the true marginal $p_\text{true}(X\mid C_i)$ when $C_i\not\perp\not\perp C_j$. Refer to App. B.2.
> >
> > To summarize, as explained in Appendix B.3, we proposed CoInD to ensure the JM relation is satisfied by explicitly modeling the joint as the product of the marginals according to the causal factorization (Eq. (4)). In the experiments section, we used JSD as a diagnostic measure, and in Fig. 4(b), Fig. 6(a), and Tab. 2, we showed that CS increased as JSD decreased.
> >
> > **4. [**jrRB**, **BxVf**] What issues will the proposed algorithm face if the attributes are actually dependent on the true data-generating process?**
> >
> > CoInD can be trivially adapted to any *known* causal relations between the attributes in the true data-generating process. For logical compositionality, while respecting the underlying causal relations, Eq. (4) can be modified to preserve the *known* causal relations. For example, for a three-attribute ($C_1$, $C_2$, and $C_3$) case, if $C_1$ and $C_2$ are causally related to each other and $C_3$ is independent of other attributes, Eq. (4) will be $L_{\text{comp}} = W_2\left(p(X\mid C), \frac{p_\theta(X\mid C_1, C_2)p_\theta(X\mid C_3)p_\theta(C_1, C_2)p_\theta(C_3)}{p_\theta(C)p_\theta(X)}\right)$.
> >
> > Having said that, compositionality is formally defined over independently identifiable attributes in the true data-generating process (Wiedemer et al., 2024). As such, the dependent attribute scenario in the true data-generating process does not satisfy the formal definition of compositionality. In other words, one cannot achieve the defined (full) compositionality if attributes are dependent in the true data-generating process.
> >
> > Directly applying CoInD to the scenario where attributes are dependent in the true data-generating process will *break* those dependencies, but could lead to relatively poor independent control.

---

> > > ### Author Response · Authors · 2024-11-22
> > > **Common Response (part 4/4)**
> > >
> > > **5. [**BxVf**, **jrRB**] What is the criterion used to select the hyperparameter λ?**
> > >
> > > $\lambda$ is chosen to ensure gradients from the score-matching objective $L_{\text{score}}$ and the conditional independence objective $L_{\text{CI}}$ are balanced in magnitude. So it can be set as λ = $\frac{L_\text{score}}{L_\text{CI}}$. We describe this in Appendix C.3. As a rule of thumb, a simplified value: λ = $L_\text{score}$ × 4000 also worked well in practice. Moreover, setting $\lambda$ to very high values strongly enforced conditional independence at the cost of lower CS, generating images similar to unconditional generation.
> > >
> > > **6. [**5mnE**, **BxVf**] Open set vs closed set composition**
> > >
> > > By open-set composition, we believe that the reviewers meant the task of generating images where some attribute $C_1$  takes a value $c^*$ when $C_1 = c^*$ has never appeared in the training set.
> > >
> > > As opined by 5mnE, open-set compositions are beyond the scope of the paper. Our primary motivation was to ensure logical compositionality in scenarios where it is expected i.e., compositions in closed-sets as defined in Definition 1 (L141-145 in the main paper) (see Figure 3 A-D in (Wiedemer et al., 2024)). In such scenarios, we identified a fundamental flaw in vanilla diffusion models that limited logical compositionality even in the easiest case i.e., compositions of closed-set support where all combinations of attributes are observed. In contrast, we also showed that CoInD can generate images in the same closed-sets.

---

### Author Response · Authors · 2024-11-25
**Could the reviewers please let us know whether our responses have addressed their concerns?**

We would like to thank all the reviewers for their valuable questions, which have significantly contributed to improving our manuscript.

As the discussion phase draws to a close, we kindly request the reviewers to let us know if we have addressed their concerns. If there are any additional questions regarding the manuscript, we would be happy to provide further clarification.

If we have addressed your concerns, we would greatly appreciate it if you could consider re-evaluating your initial scores.

---

### Meta-Review · Area_Chair_9HTE · 2024-12-18

**Metareview:**

Authors hypothesize that diffusion models struggle with compositionality due to not respecting conditional independence relations in the data. They use a theoretically-motivated Wasserstein loss proxy to encourage this with hopes that it would help with decompositionality. Here are my own comments:

"the principle of independent causal mechanisms"
This really has nothing to do with the contents of the paper. This principle is a meta assumption that "nature" samples each mechanism independently from each other, in some sense. Authors already assume a well-defined causal model with independent features so this is unrelated and confusing. I recommend authors removing these statements, which repeat several times.

On a related note, "Last, we assume that C1, . . . , Cn affect X independently
of each other." does not mean anything formal. Please remove this. independence of causal mechanism the way you are imagining it will not give you independence statement you need. Only unfaithfulness or deterministic inverse from X will.

Authors consider an unfaithful distribution in the sense that features are independent given X although they are d-connected. This might be OK in some settings. Especially, when these features are deterministic maps from X, such as reading off whether someone is wearing eyeglasses or not from an image. The authors indeed make this assumption, that there is a deterministic map from X to features. Using this they can utilize joint independence of features given X. However, note that the conditionals p(Ci|X) simply become indicators. These are not arbirary distributions. One explanation on why the proposed method still works might be that in the diffusion model and also to some degree in ground truth these may be non-deterministic.

**Additional Comments On Reviewer Discussion:**

Most reviewers are leaning towards, perhaps an unenthusiastic, acceptance of the paper. The authors have put real effort into the rebuttals adding real-world image experiments and more. I am interpreting reviewers' lack of engagement as them not being against acceptance of this paper. The paper is not going to be the ultimate paper in the area, but starting a discussion about independence relations' violations by diffusion models is worthy of acceptance in my opinion. Especially the experimental evidence helps push the paper above the acceptance bar.

---

> ### Public Comment · ~Sachit_Gaudi1 · 2025-03-02
> **Refined Manuscript: Addressing All Reviewer and Meta-Reviewer Comments**
>
> **Code and Reproducibility:** We have released the complete code to replicate all experiments along with checkpoints that enable others to reproduce our experimental results.
>
> **Simulation Results:** Reviewer D8GZ requested a simulation study. To enhance inner workings of our approach, we now present an analysis in Section F.2 where we explain why CoInD works using 2D Gaussian simulated dataset. We selected a 2D Gaussian because its score function can be derived in closed form.
>
> **Clarifications:** During the review process, we identified additional relevant work and have incorporated these references into the revised manuscript. Additionally, to address potential confusion among readers, we have refined the problem statement and clarified our underlying assumptions. In response to Reviewer [D8GZ, 5mnE]'s concerns regarding the failure mode (RQ1), we have rewritten Section 3 to improve clarity for the readers.
>
> Additionally we would like to clarify questions raised by the meta reviewer:
>
> **Meta Assumption about Conditional Independence:** The conditional independence assumption is a common modelling assumption applicable in many scenarios. However, it is rather not a meta assumption per se. For example, when modelling velocity, acceleration, and force of an object on Earth, these attributes are not conditionally independent. Therefore, we want to explicitly state to settings where CoInD is applicable.
>
> **Clarification of Independence Statement:** The statement “Last, we assume that C₁, …, Cₙ affect X independently of each other” was recognised as informal and potentially confusing. We have revised this description to provide a more formal explanation of our assumption regarding the independent contributions of the components. We also slightly modified causal graph to reflect these assumptions.
>
> **Unfaithful Distributions and Deterministic Maps:** ``Authors consider an unfaithful distribution in the sense that features are independent given X although they are d-connected ….. Especially, when these features are deterministic maps from X, such as reading off whether someone is wearing eyeglasses or not from an image." Although “deterministic maps” is one particular case  to state of more general case, where the underlying generative model has a Jacobian of at least rank m ( number of attributes ) [1]. We have modified the problem definition to reflect meta reviewer’s comments.
>
> **Conditionals $p(C_i|X)$  simply become indicators:** While the true conditionals $p(C_i|X)$  reduce to indicator functions, the learned approximations $p_\theta(C_i|X)$  are obtained via an implicit classifier within the diffusion model are not encouraged to learn these indicator functions. Importantly, the diffusion model is not incentivised to learn the correct $p_\theta(C_i|X)$ , and as commonly observed, machine learning models fail to capture the true indicator function due to confounder shift. In the partial support setting of Colored MNIST the diffusion model  relies on color information to guide the digit generation. The following results illustrate this behaviour:
>
> | `Implicit classifier Accuracy` $\rightarrow$  `    ` `    ` `    `    `Method` $\downarrow$ | Train $p_\theta(\text{digit}\mid X)$ | Train $p_\theta(\text{color}\mid  X)$  | Unseen $p_\theta(\text{digit}\mid  X)$ | Unseen $p_\theta(\text{color} \mid  X)$   |
> | --- | --- | --- | --- | --- |
> | Diffusion | 0.959 | 0.991  |  0.136   |  0.874       |
> | CoInD | 0.922  | 0.922  |  **0.329**   | 0.765 |
>
>
>
> The decrease in accuracy of implicit classifier $p_\theta(\text{digit}|X)$  accuracy in the unseen set compared to the training set indicates that the inherent classifier in the diffusion model relies on spurious correlations. Diffusion models are not immune to these spurious correlations our method addresses this issue by correcting these spurious correlations, rather than attributing the improvement to any “magic” of the diffusion process or its non-deterministic nature.
>
> We appreciate the valuable feedback from AC and the reviewers, which has significantly contributed to the clarity of our work.
>
> [1] Thaddäus Wiedemer, Prasanna Mayilvahanan, Matthias Bethge, and Wieland Brendel. Compositional generalization from first principles. In Advances in Neural Information Processing Systems, 2024

---

### Decision · Program_Chairs · 2025-01-22

Accept (Poster)